# ER proteins decipher the tubulin code to regulate organelle distribution

Pengli Zheng[1]✉, Christopher J. Obara[2], Ewa Szczesna[3], Jonathon Nixon-Abell[1,2,7], Kishore K. Mahalingan[3], Antonina Roll-Mecak[3,4], Jennifer Lippincott-Schwartz[2] & Craig Blackstone[1,5,6]✉

Organelles move along differentially modified microtubules to establish and maintain their proper distributions and functions[1,2]. However, how cells interpret these post-translational microtubule modification codes to selectively regulate organelle positioning remains largely unknown. The endoplasmic reticulum (ER) is an interconnected network of diverse morphologies that extends promiscuously throughout the cytoplasm[3], forming abundant contacts with other organelles[4]. Dysregulation of endoplasmic reticulum morphology is tightly linked to neurologic disorders and cancer[5,6]. Here we demonstrate that three membrane-bound endoplasmic reticulum proteins preferentially interact with different microtubule populations, with CLIMP63 binding centrosome microtubules, kinectin (KTN1) binding perinuclear polyglutamylated microtubules, and p180 binding glutamylated microtubules. Knockout of these proteins or manipulation of microtubule populations and glutamylation status results in marked changes in endoplasmic reticulum positioning, leading to similar redistributions of other organelles. During nutrient starvation, cells modulate CLIMP63 protein levels and p180–microtubule binding to bidirectionally move endoplasmic reticulum and lysosomes for proper autophagic responses.

Eukaryotes compartmentalize cellular functions within distinct organelles, and regulation of organelle position is critical for cell health. Organelles are transported bidirectionally by motor and adaptor proteins along microtubules[1], which are modulated by multiple post-translational modifications that comprise part of the 'tubulin code'. Although this code has been implicated in cargo selection and directed organelle movement[2], how it is decoded to mediate transport and control distribution remains largely unknown.

Endoplasmic reticulum (ER) comprises structurally and functionally divergent membrane compartments that include interconnected tubules, perinuclear matrices and sheets, and the nuclear envelope[3,7]. The ER is a compelling candidate for exploiting the complexity of the tubulin code, since it spreads throughout the cytoplasm in association with microtubules and makes abundant organelle contacts[8–10]. Most studies of ER shaping and organelle contacts have emphasized peripheral tubular ER. How the denser perinuclear ER is shaped and asymmetrically distributed remains largely unknown, although three ER membrane-bound proteins—CLIMP63, p180 and KTN1—localize prominently to perinuclear ER and are considered sheet-forming proteins[11]. Even so, depletion of CLIMP63 may paradoxically lead to the expansion of ER matrices or sheets in the periphery[11,12], and perinuclear ER matrices or sheets remain abundant even upon simultaneous knockdown of all three proteins, prefiguring more complex functional roles[11].

## CLIMP63, p180 and KTN1 position ER

We used CRISPR–Cas9 to knock out these proteins in human U2OS cells stably expressing the ER marker mEmerald–Sec61β (Extended Data Fig. 1a, b). As previously reported[11,12], peripheral ER in CLIMP63-knockout cells is populated with increased numbers of dense matrices or sheets—a 'dispersed' phenotype. KTN1 knockout also disperses ER, whereas p180-knockout cells exhibit a contrasting 'clustered' ER phenotype, with the peripheral network remaining tubular and perinuclear ER collapsing asymmetrically into a smaller area at one side of the nucleus (Fig. 1a, Extended Data Fig. 1c, d). These morphologic changes are not secondary to alterations in levels of other ER-shaping proteins or cell cycle disruption (Extended Data Fig. 1b, e, f). Double knockout of CLIMP63 and KTN1 substantially disperses ER. Conversely, ER in CLIMP63 and p180 double-knockout cells resembles the wild type, consistent with their opposing single-knockout phenotypes. Surprisingly, p180 and KTN1 double knockout causes more ER clustering than in p180-knockout cells (Fig. 1a, Extended Data Fig. 1d), suggesting a more complex interplay. In CLIMP63–p180–KTN1 triple-knockout cells, high-density ER matrices or sheets are abundant in the perinuclear region (Fig. 1a), although perinuclear ER appears less evenly distributed compared with wild-type cells, with 'hot spots' (Extended Data Fig. 1g) that may reflect ER positioning defects.

[1]Cell Biology Section, Neurogenetics Branch, National Institute of Neurological Disorders and Stroke, National Institutes of Health, Bethesda, MD, USA. [2]Janelia Research Campus, Howard Hughes Medical Institute, Ashburn, VA, USA. [3]Cell Biology and Biophysics Section, National Institute of Neurological Disorders and Stroke, National Institutes of Health, Bethesda, MD, USA. [4]Biochemistry and Biophysics Center, National Heart, Lung and Blood Institute, National Institutes of Health, Bethesda, MD, USA. [5]MassGeneral Institute for Neurodegenerative Disease, Massachusetts General Hospital, Charlestown, MA, USA. [6]Department of Neurology, Massachusetts General Hospital and Harvard Medical School, Boston, MA, USA. [7]Present address: Cambridge Institute for Medical Research, Cambridge, UK. ✉e-mail: shalemander@gmail.com; cblackstone@mgh.harvard.edu

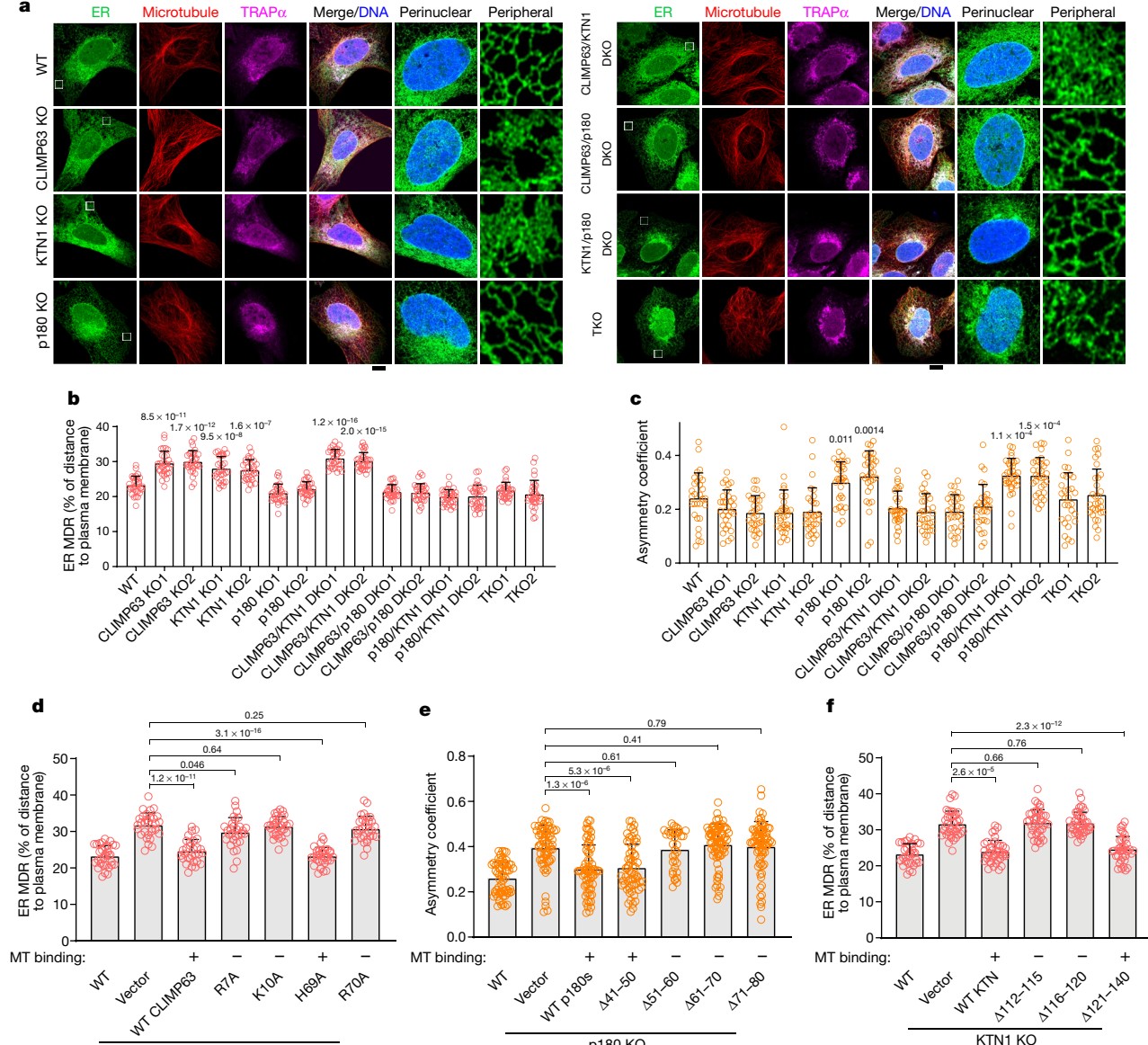

**Fig. 1 | CLIMP63, p180 and KTN1 differentially regulate ER morphology.**
**a**, Representative images of wild-type (WT), knockout (KO), double knockout (DKO) or triple knockout (TKO) of CLIMP63, p180 and/or KTN1 in U2OS cells stably expressing mEmerald–Sec61β (green, ER marker) and co-labelled with DAPI (blue, nuclear marker), anti-α-tubulin (red, microtubule marker) and anti-TRAPα (magenta, rough ER marker). Perinuclear and peripheral regions (left, outlined) are enlarged on the right. Scale bar, 10 μm. **b**, **c**, ER MDR (**b**) and asymmetry (**c**) (see Supplementary Text) in cells as in **a**. $n = 30$ cells.
**d**–**f**, Quantifications of ER morphology in wild-type or CLIMP63-knockout

(**d**), p180-knockout (**e**) or KTN1-knockout (**f**) cells expressing the indicated CLIMP63, p180 or KTN1 constructs. Since recombinant KTN1 levels are typically very low, only cells with detectable KTN1–mApple signals were quantified. Ability (+) or inability (−) of the mutants to bind microtubules (MT) is indicated. $n = 32, 30, 30, 30, 31, 30$ and $30$ cells (left to right) in **d**; $n = 59, 62, 64, 61, 34, 70$ and $70$ (left to right) in **e**; and $n = 32, 38, 38, 39, 40$ and $36$ (left to right) in **f**. Data are mean ± s.d. with individual data points shown. Two-tailed $t$-test; $P$ values are shown.

To quantitatively assess changes in ER morphology and distribution, we devised complementary algorithms. First, we harnessed a statistical approach based on probability density estimation to analyse spatial distributions of fluorescently labelled ER and other organelles. Next, we used an experimentally derived spatial probability mass function, which quantifies fluorescence changes across an image, to calculate the radial distribution and degree of cellular asymmetry of organelles (Extended Data Fig. 2a–g, Supplementary Text). Single or double knockout of CLIMP63 and KTN1 increases ER mean distribution radius (MDR) (Fig. 1b), indicating that ER is spread more peripherally. By contrast, p180 knockout or p180 and KTN1 double knockout increases ER asymmetry (Fig. 1c). Quantification assessing the rough ER marker

TRAPα instead of mEmerald–Sec61β shows similar results (Extended Data Fig. 2h, i). Microtubule MDR and asymmetry change only slightly (Extended Data Fig. 2j–m).

## ER proteins bind subsets of microtubules

We assessed microtubule binding of numerous ER proteins by co-sedimentation. CLIMP63 and p180, both known microtubule-binding proteins[13,14], co-sediment with microtubules as expected. KTN1 also sediments robustly with microtubules (Extended Data Fig. 3a, b). Since full-length p180 (p180L) is degraded during cell lysis (Extended Data Fig. 3a, c), we used a smaller, more stable splice variant (p180s) that

lacks the numerous ribosome-binding decapeptide repeats present in p180L (Extended Data Figs. 1a, 3b); p180s was undetectable by immunoblotting in the cell lines studied, facilitating identification of recombinant protein (Extended Data Fig. 3d). For each protein, we mapped microtubule-binding domains; only wild-type proteins or mutants capable of microtubule binding restored ER morphology in corresponding knockout cell lines (Fig. 1d–f, Extended Data Figs. 3, 4). For instance, CLIMP63 missense mutants R7A, K10A and R70A did not bind microtubules or suppress ER distribution defects in CLIMP63-knockout cells, whereas CLIMP63(H69A), which binds microtubules, rescues the phenotype. A phosphomimetic CLIMP63 mutant defective in microtubule binding[13] also did not rescue ER distribution defects (Extended Data Fig. 4a–d). For KTN1, only the deletion mutant that binds microtubules suppressed the abnormal ER phenotype (Fig. 1f, Extended Data Fig. 4g–i). Finally, p180s lacking the kinesin-1 binding domain still suppressed the clustered ER phenotype in p180-knockout cells (Extended Data Fig. 4j–l). Thus, despite distinct phenotypes, ER morphology changes in CLIMP63-, p180- and KTN1-knockout cells are likely to all reflect alterations in microtubule binding.

We hypothesized that these proteins bind different microtubule populations and used a proximity ligation assay (PLA) to visualize their microtubule associations in cells (Extended Data Fig. 5a–c). We depleted centrosomal microtubules using centrinone B treatment[15], and Golgi-derived microtubules by knocking down AKAP450[16] (Extended Data Fig. 5d). We found that microtubule association of CLIMP63 was sensitive to centrosome depletion but not Golgi microtubule depletion, whereas KTN1–microtubule association was sensitive to both; p180–microtubule association was not sensitive to depletion of either centrosomes or Golgi microtubules (Extended Data Fig. 5e–h). Admittedly, these microtubule subsets can be interdependent, and centrosome depletion can boost AKAP450-dependent microtubule nucleation at the Golgi[16,17]. Even so, disrupting Golgi microtubules did not alter centrosome activity[17] or CLIMP63–microtubule association (Extended Data Fig. 5e, f).

We inferred that CLIMP63 preferentially binds centrosomal microtubules, KTN1 preferentially binds perinuclear microtubules derived from either centrosome or Golgi, and p180 preferentially binds more peripheral microtubules regardless of origin. In this scenario, PLA distributions for microtubules with CLIMP63 should be more asymmetric than with p180 or KTN1, and PLA distributions for p180 and microtubules should be more dispersed. Indeed, PLA signals for CLIMP63 and microtubules were more asymmetric than those for p180 and KTN1 with microtubules. However, PLA MDR for p180 with microtubules resembled that for KTN1 with microtubules in wild-type cells (Extended Data Fig. 5i). We reasoned that because ER is densely packed perinuclearly in wild-type cells, PLA signals were also mostly perinuclear, making differences challenging to identify. As a workaround, we quantified PLA distributions in CLIMP63-knockout cells, in which ER is more dispersed (Fig 1a); MDR for p180–microtubule PLA signals was larger than MDR for KTN1–microtubule PLA signals (Extended Data Fig. 5j), suggesting that p180 binds more peripheral microtubules than KTN1. Consistent with this specificity, centrosome depletion led to highly dispersed ER in wild-type but not p180-knockout cells, whereas depletion of Golgi-derived microtubules clustered ER in wild-type but not CLIMP63 and KTN1 double-knockout cells (Extended Data Fig. 5k–n).

## Graded binding to modified microtubules

For regulatory specificity, microtubules undergo reversible post-translational modifications including acetylation, detyrosination and glutamylation, which together constitute key elements of the tubulin code[2]. Although CLIMP63, p180 or KTN1 knockout did not affect overall levels of these modifications, tubulin polyglutamylation was decreased in centrosome or Golgi microtubule-depleted cells (Extended Data Fig. 6a–c). We thus considered whether variations in tubulin glutamylation underlie binding selectivity for different microtubule populations and differential effects of the proteins on ER distribution.

CLIMP63 overexpression caused tight ER–microtubule alignment[13] that is suppressed in centrosome-depleted cells, whereas p180s or KTN1 overexpression did not trigger ER–microtubule alignment (Extended Data Fig. 6d–f). Co-expression of TTLL4, which monoglutamylates microtubules[18] (Extended Data Fig. 6g, h), slightly enhanced ER–microtubule alignment in p180- but not KTN1-overexpressing cells (Extended Data Fig. 6d, e). By contrast, co-expression of TTLL7, which polyglutamylates microtubules[19,20] (Extended Data Fig. 6g, h), led to significant microtubule–ER alignment in both p180- and KTN1-overexpressing cells (Extended Data Fig. 6d, e). Although co-expression of TTLL7 slightly enhanced ER-microtubule alignment in CLIMP63-overexpressing cells, co-expression with TTLL4 or microtubule de-glutamylases CCP1 or CCP5 (CCP1 shortens glutamate chains, whereas CCP5 is thought to remove the branch-point glutamate[21], Extended Data Fig. 6g, i) did not influence ER–microtubule alignment (Extended Data Fig. 6d, e). Since CLIMP63–microtubule associations as assessed using PLA were unaffected by overexpression of TTLL4, TTLL7, CCP1 or CCP5 (Extended Data Fig. 6j), we inferred that CLIMP63–microtubule binding is not altered by changes in microtubule glutamylation. PLA signals for KTN1–tubulin were significantly increased by TTLL7 but not TTLL4 and decreased in cells overexpressing CCP1 or CCP5 (Extended Data Fig. 6k). By contrast, PLA signals of p180–tubulin were slightly increased by TTLL4 overexpression, markedly increased by TTLL7, slightly decreased by CCP1, and significantly decreased by CCP5 (Extended Data Fig. 6l). We conclude that KTN1 and p180 respond differentially to glutamylation levels, with KTN1 preferentially associating with polyglutamylated versus monoglutamylated microtubules, whereas p180 broadly associates with mono- and polyglutamylated microtubules.

We purified fragments of p180, KTN1 and CLIMP63 containing their microtubule-binding domains (Extended Data Fig. 6m, n) and investigated binding to differentially glutamylated microtubules in vitro, using TTLL6 to generate microtubules functionalized with polyglutamate chains of various lengths primarily on α-tubulin[18] (Extended Data Fig. 6o). Both p180 and KTN1 showed substantial increases in binding to microtubules polyglutamylated by TTLL6, with only background binding to unmodified microtubules (Fig. 2a, b). Moreover, as the average glutamate number $<n_E>$ on α-tubulin increased from 3.5 to 8.3, binding affinities increased in lockstep, with 2.7- and 5.6-fold increases for p180 and KTN1, respectively. Notably, p180 had a 2.9-fold stronger affinity than KTN1 for microtubules with shorter chains (Fig. 2b). Next, we interrogated how microtubule binding is affected by β-tubulin monoglutamylation induced by TTLL4 and polyglutamylation induced by TTLL7[18,19] (Extended Data Fig. 6p). Both p180 and KTN1 showed binding preferences toward microtubules functionalized with polyglutamates by TTLL7 (Fig. 2c, d) but weaker binding to TTLL4-modified microtubules (Fig. 2c, e). Of note, p180 exhibited higher in vitro binding (3.9-fold) to microtubules monoglutamylated by TTLL4 compared with KTN1, whereas p180 and KTN1 bound similarly to microtubules polyglutamylated by TTLL7 (Fig. 2c, e). This difference was evident even though numbers of glutamates added by TTLL4 (mean of 1.5) and TTLL7 (mean of 1.2) were similar (Extended Data Fig. 6p), indicating that KTN1 prefers polyglutamate chains introduced by TTLL7 to multiple monoglutamates introduced by TTLL4. At higher glutamylation levels, both KTN1 and p180 formed patches on the microtubule lattice (Fig. 2c), indicative of cooperative binding that may be physiologically relevant in cells when these molecules are tethered and concentrated on the ER membrane. In contrast to p180 and KTN1, CLIMP63 was less responsive to microtubule glutamylation; it lacked detectable binding to unmodified and polyglutamylated microtubules with $<n_E>$ of 2.7, exhibiting microtubule binding only when $<n_E>$ reached 3.8 (Extended Data Fig. 6q, r). Thus, hyperglutamylation can enhance CLIMP63–microtubule binding in vitro, but since overexpression of CCP1 or CCP5 did not seem to affect

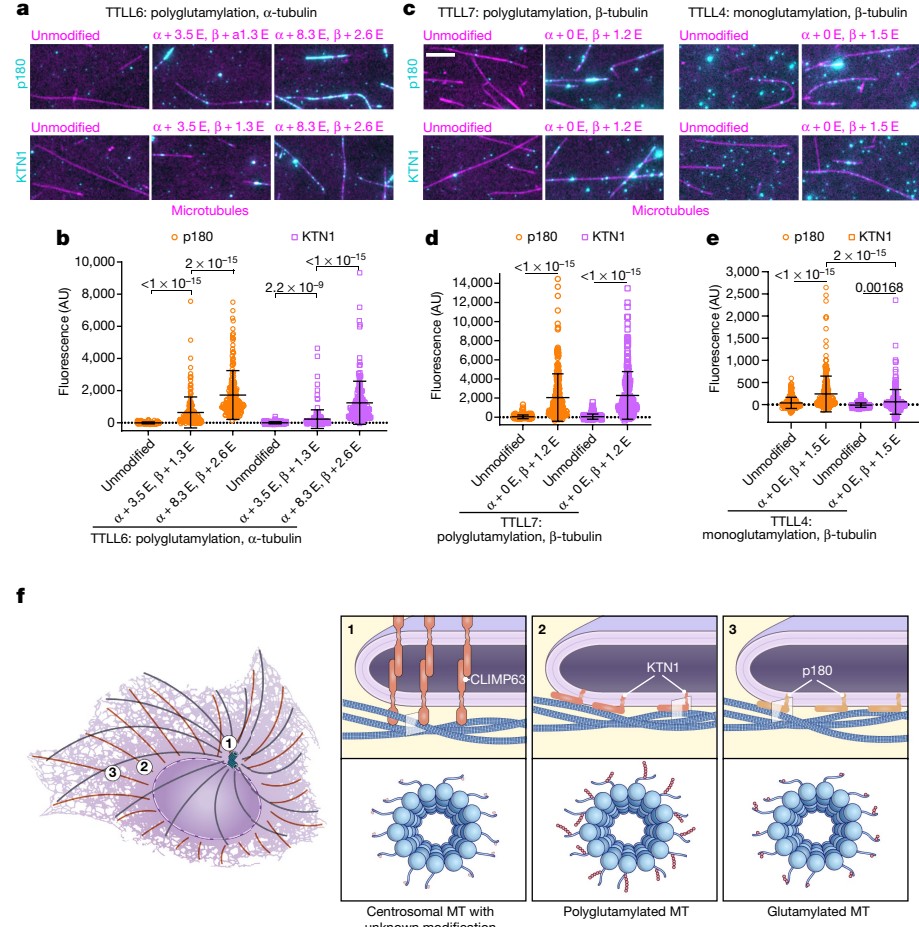

**Fig. 2 | p180 and KTN1 bind glutamylated and polyglutamylated microtubules, respectively. a**, Representative micrographs of p180 and KTN1 fragments (cyan) binding to unmodified microtubules or microtubules glutamylated in vitro by TTLL6 (magenta). Internal reflection microscopy images for the microtubule channel were background subtracted and inverted. Average numbers of glutamate molecules (E) added to microtubules as quantified from mass spectroscopy data in each group are indicated. Scale bar, 5 µm. **b**, Binding of p180 and KTN1 fragments to unmodified and TTLL6-glutamylated microtubules. $n = 110$ (unmodified), 165 ($\alpha + 3.5$ E, $\beta + 1.3$ E) and 112 ($\alpha + 8.3$ E, $\beta + 2.6$ E) microtubules for p180; $n = 141$ (unmodified), 186 ($\alpha + 3.5$ E, $\beta + 1.3$ E) and 156 ($\alpha + 8.3$ E, $\beta + 2.6$ E) microtubules for KTN1. **c**, Representative micrographs of p180 and KTN1 microtubule interacting fragments (cyan) showing binding to unmodified microtubules or microtubules glutamylated by TTLL4 or TTLL7 (magenta). KTN1 and p180 are shown with different brightness/contrast settings for TTLL4- and

TTLL7-modified microtubules, reflecting large differences in binding between mono- and polyglutamylated microtubules. Scale bar, 5 µm. **d**, **e**, Affinities of p180 and KTN1 for unmodified microtubules and microtubules glutamylated by TTLL7 (**d**) or TTLL4 (**e**). The x-axis shows weighted averages of glutamate residues attached to α- and β-tubulin. $n = 185$ (unmodified) and 117 ($\alpha + 0$ E, $\beta + 1.2$ E) microtubules for p180; $n = 128$ (unmodified) and 225 ($\alpha + 0$ E, $\beta + 1.2$ E) microtubules for KTN1; $n = 179$ (unmodified) and 237 ($\alpha + 0$ E, $\beta + 1.5$ E) microtubules for p180; $n = 163$ (unmodified) and 224 ($\alpha + 0$ E, $\beta + 1.5$ E) microtubules for KTN1. **f**, CLIMP63 binds centrosomal microtubules, KTN1 binds perinuclear polyglutamylated microtubules, and p180 can bind peripheral microtubules with less glutamylation. Together, these proteins maintain proper asymmetric ER distribution, which regulates organelle distributions. MT, microtubules. Data are mean ± s.d. with individual data points shown. Kruskal–Wallis (**b**) or Mann–Whitney tests (**d**, **e**); P values are shown.

CLIMP63–microtubule binding in cells (Extended Data Fig. 6e, j), and centrosome depletion suppressed CLIMP63 overexpression-mediated ER–microtubule alignment (Extended Data Fig. 6f), a different tubulin modification or interaction probably mediates preferential binding of CLIMP63 with centrosome microtubules.

## Glutamylation regulates ER distribution

Perinuclear microtubules harbour more polyglutamylation, whereas monoglutamylation is generally more prominent peripherally (Extended Data Fig. 7a–d), consistent with KTN1 binding preferentially to perinuclear microtubules and p180 binding to peripheral microtubules. TTLL overexpression glutamylated microtubules throughout the cell, eliminating the relatively discrete perinuclear distribution of polyglutamylated microtubules and thus drawing ER towards the cell periphery, whereas overexpression of CCP1 or CCP5

decreased binding of p180 and KTN1 to perinuclear microtubules; thus, overexpression of TTLL, CCP1 or CCP5 all lead to dispersed ER (Extended Data Fig. 7e–g). In p180 and KTN1 double-knockout cells, overexpression of TTLL4, CCP1 or CCP5 did not change ER MDR (Extended Data Fig. 7h), yet TTLL7 overexpression still had minor effects, possibly through other pathways. When TTLL7 is overexpressed, KTN1 should bind all microtubules, rather than preferring perinuclear ones. Thus, with TTLL7 overexpression, KTN1 knock-out resulted in less dispersed ER (Extended Data Fig. 7i). We also knocked down CCP5, which increases tubulin glutamylation. Similar to TTLL4 overexpression, CCP5 knockdown dispersed ER (Extended Data Fig. 7j–m).

We examined several cell lines widely used in ER morphology studies to assess whether they had different microtubule glutamylation levels. Notably, COS7 cells had particularly high polyglutamylation levels (Extended Data Fig. 7c), and although polyglutamylation

in COS7 cells remained relatively more perinuclear compared to monoglutamylation and microtubule distribution, the difference was much less than in U2OS cells (Extended Data Fig. 7b, d). We hypothesized that KTN1 knockout in COS7 cells would show a distinct ER phenotype, possibly mimicking TTLL7 overexpressing cells (Extended Data Fig. 7i). Indeed, although knockout of CLIMP63 or p180 in COS7 cells showed similar phenotypes as in U2OS cells, KTN1 knockout in COS7 cells led to clustered ER (Extended Data Fig. 8a–d), in contrast to dispersed ER in KTN1 knockout U2OS cells. Moreover, overexpression of CCP6 (which has similar activity to CCP1) also led to clustered ER (Extended Data Fig. 8e–h). We conclude that CLIMP63, p180 and KTN1 preferentially bind centrosomal, polyglutamylated and glutamylated microtubules, respectively, to cooperatively distribute ER (Fig. 2f).

## Organelle positioning and glutamylation

Live imaging of six organelles[22] simultaneously revealed that most have a distribution similar to ER (Extended Data Fig. 9a), suggesting that ER might broadly regulate organelle distribution. Notably, in CLIMP63-, p180- and KTN1-knockout cells, all organelles that we examined exhibited similar distribution changes to those of ER—more dispersed in CLIMP63- or KTN1-knockout cells and more asymmetric in p180-knockout cells (Extended Data Fig. 9b–d). Moreover, CCP1 overexpression, which disperses ER, also increased MDR for lysosomes, mitochondria and peroxisomes in wild-type cells but not in p180 and KTN1 double-knockout cells (Extended Data Fig. 9e–g). Thus, perinuclear ER morphology specifies the distributions of other organelles downstream of microtubule glutamylation.

## ER and lysosome movements in autophagy

Perinuclear lysosome clustering, a signature event in early autophagy, is important for proper autophagic flux[23,24]. Similar to lysosomes, ER migrates perinuclearly during early autophagy, and subsequently redistributes to the periphery (Fig. 3a, b, Supplementary Video 1). CLIMP63 levels increased significantly during early autophagy, and this increase did not appear to require new protein synthesis or inhibition of lysosomal or proteasomal degradation (Fig. 3c, Extended Data Fig. 10a). CLIMP63 knockout prevented ER movement toward the perinuclear region (Fig. 3b) and suppressed autophagosome–lysosome fusion and autophagic degradation, but not lysosomal activity (Extended Data Fig. 10b–f). Since p180 and KTN1 protein levels remained unchanged (Extended Data Fig. 10a), we examined their binding to microtubules. KTN1–microtubule binding did not change upon nutrient starvation, but p180–microtubule binding increased (Fig. 3d). Consistently, ER and lysosomes in p180-knockout cells remained perinuclear (Fig. 3b), and thus p180-knockout cells showed defects in recovery of mTOR signalling[24] after nutrient re-supplementation, but not in autophagic degradation (Extended Data Fig. 10g, h).

Microtubule modification levels were unaffected by starvation (Extended Data Fig. 10i). Notably, the ribosome-binding region of p180L (the major cellular isoform) includes 41 positively charged decapeptide repeats (Fig. 3e). We hypothesized that this region is occupied by ribosomes under normal cellular conditions but then ribosomes dissociate during starvation, exposing these positively charged regions that can then bind microtubules (Fig. 3f). Indeed, starvation significantly decreased p180–ribosome binding (Fig. 3g, Extended Data Fig. 10j); puromycin treatment, which dissociates ribosomes from ER in fed conditions, markedly enhanced p180–microtubule binding (Fig. 3d). In contrast to p180s, which lacks most ribosome-binding decapeptide repeats, p180L overexpression increased ER–microtubule alignment. This alignment was enhanced by starvation or puromycin treatment (Extended Data Fig. 10k–m), further indicating that ribosome-binding repeats of p180L bind microtubules upon ribosome dissociation.

## Discussion

Peripheral ER network morphology is maintained by hydrophobic hairpin domain proteins (reticulons and receptor expression enhancing proteins (REEPs)) that shape the tubules. The polygonal network is generated via atlastin-mediated tethering and fusion of tubules at three-way junctions and distributed via cytoskeletal interactions[3,25,26]. Much less is known about the dynamic organization of perinuclear ER. Although microtubules have important roles in establishing ER morphology, most studies have emphasized peripheral tubular ER[27] or identification of ER proteins that bind microtubules[13,14,28]. Proteins including CLIMP63, p180 and KTN1 are enriched in dense, sheet-like perinuclear ER, and they each bind microtubules. However, phenotypes of cells deficient in these proteins differ considerably, raising the question of how microtubule-binding specificity is maintained. Here we demonstrate that CLIMP63, p180 and KTN1 preferentially bind different subsets of microtubules to maintain perinuclear ER in its characteristic distribution, explaining the differential effects of their absence. Furthermore, depletion of centrosome or Golgi-derived microtubules has distinct effects on the microtubule binding of these three proteins.

Microtubule diversity can be achieved via different tubulin gene products, differential interactions with microtubule-associated proteins and numerous post-translational modifications[2]. Modifications are dynamic and rapidly reversible, but evidence for how they affect microtubule-related functions has been limited[2]. We have shown here that KTN1 preferentially binds perinuclear polyglutamylated microtubules with long glutamate chains, whereas p180 binds glutamylated microtubules with either short or long chains. By contrast, CLIMP63 has a higher threshold for response to microtubule glutamylation. We cannot exclude that increased affinity at higher glutamate numbers for TTLL7-modified microtubules stems from additional chains that TTLL7 initiates on tubulin tails, and not only from introduction of longer chains. Conversely, p180 is more sensitive to any increase in glutamate numbers on the tubulin tail and robustly binds both mono- and polyglutamylated microtubules. This differential effect on microtubule binding according to glutamylation state has previously been observed for the microtubule-severing ATPase spastin[29] and may represent a general feature of this modification, enabling fine tuning of molecular interactions. Thus, a small difference in the number of glutamates added to tubulin side chains may exert a substantial qualitative effect on ER distribution. Other ER-localized, microtubule-binding proteins[30] are likely to contribute to overall cellular ER positioning. Indeed, even in p180 and KTN1 double-knockout cells, TTLL7 overexpression still disperses ER, suggesting the involvement of other ER proteins. Moreover, tubular ER selectively moves along acetylated microtubules[27], further indicating that ER distribution is broadly sensitive to microtubule modifications.

The ability of cells to dynamically control ER distribution through differential microtubule modifications has important functional implications. For instance, p180 regulates microtubule remodelling in axons[31], and axonal microtubules are highly glutamylated[2]. Thus, p180 may affect microtubule remodelling by differentially recognizing glutamylated axonal microtubules. Of note, although dysregulation of ER shaping and microtubule polyglutamylation lead to different neurodegenerative diseases[32,33], these diseases share some similar cellular phenotypes, including mitochondrial distribution defects and axon degeneration, suggesting possible convergence.

When ER positioning is disrupted, distributions of other organelles are affected. Microtubules have key roles in organelle distribution[1], and their ability to selectively distribute organelles relies on a tubulin code. Our results indicate that ER distribution is mediated via specific membrane-bound proteins with differential binding to different levels and types of microtubule glutamylation, broadly affecting distributions of most other organelles. ER thus interprets the tubulin code to regulate movement and positioning of cellular organelles. Rather than imbuing each organelle with its own sensing and response mechanisms, cells

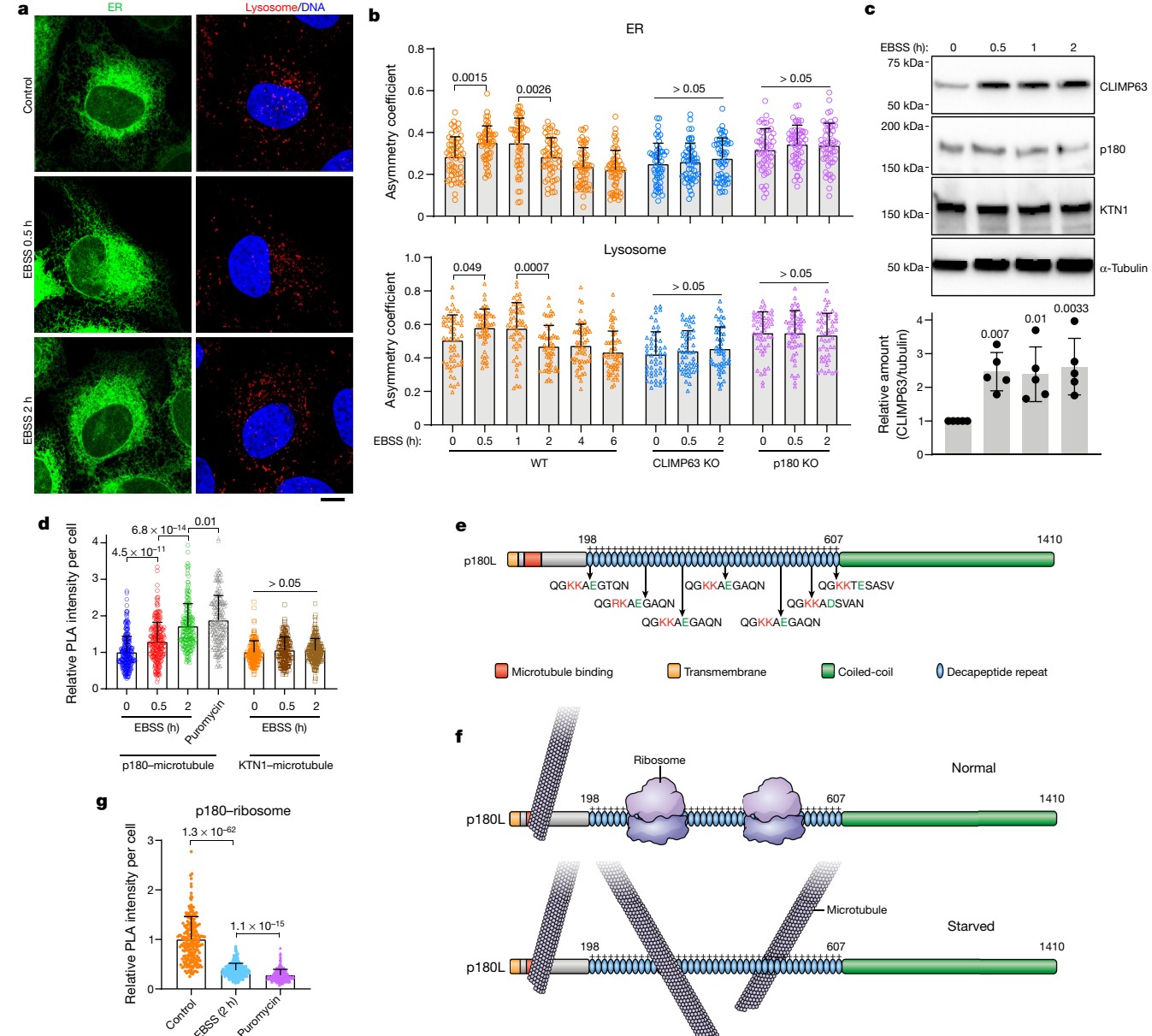

**Fig. 3 | ER distribution changes during autophagy. a**, U2OS cells stably expressing mEmerald−Sec61β (green, ER marker) were starved in EBSS for 0, 0.5, or 2 h and labelled with anti-LAMP1 (red, lysosome marker). Scale bar, 10 μm. **b**, ER and lysosome distribution in wild-type, CLIMP63- or p180-knockout cells starved in EBSS for the indicated times. $n = 55, 53, 52, 52, 53, 54, 51, 50, 51, 50, 46$ and 44 cells (left to right) for both ER and lysosome. **c**, Top, U2OS cells were starved in EBSS for 0–2 h and immunoblotted. Bottom, protein levels relative to α-tubulin. $n = 5$ experiments. **d**, Average intensities of PLA for p180 and KTN1 with α-tubulin at 0, 0.5, or 2 h of EBSS starvation or puromycin treatment (2 μg ml$^{-1}$ for 2 h). $n = 273, 254, 188, 180, 143, 144, 182$ cells (left to right). **e**, Schematic of p180 domain composition. The ribosome-binding domain includes 41 positively charged decapeptide repeats, which can potentially bind microtubules once ribosomes are dissociated. Amino acid sequences for several repeats are shown, with positively charged residues in red and negatively charged residues in green. Repeat sequences vary slightly but are all positively charged. **f**, Under normal cellular conditions, p180L ribosome-binding repeats are occupied by ribosomes and cannot bind microtubules. When cells are starved, ribosomes dissociate from ER, and the repeats can then bind microtubules. **g**, PLA for p180 and RPL3 (ribosome marker) upon starvation or puromycin treatment (2 μg ml$^{-1}$ for 2 h). $n = 230$ cells. Data are mean ± s.d. with individual data points shown. One-way ANOVA followed by Dunnett's multiple comparisons test for (**b**), (**c**); two-sided $t$-test for (**d**), (**g**); $P$ values are shown. See Supplementary Information for uncropped western blots.

achieve organizational efficiency by using ER as a first-line sensor and responder. This role is exemplified during nutrient starvation, when cells increase CLIMP63 protein levels to move ER towards the perinuclear region, which also clusters lysosomes for efficient autophagic degradation. Then, cells harness enhanced p180−microtubule binding to redistribute ER and lysosomes for a proper reset. There are likely to be other ER proteins that also decipher the tubulin code, with important implications for ER function in health and disease.

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

# Methods

## Plasmids and reagents

GFP-mCherry-LC3 was a gift from Juan S. Bonifacino. mEmerald-Sec61β, pcDNA3.1_CLIMP63-HA, mApple-SiT (Golgi apparatus), mKO-SKL (peroxisome), YFP-KDEL (ER) and CFP-LAMP1 (lysosome) were constructed as described previously[6,7,18,21]. pCMV6_p180s-myc-Flag (RC218816), pCMV6_KTN1-myc-Flag (RC219832), pCMV6_TTLL4-myc-Flag (RC205206) and pCMV6_CCP1-myc-Flag (RC220826) were obtained from Origene Technologies. pcDNA3.1_TTLL6-Flag (OHu07095), pcDNA3.1_CCP5-Flag (Ohu28493), pcDNA3.1_CCP6-Flag (OHu24335) and pcDNA3.1_p180L (OHu24745) were obtained from GenScript. Mutants of CLIMP63, p180s and KTN1 were generated in a pcDNA3.1(+) vector with a C-terminal HA-epitope tag. Specifically, Ser-to-Glu mutants (S3E, S17E, S19E) of CLIMP63 were synthesized by GenScript. pcNDA3.1-KTN1, pN1-KTN1-mApple, pC1-sp-mScarlet-KTN1, pN1-KTN1-mNeonGreen, 2×Strep-p180s_29-381-mNeonGreen, 2×Strep-p180s_29-381-mNeonGreen and CLIMP63-mNeonGreen-2×Strep were also constructed using standard cloning procedures. 5′ and 3′ UTR of KTN1 were synthesized by Integrated DNA Technologies and ligated into the pN1-KTN1-mNeonGreen vector. Human TTLL4, TTLL6, TTLL7, CCP1, CCP5 and CCP6 inserts were also cloned into pCMV14-3×Flag and pN1-mApple. DNA oligonucleotides were synthesized by Integrated DNA Technologies.

LD540 was provided by C. Thiele[34]. Camptothecin (S1288), DC661 (S8808), etoposide (S1225) and MG132 (S2619) were purchased from Selleckchem. LysoSensor Green DND 189 (L7535), G418 (11811098) and puromycin (A1113802) were from Thermo Fisher Scientific. Centrinone B (CNB, CN5690) was from Tocris. GTP (G8877), ATP, Taxol (T7402), tunicamycin (SML1287), Earle's Balanced Salts (EBSS, E2888) and Duolink In Situ PLA kit (DUO92002, DUO92004, DUO92008, DUO92013) were from Sigma-Aldrich. The Cathepsin L Activity Assay Kit (Fluorometric) (ab65306) was obtained from Abcam.

## Cell culture and transfections

All cell lines were obtained from the American Type Culture Collection: HEK 293T (CRL-11268), COS7 (CRL-1651), HeLa (CCL-2), RPE1 (CRL-4000) and U2OS (HTB-96) cells. HEK 293T, COS7 and HeLa cells were cultured in Dulbecco's Modified Eagle Medium (DMEM, Thermo Fisher Scientific 11995065), RPE1 cells were cultured in DMEM/F12 (1:1) Medium (Thermo Fisher Scientific, 11330-057), and U2OS cells were cultured in McCoy's 5A medium (Thermo Fisher Scientific 16600108); all were supplemented with 10% fetal bovine serum (FBS, Thermo Fisher Scientific 2614007) and 1×penicillin/streptomycin/amphotericin B (Thermo Fisher Scientific 15240112) at 37 °C with 5% $CO_2$. HEK 293T cells were transfected with Avalanche-Everyday Transfection Reagent (EZ Biosystems, EZT-EVDY-1). U2OS and COS7 cells were electroporated using Cell Line Nucleofector Kit V (for U2OS; Lonza, VVCA-1003) and Cell Line Nucleofector Kit R (for COS7; Lonza, VVCA-1001) following the manufacturer's instructions. Amounts of key plasmids transfected are (per $1 × 10^6$ U2OS cells or $5 × 10^5$ COS7 cells): 0.3 μg CLIMP63-HA, 0.5 μg p180s-HA, 2 μg KTN1-mApple, 1 μg CLIMP63-mEmerald (for overexpression), 2 μg p180s-mEmerald, 4 μg p180L-mEmerald, 4 μg KTN1-mEmerald, as well as (in various vectors) 1 μg TTLL4, 1 μg TTLL6, 0.3 μg TTLL7, 4 μg CCP1, 0.5 μg CCP5, and 1 μg CCP6.

For RNAi knock down of AKAP450, two siRNAs targeting ATATG AACACAGCTTATGA and AACTTTGAAGTTAACTATCAA were synthesized by Eurofins Genomics. Cells were transfected using Avalanche-Omni Transfection Reagent (EZ Biosystems, EZT-OMNI-1) with 20 pmol siRNA for 3 days. For RNAi knockdown of CCP5, ON-TARGETplus siRNA sets targeting human CCP5 were purchased from Horizon Discovery (LQ-009468-00-0005), and 60 pmol siRNAs were transfected per $1 × 10^6$ U2OS cells using Lonza Cell Line Nucleofector Kit V.

## Antibodies

Primary antibodies used: mouse monoclonal anti-AKAP450 (BD Biosciences, 611518, Clone 7/AKAP450, immunoblot 1:250), rabbit polyclonal anti-Atlastin2 (Bethyl Laboratories, A303-333A, immunoblot 1:500), rabbit polyclonal anti-Atlastin3 (Proteintech, 16921-1-AP, immunoblot 1:1,000), rabbit monoclonal anti-Catalase (Cell Signaling Technology, 12980, clone D4P7B, immunofluorescence 1:800), mouse monoclonal anti-Climp63 (Enzo, ALX-804-604, clone G1/296, immunofluorescence 1:500 immunoblot 1:5,000), mouse monoclonal anti-Flag M2 (Sigma-Aldrich, F1804, clone M2, immunoblot 1:1,000), rabbit polyclonal anti-GFP (MBL, 598, immunoblot 1:5,000, immunofluorescence 1:500), mouse monoclonal anti-GM130 (BD Biosciences, 610822, Clone 35/GM130, immunofluorescence 1:200), rabbit polyclonal anti-GM130 (Proteintech, 11308-1-AP, immunofluorescence 1:200), mouse monoclonal anti-HA (Covance, MMS-101P, clone 16B12, immunofluorescence 1:500, immunoblot 1:5,000), rabbit polyclonal anti-kinectin (Proteintech, 19841, immunoblot 1:2,000), rabbit monoclonal anti-kinectin (Cell Signaling Technology, 13243, clone D5F7J, immunofluorescence 1:100), mouse monoclonal anti-Lamp1 (DSHB, clone 1D4B, immunofluorescence 1:2,000), rabbit polyclonal anti-LC3 (Cell Signaling Technology, 4108, immunofluorescence 1:200, immunoblot 1:1,000), rabbit polyclonal anti-Lunapark (Sigma-Aldrich, HPA014205, immunoblot 1:250), mouse monoclonal anti-Myc (Santa Cruz, sc-40, clone 9E10, immunoblot 1:2,000), rabbit polyclonal anti-p180 (Thermo Fisher Scientific, PA5-21392, immunofluorescence 1:500, immunoblot 1:5,000), rabbit polyclonal anti-Pericentrin (Abcam, ab4448, immunofluorescence 1:1,000), rabbit polyclonal anti-polyglutamylation (polyE) (AdipoGen, AG-25B-0030, immunofluorescence 1:200, immunoblot 1:1,000), mouse monoclonal anti-glutamylation clone GT335 (AdipoGen, AG-20B-0020, immunofluorescence 1:200, immunoblot 1:200), rabbit polyclonal anti-REEP2 (Proteintech, 15684, immunoblot 1:3,000), rabbit polyclonal anti-REEP3 (Abcam, ab106463, immunoblot 1:1,000), rabbit polyclonal anti-REEP4 (Proteintech, 26650, immunoblot 1:1,000), rabbit polyclonal anti-REEP5 (Proteintech, 14643, immunoblot 1:1,000), rabbit polyclonal anti-reticulon3 (Proteintech, 12055, immunoblot 1:2,000), rabbit polyclonal anti-reticulon4 (Proteintech, 10740, immunoblot 1:1,000), rabbit polyclonal anti-RPL3 (Proteintech, 66130, immunofluorescence 1:100), rabbit polyclonal anti-TOM20 (Santa Cruz, sc-11415, immunofluorescence 1:1,000), mouse monoclonal anti-TOM20 (BD Biosciences, 612278, Clone 29/Tom20, immunofluorescence 1:1,000), rabbit polyclonal anti-TRAPα (Proteintech, 10583, immunofluorescence 1:50), rat monoclonal anti-α-tubulin Alexa Fluor 647 (Abcam, ab195884, clone YOL1/34, immunofluorescence 1:50), mouse monoclonal anti-α-tubulin (Proteintech, 66031, clone 1E4C11, immunofluorescence 1:1,000, western blot 1:10,000), mouse monoclonal anti-β-tubulin (Proteintech, 66240, clone 1D4A4, immunofluorescence 1:1,000). Alexa Fluor 405/488/568/633 conjugated goat anti-rabbit/mouse IgG (H+L) highly cross-adsorbed secondary antibodies were from Thermo Fisher Scientific. HRP-conjugated goat anti-mouse or anti-rabbit secondary antibodies were from Santa Cruz Biotechnology.

## Stable cell lines

To generate U2OS cells stably expressing mEmerald-Sec61β, cells were transfected with the mEmerald-Sec61β[7] and selected using 200–1,000 μg μl$^{-1}$ (gradually increasing) G418 for two weeks; green-positive cells were sorted into mono-clones by flow cytometry using a MoFlo Astrios cell sorter (Beckman Coulter) and cultured in the presence of 200 μg μl$^{-1}$ G418 for 2–3 weeks. Proliferated clones were verified by immunoblotting and fluorescence imaging.

## CRISPR–Cas9 gene editing

All CRISPR–Cas9 knockout assays used eSpCas9(1.1)[35]. The targets used were: CLIMP63:GCCGCGCCCGCCATGCCCTCGG; p180 in U2OS: GGTGTCGACTTTCTCCATGAAGG; p180 in COS7: GACACCAGGAAGAT

GCCAATGG; KTN1: GAAAAGCCAGAAGAAGAGG and GTTAGGGAAAG AAAAAAGAAGG.

For knock-in of CLIMP63, the same target as in CLIMP63 knockout was used, and a PCR fragment with 37 bp homology arms on each side of the mEmerald-coding sequence was used as a homologous recombination template as follows: **CCAGCCCGCGGCCCGAGCCGCCGCC GCGCCCGCCATG**GTGAGCAAGGGCGAGGAGCTGTTCACCGGGGTGGTG *CCCATCCTGGTCGAGCTGGACGGCGACGTAAACGGCCACAAGTTCAGCG TGTCCGGCGAGGGCGAGGGCGATGCCACCTACGGCAAGCTGACCCTGAAG TTCATCTGCACCACCGGCAAGCTGCCCGTGCCCTGGCCCACCCTCGTGA CCACCTTGACCTACGGCGTGCAGTGCTTCGCCCGCTACCCCGACCACATG AAGCAGCACGACTTCTTCAAGTCCGCCATGCCCGAAGGCTACGTCCAGGA GCGCACCATCTTCTTCAAGGACGACGGCAACTACAAGACCCGCGCCGAG GTGAAGTTCGAGGGCGACACCCTGGTGAACCGCATCGAGCTGAAGGGCA TCGACTTCAAGGAGGACGGCAACATCCTGGGGCACAAGCTGGAGTACAAC TACAACAGCCACAAGGTCTATATCACCGCCGACAAGCAGAAGAACGGCATC AAGGTGAACTTCAAGACCCGCCACAACATCGAGGACGGCAGCGTGCAGC TCGCCGACCACTACCAGCAGAACACCCCCATCGGCGACGGCCCCGTGCT GCTGCCCGACAACCACTACCTGAGCACCCAGTCCAAGCTGAGCAAAGACC CCAACGAGAAGCGCGATCACATGGTCCTGCTGGAGTTCGTGACCGCCGC CGGGATCACTCTCGGCATGGACGAGCTGTACAAG*tccggactcagatctcgagc tcaagcttcgaattctgcagtcgacggtaccgcgggcccgggatcc**CCCTCGGCCAAA CAAAGGGGCTCCAAGGGCGGCACG**; (in which bold denotes homology arms; italic denotes mEmerald coding sequence; and lowercase denotes linker). To generate mEmerald-calreticulin knock-in COS7 cells, wild-type Cas9 with a gRNA targeting the end of the signal sequence of calreticulin (GAGCCCGCCGTCTACTTCAAGG) was selected, and a PCR fragment with 36 bp homology arms on each side of the mEmerald-coding sequence was used as a homologous recombination template as follows: **GGCCTCCTCGGCTTGGCCGCCG TCGAGCCCGCCGTC***ATGGTGAGCAAGGGCGAGGAGCTGTTCACCGGGGT GGTGCCCATCCTGGTCGAGCTGGACGGCGACGTAAACGGCCACAAGTTCA GCGTGTCCGGCGAGGGCGAGGGCGATGCCACCTACGGCAAGCTGACCCT GAAGTTCATCTGCACCACCGGCAAGCTGCCCGTGCCCTGGCCCACCCTCG TGACCACCTTGACCTACGGCGTGCAGTGCTTCGCCCGCTACCCCGACCACA TGAAGCAGCACGACTTCTTCAAGTCCGCCATGCCCGAAGGCTACGTCCAG GAGCGCACCATCTTCTTCAAGGACGACGGCAACTACAAGACCCGCGCCGA GGTGAAGTTCGAGGGCGACACCCTGGTGAACCGCATCGAGCTGAAGGGC ATCGACTTCAAGGAGGACGGCAACATCCTGGGGCACAAGCTGGAGTACAA CTACAACAGCCACAAGGTCTATATCACCGCCGACAAGCAGAAGAACGGCAT CAAGGTGAACTTCAAGACCCGCCACAACATCGAGGACGGCAGCGTGCAGC TCGCCGACCACTACCAGCAGAACACCCCCATCGGCGACGGCCCCGTGCTG CTGCCCGACAACCACTACCTGAGCACCCAGTCCAAGCTGAGCAAAGACCCC AACGAGAAGCGCGATCACATGGTCCTGCTGGAGTTCGTGACCGCCGCCGG GATCACTCTCGGCATGGACGAGCTGTACAAG***GAGCCCGCCGTCTACTTC AAGGAGCAGTTTCTGGAC**. Note that amino acids 18–20 (EPA) were appended to both sides, acting as a linker.

### Centrosome depletion

To deplete the centrosome, cells were treated with 125 μM CNB for 1 week as described[15] before further analysis.

### Western blotting

Cells were quickly rinsed with PBS, directly lysed with sample buffer (50 mM Tris, pH 6.8, 1 mM DTT, 10% glycerol, 2% SDS, 0.1% Bromophenol blue), and boiled for 5 min. Proteins were then resolved by SDS–PAGE using Mini-PROTEAN TGX Precast Protein Gels (Bio-Rad Laboratories) and transferred to nitrocellulose membranes using the Trans-Blot Turbo RTA Midi Nitrocellulose Transfer Kit (Bio-Rad Laboratories) following the manufacturer's instructions. Membranes were blocked with 4% milk in TBST (20 mM Tris, pH 7.4, 150 mM NaCl, 0.1% Tween-20), and incubated with primary antibody (diluted in blocking buffer) at 4 °C overnight. After washing with TBST, membranes were incubated with secondary antibody at room temperature for 2 h, followed by intensive washing with TBST. Immunoreactive proteins were visualized with GE Healthcare LS ECL Prime Western Blotting Detection Reagent (RPN2236) and imaged using a ChemiDoc XRS+ (Bio-Rad). Band intensities were quantified using Fiji software (NIH).

### Immunofluorescence and imaging

Cells were fixed with 4% paraformaldehyde in PBS (Lonza) for 30 min at room temperature and permeabilized with 0.1% Triton X-100 in PBS for 10 min. Alternatively, for immunostaining of glutamylation (GT335) and polyglutamylation (polyE), cells were fixed and permeabilized with cold methanol for 5 min at −20 °C. Then, after blocking with 3% BSA for 30 min, cells were immunostained with polyE antibody at 4 °C overnight, then with polyE and GT335 together at 4 °C overnight, followed by secondary antibody staining at room temperature for 1 h, and finally with anti-α-tubulin Alexa Fluor 647 at room temperature for 2.5 h. For staining of lipid droplets with LD540 dye, cells were incubated with 0.1 μg ml$^{-1}$ LD540 in PBS for 5 min. Cells were mounted using Fluoromount-G (SouthernBiotech) and imaged using a Zeiss LSM880 confocal microscope in Airyscan mode equipped with a 63 × 1.4 NA Plan-Apochromat oil objective (Carl Zeiss). Images were acquired using ZEN software (Carl Zeiss) and processed with ZEN software or Fiji (NIH).

### Quantification of ER distribution

Three-dimensional images were acquired using a Zeiss LSM880 confocal microscope in Airyscan mode and reconstructed using ZEN software (Zeiss Microscopy). Summed intensity projections were generated using floating point notation to carry precision. A custom macro in Fiji-ImageJ was used to define the centre of the nucleus and remove the signal of neighbouring cells to avoid perturbing the results. From the manually defined centre, a radius was drawn out past the furthest point on the cell and swept through 360° in 0.1° steps, taking a line profile each time and rescaling the data to correct for artifacts generated by the square shape of the pixels. The resulting data represents an ($r,\theta$)-space representation of the cell's fluorescence distribution. For analysis referring to 'normalized' data, we account for the shape of the cytoplasm by finding the radius at each angle where the nuclear envelope and the edge of the cell are located. The fluorescence data were then rescaled to a normalized axis with the cytoplasm between the nuclear envelope and the cell periphery scaled from 0 to 100%. The nucleoplasm is scaled to stretch between −25 and 0, as a control. (Note that, in this 2D implementation, the nucleoplasm also contains the regions of cytoplasm and nuclear envelope above and below the nucleus).

The MDR and asymmetry of each compartment were calculated using custom Matlab scripts as described in the Supplementary Text. Where true values are given by integrals over space, the value was estimated at the resolution limit of the microscope using a sum over the pixels.

### Microtubule co-sedimentation assay

To test the microtubule-binding affinities of CLIMP63, p180 and KTN1, cells were lysed in PIPES buffer (80 mM PIPES, pH 6.8, 1 mM MgCl$_2$, 1 mM EGTA, 100 mM NaCl, 1% Triton X-100, plus Complete protease inhibitors) for 30 min on ice. Cell lysates were centrifugated twice at 20,000$g$ for 20 min at 4 °C. The supernatant was supplemented with 1 mM GTP and 40 μM Taxol and incubated at 4 °C or 37 °C for 30 min for tubulin polymerization before centrifugation at 20,000$g$ for 30 min at 4 °C or 37 °C, respectively. The resulting pellets (P) and supernatants (S) were collected and subjected to immunoblot analysis. In some experiments, only the pellets and supernatants of the 37 °C samples are shown.

### Proximity ligation assay

PLA (Sigma-Aldrich, DUO92101) was performed according to the manufacturer's instructions. Samples were observed under a Zeiss LSM880 confocal microscope with a 20 × 1.0 NA objective using the Airyscan function. The total intensity of the PLA signal per cell was quantified using Fiji software.

## Protein purification

Deletion fragments of p180 (short isoform NM_001042576, residues 29-381) and KTN1 (NM_001079521, residues 29–400) as well as full-length CLIMP63 were expressed as fusions with mNeonGreen-2×Strep in HEK 293T cells. 48 h post-transfection, cells were lysed in PBS (Lonza) plus 500 mM NaCl, 1% Triton X-100, and protease inhibitors and then centrifuged at 30,000*g* at 4 °C for 30 min. Supernatants were combined with Strep-Tactin XT beads (IBA Lifesciences) and rotated gently for 3 h. After extensive washing with lysis buffer (PBS plus 500 mM NaCl and 1% Triton X-100) and then wash buffer (IBA Lifesciences), bound proteins were eluted with Strep-Tactin XT Elution Buffer (IBA Lifesciences). Eluted proteins were subjected to multiple rounds of PBS dilution and concentration using 10 kDa protein concentrators (Sigma-Aldrich), before being aliquoted and frozen in liquid nitrogen.

## TIRF-based assays for protein binding to differentially glutamylated microtubules in vitro

Unmodified human tubulin was purified from tsA201 cells as described previously[36]. TTLL4 and TTLL6 were expressed in *Escherichia coli* and purified as previously described[18]. TTLL7 was also expressed in *E. coli* and purified as previously described[19]. Taxol-stabilized microtubules were polymerized out of 98.5% unmodified tubulin and 1.5% biotinylated brain tubulin[36,29] (Cytoskeleton T333P). Unmodified microtubules were modified using TTLL4, TTLL7 or TTLL6 at 1:10 molar ratio of enzyme to tubulin at room temperature in 20 mM HEPES (pH 7.0), 50 mM NaCl, 10 mM $MgCl_2$, 1 mM glutamate, 1 mM ATP, 0.5 mM TCEP, and 10 µM Taxol for 4.5 h for TTLL4, between 20 min and 2 h for TTLL7, and between 7.5 and 22 h for TTLL6. Control microtubules were incubated with the enzymes under the same conditions but with aspartate, which is not a substrate for TTLL glutamylases, instead of glutamate. Enzymes were removed through a high-salt wash as previously described[29]. The extent of glutamylation was determined by liquid chromatography–electrospray mass spectrometry[36,29] (LC–MS). The spectra display the characteristic distributions of masses with peaks separated by 129 Da, which corresponds to one glutamate (Extended Data Fig. 6o, p). The extent of tubulin glutamylation on α- or β-tubulin was determined by calculating the weighted average of peak intensities for each tubulin species present[29].

For microtubule-binding assays, microtubules were immobilized in chambers made of silanized glass[37] using Neutravidin (Thermo Fisher Scientific). Next, a solution containing 60 mM Pipes (pH 6.8), 0.7 mM $MgCl_2$, 0.7 mM EGTA, 50 mM KCl, 10 mM 2-mercaptoethanol, 10 µM Taxol, 1% F127 Pluronic, 1.4 mg/ml casein, 20 mM glucose, glucose oxidase, and catalase was flushed into the chamber, followed by the same solution containing 4.7 nM mNeon-labeled p180, KTN1 or CLIMP63. Images were acquired after allowing for equilibration for 5 min at room temperature using total internal reflection fluorescence (TIRF) microscopy at an exposure of 100 ms for the GFP channel. Unlabelled microtubules were visualized using interference reflection microscopy[38]. Multiple fields of view were imaged. Background corrected line scan average intensities were measured using Fiji software. Multiple chambers were quantified for each condition.

## Real-time PCR

Total mRNA were extracted using TRIzol (Thermo Fisher Scientific 15596018) and Direct-zol RNA Miniprep (Zymo Research, R2052), then reverse-transcribed using the SuperScript IV First-Strand Synthesis System (Thermo Fisher Scientific, 1809105). Real-time PCR primers, designed by a free online tool developed by Integrated DNA technologies, were as follows: CCP5-RT: GACTGCCAGGAACTGCTAAA and AGGAGCTCCCGATGGTAATA; GAPDH-RT: GGTGTGAACCATGAGAAGT ATGA and GAGTCCTTCCACGATACCAAAG.

Real-time PCR was performed using Applied Biosystems PowerUp SYBR Green Master Mix (Thermo Fisher Scientific, A25780) with Applied Biosystems QuantStudio 6 Flex real-time PCR instrument. Data were collected and analysed in QuantStudio Real-time PCR Software and Microsoft Excel using the $2^{-\Delta\Delta C_T}$ method.

## Multispectral imaging

Multispectral imaging was performed as described previously[22]. Images were acquired with a Zeiss LSM880 confocal microscope equipped with a 32-channel multi-anode spectral detector (Carl Zeiss) using a 63×/1.4 NA objective lens, at 37 °C and with 5% $CO_2$. Fluorophores were excited simultaneously using 458, 514 and 594 nm lasers and a 458/514/594 nm beam splitter, with images collected onto a linear array of 32 photo-multiplier tube elements in λ mode at 9.7 nm bins from 468 to 687 nm. Spectra were defined by imaging singly labelled cells for each of the fluorophore reporters, using the same acquisition and laser settings as for multiply labeled cells. Multispectral images were unmixed using the linear unmixing package in ZEN (Carl Zeiss).

## Measurements of autophagosome–lysosome fusion and lysosome activity

For autophagosome–lysosome fusion assessments, U2OS cells were transfected with GFP-mCherry-LC3 for 24 h, treated with EBSS for 2 h before fixation with 4% paraformaldeyde in PBS, and imaged using a Zeiss LSM880 confocal microscope in Airyscan mode equipped with a 63 × 1.4 NA Plan-Apochromat oil objective (Carl Zeiss). A *z*-projection was performed using maximum projection before quantification. The mCherry-positive vesicles indicate autophagosomes already fused with lysosomes, as the GFP signal would be quenched by the acidic environment of lysosomes; vesicles with both GFP and mCherry fluorescence indicate autophagosomes not yet fused with lysosomes. Quantifications of these two types of vesicles were performed manually using Fiji software.

For lysosome acidification assays, U2OS cells were labeled with 1 µM LysoSensor Green DND 189 for 4 min and immediately imaged within one minute with a Zeiss Axio microscope using a 20×/0.4 NA objective. Images were captured with ZEN software, and total intensities of each cell were quantified in Fiji.

Cathepsin L activity assays were carried out using the Abcam Cathepsin L Activity Assay kit (Fluorometric; ab65306) following the manufacturer's instructions; $1 \times 10^6$ cells were assayed in each sample.

## Statistics and reproducibility

No statistical method was used to predetermine sample size. All groups were randomly assigned and every group represents a distinct treatment or condition. Data were not analysed in a double-blinded manner. All comparisons were performed using Graphpad Prism or Microsoft Excel software. Data are expressed as means ± s.d., *P* values are shown on top of the corresponding columns, as determined by one-way ANOVA followed by Dunnett's multiple comparisons test, Mann–Whitney test, Kruskal–Wallis test or by unpaired two-sided *t*-test as indicated in the figure legends. When representative images are shown, at least three repeats were performed except for Extended Data Figs. 1b, 3a–g, 5a, d, 6a, m, n, 8a, e, for which repeats are not necessary because they represent sequential sequence mapping data that build upon one another or else they show representative knock-down or knockout efficiencies that can be further established by the resulting cellular phenotypes.

## Reporting summary

Further information on research design is available in the Nature Research Reporting Summary linked to this paper.

## Data availability

All research materials are available upon request. Source data are provided with this paper.

## Code availability

Computer algorithms can be accessed at https://github.com/cjobara/ProbabilityDensityIntegrator.

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

**Acknowledgements** We thank J. S. Bonifacino at NICHD for providing the GFP-mCherry-LC3 vector; D. Maric (NINDS Flow Cytometry Facility) for cell sorting; C. Smith and V. Schram at NICHD Microscopy and Imaging Core for imaging assistance; V. Bolduc at NINDS for assistance with real-time PCR; E. He and A. Hoofring at NIH Medical Arts for drawing the model figures; and T. Fadero (UNC-Chapel Hill) for discussions and suggestions regarding radial density calculations. All codes are written in Matlab or ImageJ Macro language. This work was supported by the Intramural Research Programs of the National Institute of Neurological Disorders and Stroke and National Heart, Lung and Blood Institute, National Institutes of Health, as well as the Howard Hughes Medical Institute Janelia Research Campus.

**Author contributions** P.Z. and C.B. conceived and designed the experiments. P.Z. performed most of the experiments. C.J.O. developed and performed the ER morphology analysis. E.S. designed, performed, and interpreted in vitro microtubule binding assays together with P.Z. J.N.-A. performed multispectral imaging of organelles. K.K.M. provided purified TTLL proteins. C.B., A.R.-M. and J.L.-S. supervised the study. P.Z. and C.B. wrote the manuscript with contributions from all authors.

**Competing interests** The authors declare no competing interests.

**Additional information**
**Correspondence and requests for materials** should be addressed to Pengli Zheng or Craig Blackstone.

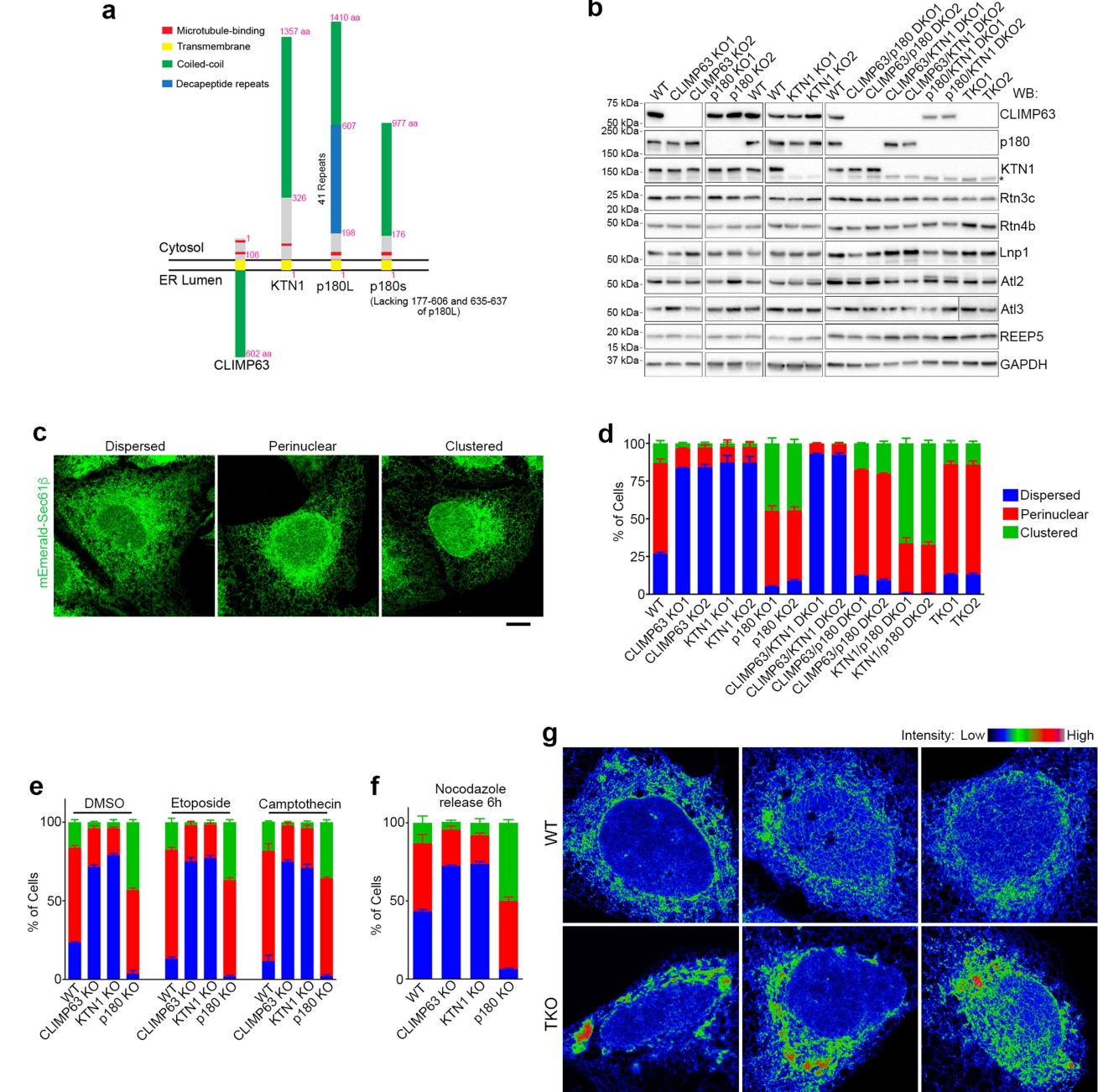

**Extended Data Fig. 1 | Knockout of CLIMP63, p180 and KTN1, and resulting ER phenotypes. a**, Schematic illustration of CLIMP63, p180 and KTN1 protein domains. Purple numbers indicate key amino acids. Shorter isoform of p180 (p180s, Uniprot Q9P2E9.5) is also shown. **b**, Western blotting (WB) of the indicated wild-type (WT) or knockout (KO) cells. The lower band in the KTN1 blots (indicated with an asterisk) corresponds to the shorter cytosolic isoform of KTN1. See Supplementary Information for uncropped western blots. **c**, Representative images of three patterns of ER distribution in U2OS cells. "Dispersed" (left) is characterized by dominant sheets or matrices at the cell periphery; "Clustered" (right) is characterized by asymmetric dense accumulation of perinuclear ER at one side of the nucleus; all other ER types are considered "Perinuclear". **d**, Proportion of wild-type or indicated KO cells with

different patterns of ER distribution. $n = 3$ experiments with at least 200 cells counted in each experiment. **e**, ER distribution of wild-type or CLIMP63, p180 or KTN1 KO cells treated with 5 μM etoposide or 100 nM camptothecin for 24 h to synchronize cells in S/G2 phase. $n = 3$ experiments with at least 200 cells counted in each experiment. **f**, ER distributions in wild-type or CLIMP63, p180 or KTN1 KO cells treated with 10 μM nocodazole for 24 h and released for 6 h to synchronize cells in G1 phase. $n = 3$ experiments with at least 200 cells counted in each experiment. **g**, Representative images of perinuclear ER in wild-type or CLIMP63, p180 and KTN1 triple-KO cells, showing LUT color grading according to intensity of the ER marker mEmerald-Sec61β. Scale bars, 10 μm. All bars represent mean ± s.d.

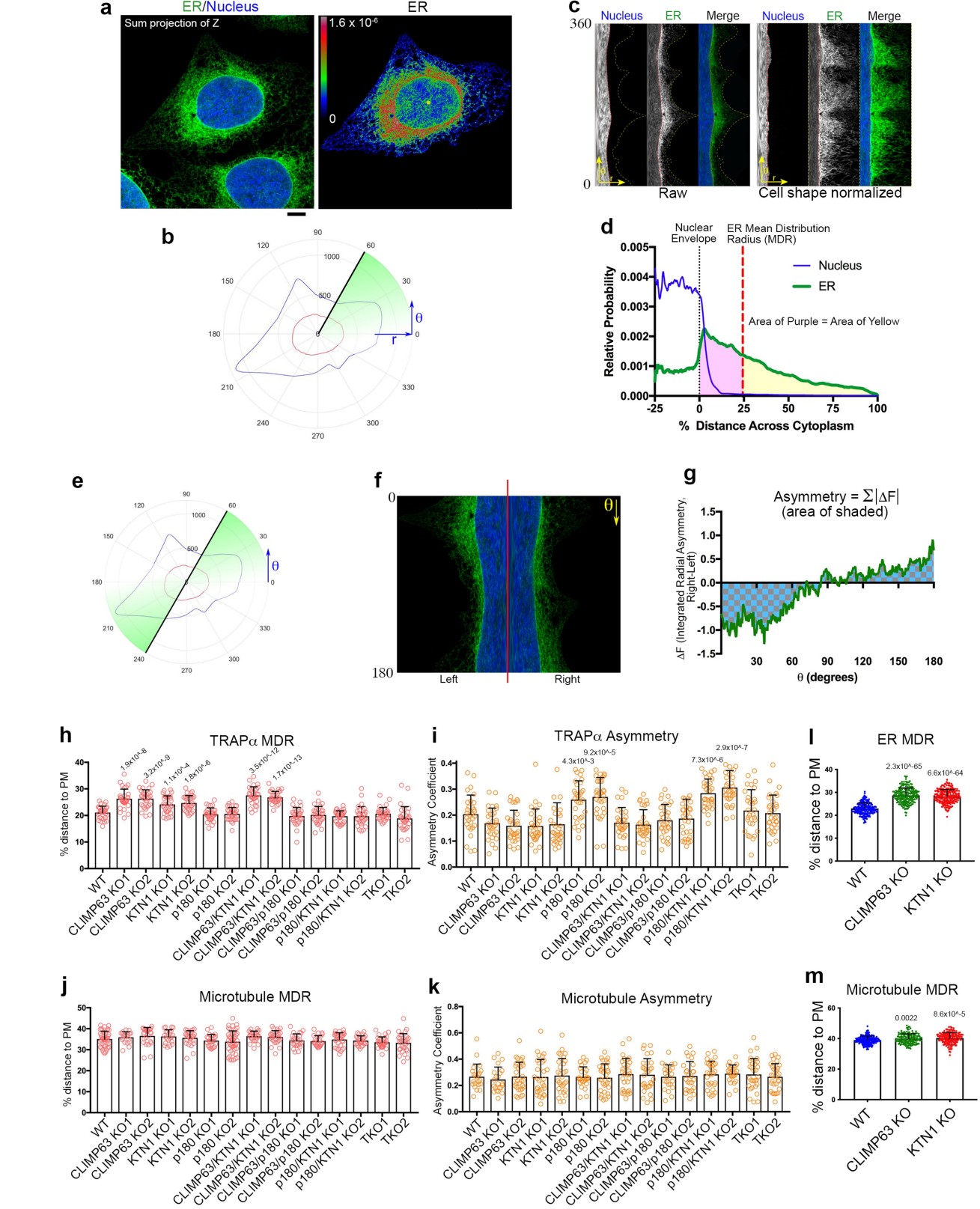

**Extended Data Fig. 2** | See next page for caption.

**Extended Data Fig. 2 | Methods for quantifying organelle distribution, and quantifications of TRAPα and microtubule distribution in knockout cells.** **a**, Summed projections generated from three-dimensional Airyscan images (left). Fluorescence from neighboring cells is removed and the center of the nucleus is manually selected to function as the origin (yellow dot in right image). Fluorescence intensities are converted to probabilities (right image, see Supplementary Text). Scale bar, 10 μm. **b**, A radius is drawn out from the center of the nucleus past the farthest point on the cell and swept through 360° in 0.1° intervals, taking a line profile each time. The nuclear envelope and edge of the cell are identified at each radius. **c**, Resulting probabilities of each channel in r- and θ-space, represented as fluorescence intensities. Red and yellow dashed lines indicate the approximate location of the nuclear envelope and the cell edge, respectively (left panel), as shown in (b). The probability distributions of nuclear and ER signals are normalized to correct for cell shape (right panel). Dashed lines indicate the location of the nuclear envelope and cell edge after normalization. **d**, Associated radial distributions of probabilities as measured in terms of distance across the cytoplasm as in (c). Relative probability (y-axis) indicates any single molecule of DAPI (nucleus) or mEmerald-Sec61β (ER) falling at a specific proportion of the distance between the nuclear envelope and the edge of the cell (x-axis). ER MDR represents the average distance of the ER on this scale and can be used to quantify the propensity of the ER to penetrate the cellular periphery; higher MDRs indicate a larger proportion of the ER in the periphery. **e**–**g**, A radius is drawn out past the farthest point on both sides of the cell and swept through 180° in 0.1° intervals, taking a line profile each time (e). The edge of the cell is identified at each radius. Resulting intensity distribution across all radii, with the red line indicating the center of the nucleus (f). For each radius, the difference between two sides of the center (ΔF) is calculated and plotted as a function of θ. The asymmetry value is then calculated as a sum of the exact values of ΔF (g). **h**, **i**, Quantifications of TRAPα (rough ER) distributions in WT or the indicated KO cells. $n = 31$ cells. **j**, **k**, Quantifications of microtubule (labeled with anti-α-tubulin) distribution for wild-type or the indicated KO cells. $n = 41, 19, 25, 21, 26, 24, 46, 25, 27, 24, 23, 30, 25, 30, 31$ cells (left to right) for j; $n = 22, 22, 30, 30, 30, 27, 31, 30, 30, 23, 26, 29, 24, 30, 31$ cells for k. **l**, **m**, Quantifications of ER and microtubule MDR for more cells to show differences in microtubule MDR. $n = 210$ cells for l, $n = 197, 161, 162$ cells for m. All bars represent mean ± s.d. $P$ values are shown on top; differences without labeling are not significant, comparisons are with the wild-type group using two-tailed $t$-tests.

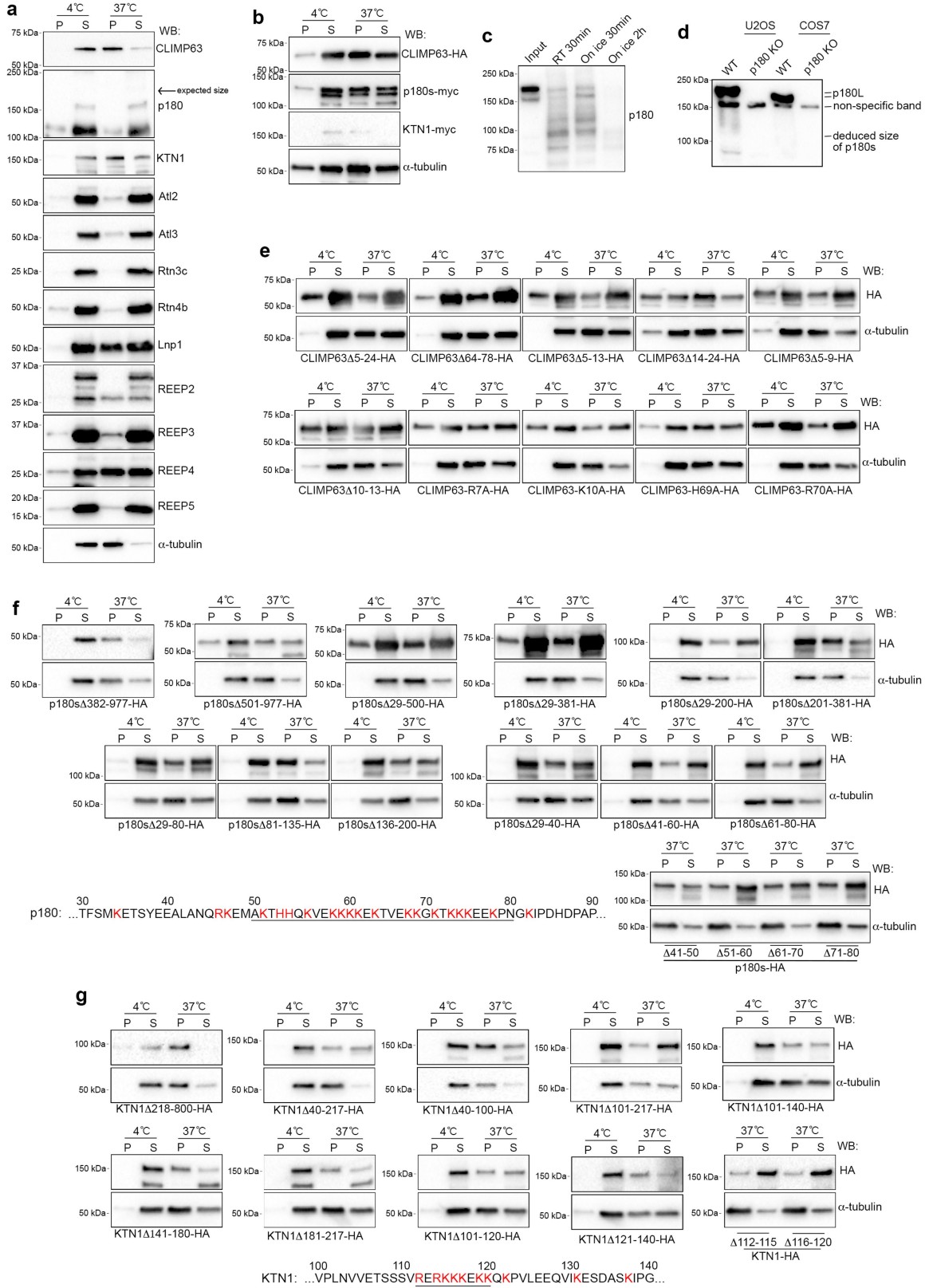

**Extended Data Fig. 3** | See next page for caption.

**Extended Data Fig. 3 | Microtubule sedimentation assays of ER proteins.**
**a**, Microtubule co-sedimentation assays of U2OS cells. The pellet (P) of 37 °C incubation indicates the microtubule-bound fraction, and supernatant (S) indicates the unbound fraction. 4 °C incubation acts as a microtubule-free control. **b**, Microtubule co-sedimentation assays of U2OS cells with exogenous CLIMP63-HA, p180s-myc or KTN1-myc expression. Proteins were expressed in corresponding knockout cells. Note that only one representative α-tubulin blot (from the CLIMP63 assay) is shown. **c**, p180 is very unstable after cell lysis. The input sample was collected by directly adding sample buffer (50 mM Tris, pH 6.8, 1 mM DTT, 10% glycerol, 2% SDS, 0.1% Bromophenol Blue) onto the plate followed by immediate boiling. Other samples were incubated in lysis buffer (50 mM Tris, pH7.4, 150 mM NaCl, 1% Triton X-100, 1 mM DTT, and protease inhibitor cocktail) at room temperature or on ice for the indicated times before adding sample buffer and boiling. **d**, Western blotting of WT or p180 knockout (KO) U2OS or COS7 cells, showing that only the long isoform is detectable in these cell lines. **e**, Detailed mapping of microtubule-binding domains of CLIMP63. **f**, Mapping of microtubule-binding domains of p180. Amino acid sequences around key microtubule-binding sites are shown at the bottom. Note that this part of the sequence is present in both long and short isoforms of p180. Positively charged amino acids are shown in red. Segments (amino acids 51-80) necessary for microtubule binding are underlined. **g**, Mapping of microtubule-binding domains of KTN1. Amino acid sequences around key microtubule-binding sites are shown at the bottom. Positively charged amino acids are in red. Segments (amino acids 112-120) necessary for microtubule binding are underlined. See Supplementary Information for uncropped western blots.

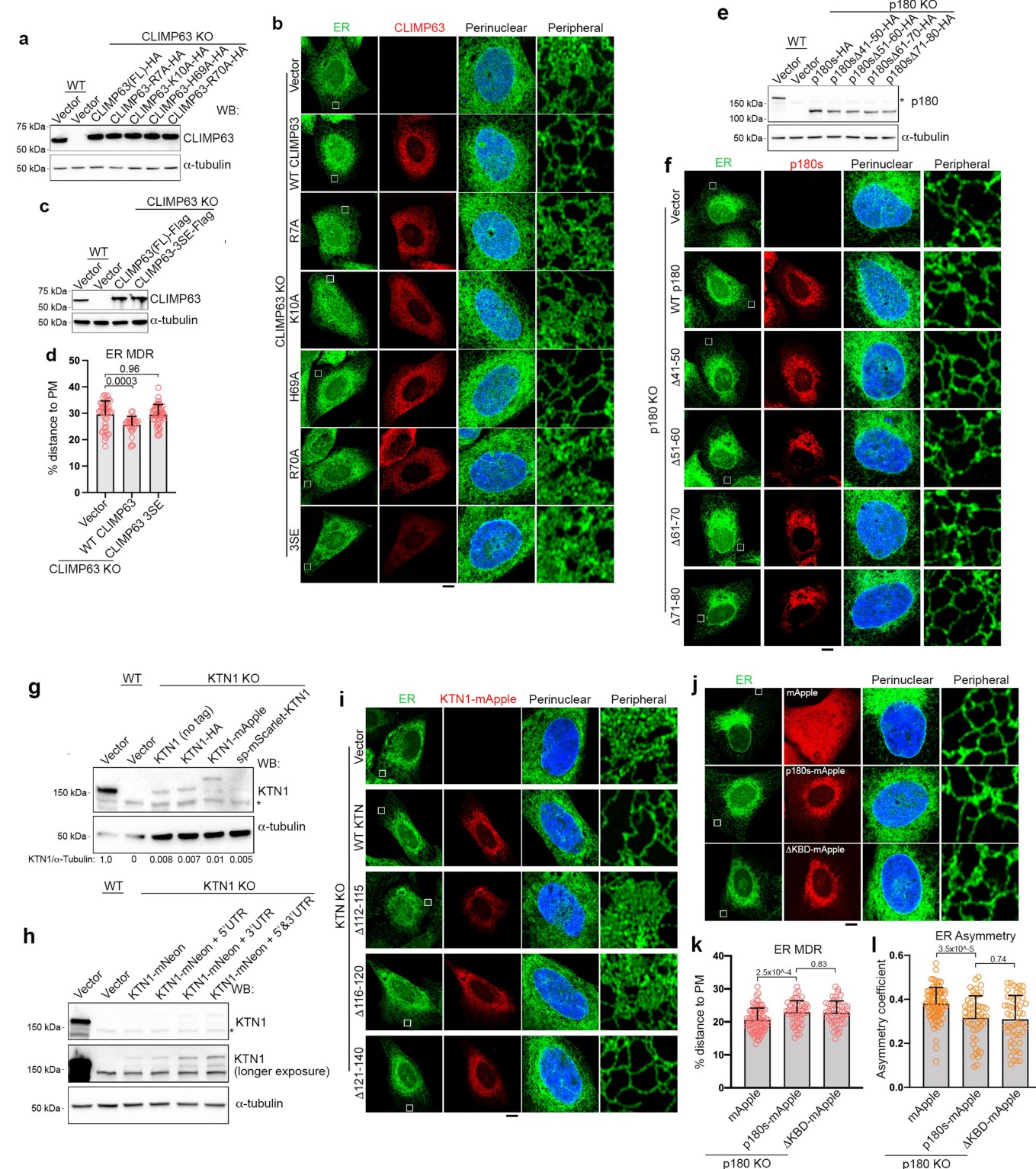

**Extended Data Fig. 4 | Rescue assays. a**, Western blots of wild-type or CLIMP63 knockout cells transfected with empty vector or the indicated CLIMP63 constructs. **b**, Representative fluorescence images and quantifications of CLIMP63 knockout cells transfected with empty vector or the indicated CLIMP63 constructs. **c**, Western blots of wild-type or CLIMP63 knockout cells transfected with empty vector or the indicated CLIMP63 constructs. CLIMP63-3SE indicates S3E, S17E and S19E triple mutation of CLIMP63. **d**, Quantifications of ER morphology in wild-type or CLIMP63 KO cells expressing the indicated CLIMP63 constructs. $n = 43, 31, 46$ cells (left to right). **e,f**, Western blots and representative images of wild-type or p180 KO cells expressing the indicated p180 constructs. **g**–**i**, Western blots and representative images of wild-type or KTN1 KO cells expressing the indicated KTN1 constructs. **j**–**l**, p180 knockout U2OS cells were transfected with mApple, p180s-mApple, or p180s mutants lacking kinesin-binding domain (KBD), then imaged and quantified for ER morphology. Scale bars, 10 μm. $n = 82, 53, 53$ cells for both (k) and (l) (left to right). Data are mean ± s.d. P values shown on top, two-tailed t-tests. See Supplementary Information for uncropped western blots.

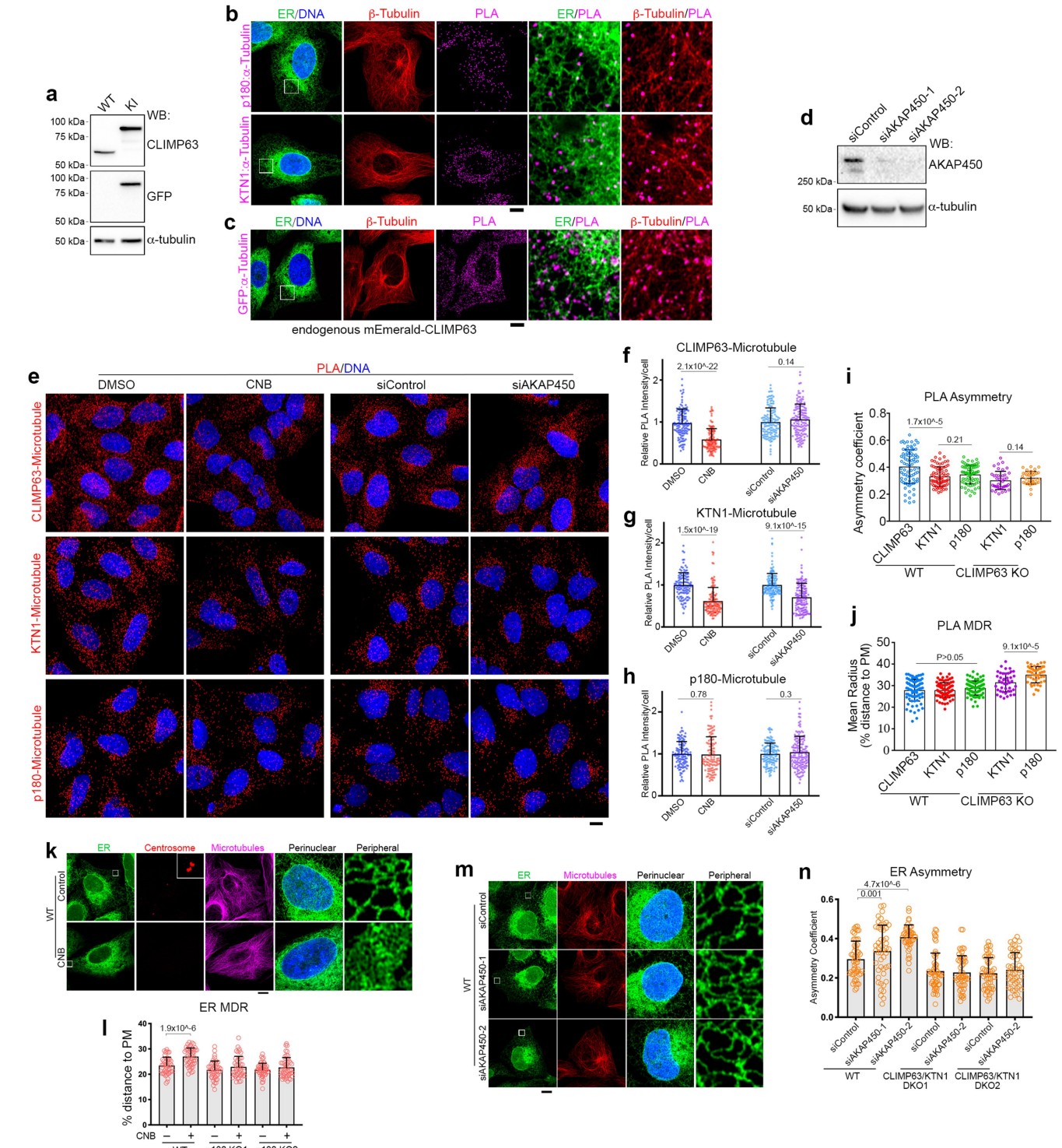

**Extended Data Fig. 5** | See next page for caption.

**Extended Data Fig. 5 | PLA of CLIMP63, p180 and KTN1 with α-tubulin, and depleting centrosome or Golgi-derived microtubules affects ER distribution. a**, N-terminal mEmerald tag was knocked into endogenous CLIMP63 using CRISPR/Cas9 to facilitate PLA, as no anti-CLIMP63 antibody targeting its cytosolic region was available. Western blots of wild-type or knock-in cells are shown. **b**, Representative PLA images of p180 or KTN1 with α-tubulin. Note that all PLA dots (a dot indicates one binding event) are localized adjacent to both ER and microtubules. **c**, Representative PLA images of endogenous mEmerald-CLIMP63 with α-tubulin. **d**, Western blot analysis of cells transfected with control or AKAP450 siRNAs. **e**–**h**, Representative data and quantifications for PLA of CLIMP63, p180, or KTN1 with α-tubulin, indicative of microtubule binding. Cells are either untreated, treated with centrinone B (CNB) to deplete the centrosome, or transfected with control siRNA or siAKAP450 (to deplete Golgi-derived microtubules). $n = 130, 134, 142, 143$ cells (left to right) for (f), $n = 116, 139, 140, 150$ cells for (g), $n = 119, 117, 139,$ 166 cells for (h). **i**, **j**, Asymmetry and MDR quantifications for PLA signals between CLIMP63, KTN1, or p180 with α-tubulin. $n = 83, 74, 72, 40, 38$ cells (left to right) for (i), $n = 81, 79, 82, 43, 44$ cells for (j). **k**, Representative images of U2OS cells with or without CNB treatment. Cells stably expressing mEmerald-Sec61β (green, ER) are stained with DAPI (blue, DNA) and immunolabeled with anti-pericentrin (red) and anti-α-tubulin (magenta) antibodies. Perinuclear and peripheral regions in the cells (boxed) are enlarged at the right. **l**, Quantifications of ER MDR in wild-type or p180 knockout (KO1 and KO2) cells, with or without CNB treatment. $n = 31, 30, 31, 43, 31, 39$ cells (left to right). **m**, **n**, Representative images and quantifications of ER asymmetry for wild-type or CLIMP63/KTN1 double-knockout cells. Cells were transfected with control siRNA or AKAP450 siRNA to deplete Golgi-derived microtubules. $n = 31, 31, 31, 32, 31, 32$ cells (left to right). Scale bars, 10 μm. Data are mean ± s.d. $P$ values shown on top, two-tailed $t$-tests. See Supplementary Information for uncropped western blots.

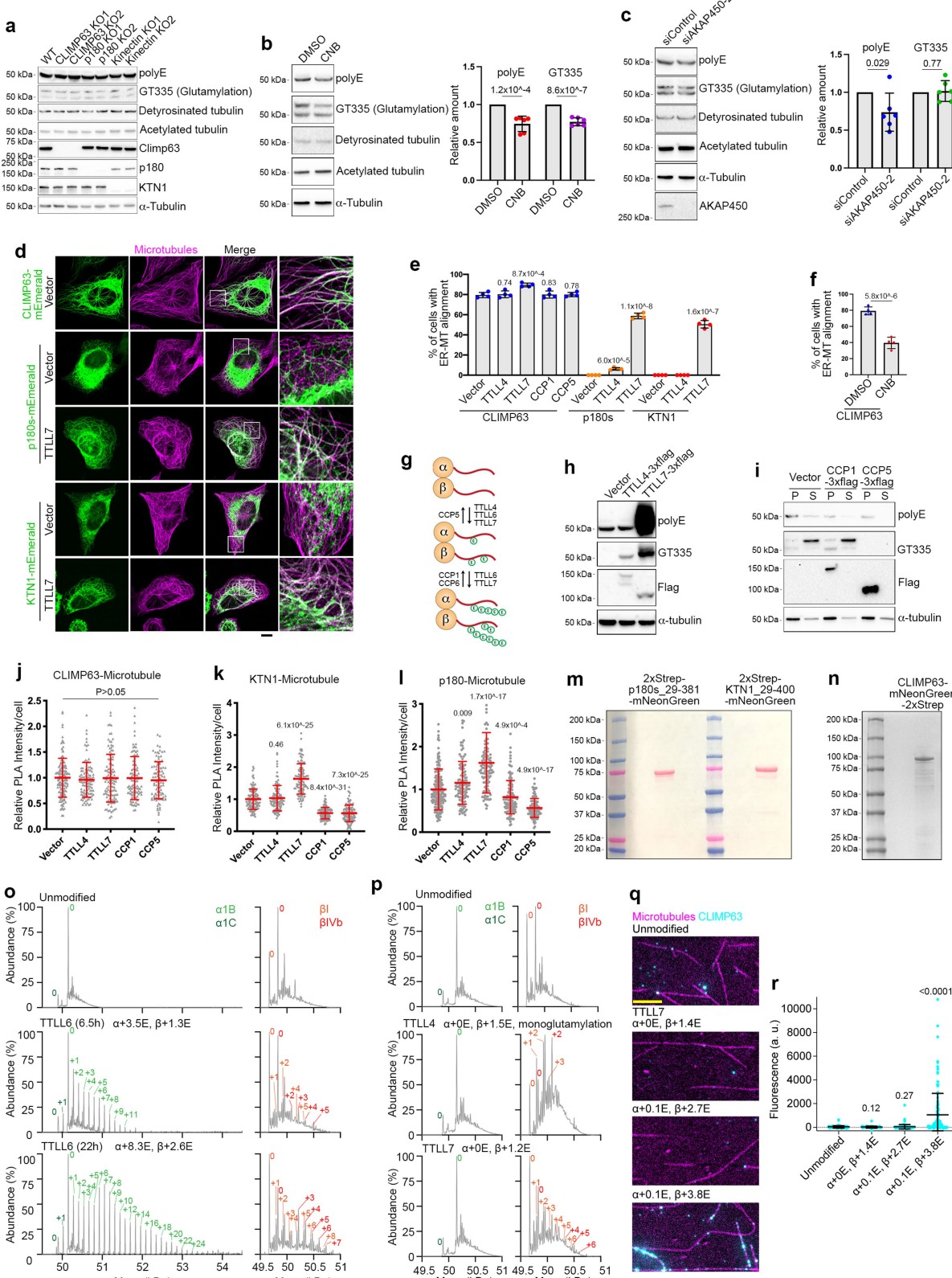

**Extended Data Fig. 6** | See next page for caption.

**Extended Data Fig. 6 | Microtubule binding of CLIMP63, p180 and KTN1 are differently affected by microtubule glutamylation. a**, Western blots of wild-type or the indicated knockout U2OS cells. **b**, **c**, Western blots and quantifications of U2OS cells either treated with CNB to deplete the centrosome or else transfected with control siRNA or siAKAP450 (to deplete Golgi-derived microtubules). $n = 6$ experiments. **d**, Representative images of U2OS cells transfected with CLIMP63-mEmerald, p180-mEmerald, or KTN1-mEmerald, with or without co-expression of TTLL7. Note that CLIMP63 overexpression leads to dramatic ER-microtubule alignment (~80% of cells), while p180 and KTN1 require co-expression of TTLL7 (which polyglutamylates microtubules) for robust ER-microtubule alignment. Scale bar, 10 μm. **e**, Quantifications of microtubule-ER alignments in cells transfected with the indicated expression plasmids; $n = 4$ experiments, with at least 100 cells counted in each experiment. **f**, Quantifications of microtubule-ER alignments in cells treated with or without CNB, and overexpressing CLIMP63-mEmerald. $n = 4$ experiments, with 100 cells counted per experiment. **g**, Schematic diagram depicting how microtubule glutamylation levels are modulated by actions of tubulin glutamylases (TTLLs) and deglutamylases (CCPs). **h**, Western blots of U2OS cells overexpressing TTLL4 or TTLL7. polyE detects microtubule polyglutamylation (at least 2 glutamates in the side chain); GT335 reacts with branch points of microtubule glutamylation and thus detects both mono- and polyglutamylation. **i**, Western blots of U2OS cells overexpressing CCP1 or CCP5. Cells were subjected to microtubule sedimentation, and pellets (P, microtubule fraction) and supernatants (S, soluble fraction) at 37 °C were analyzed. **j**–**l**, Relative PLA intensities for CLIMP63, p180, or KTN1 with

α-tubulin (indicative of microtubule binding) in U2OS cells overexpressing the indicated TTLLs or CCPs. $n = 138, 103, 112, 99, 102$ cells (left to right) for (j); $n = 112, 113, 107, 131, 121$ cells for (k); $n = 188, 108, 113, 123, 109$ cells for (l). **m**, Ponceau S staining of p180 and KTN1 proteins purified from HEK293T cells. **n**, Coomassie Brilliant Blue staining of purified CLIMP63 from HEK293T cells. **o**, **p**, Mass spectra of microtubules glutamylated by TTLL6 for different times, or by else TTLL4 or TTLL7, and used for in vitro microtubule-binding assays. Spectra display characteristic distributions of masses with peaks separated by 129 Da (corresponding to one glutamate). Peak labels show number of glutamates on α- and β-tubulin. Glutamate numbers are indicated in green, dark green, orange, and red for α1B, α1C, βI and βIVb isoforms, respectively. **q**, Representative micrographs of CLIMP63-mNeonGreen-2×Strep (CLIMP63, cyan) showing binding to unmodified microtubules or else microtubules (magenta) glutamylated in vitro by TTLL7. Interference reflection microscopy images for the microtubule channel were background subtracted and inverted. Average numbers of glutamates added to microtubules as quantified from mass spectrometry data for each group are shown on top. Scale bar, 5 μm. **r**, Affinities of CLIMP63 for unmodified microtubules and microtubules glutamylated by TTLL7. X-axis indicates the weighted average of glutamate residues attached to α- and β-tubulin. $n = 107, 61, 127$, and 136 microtubules with unmodified, α+0E/β+1.4E, α+0E/β+2.7E, and α+0.1E/β+3.8E microtubules, respectively. Data are mean ± s.d. $P$ values shown on top, Mann-Whitney test for panel r, two-tailed $t$-tests for others. See Supplementary Information for uncropped western blots.

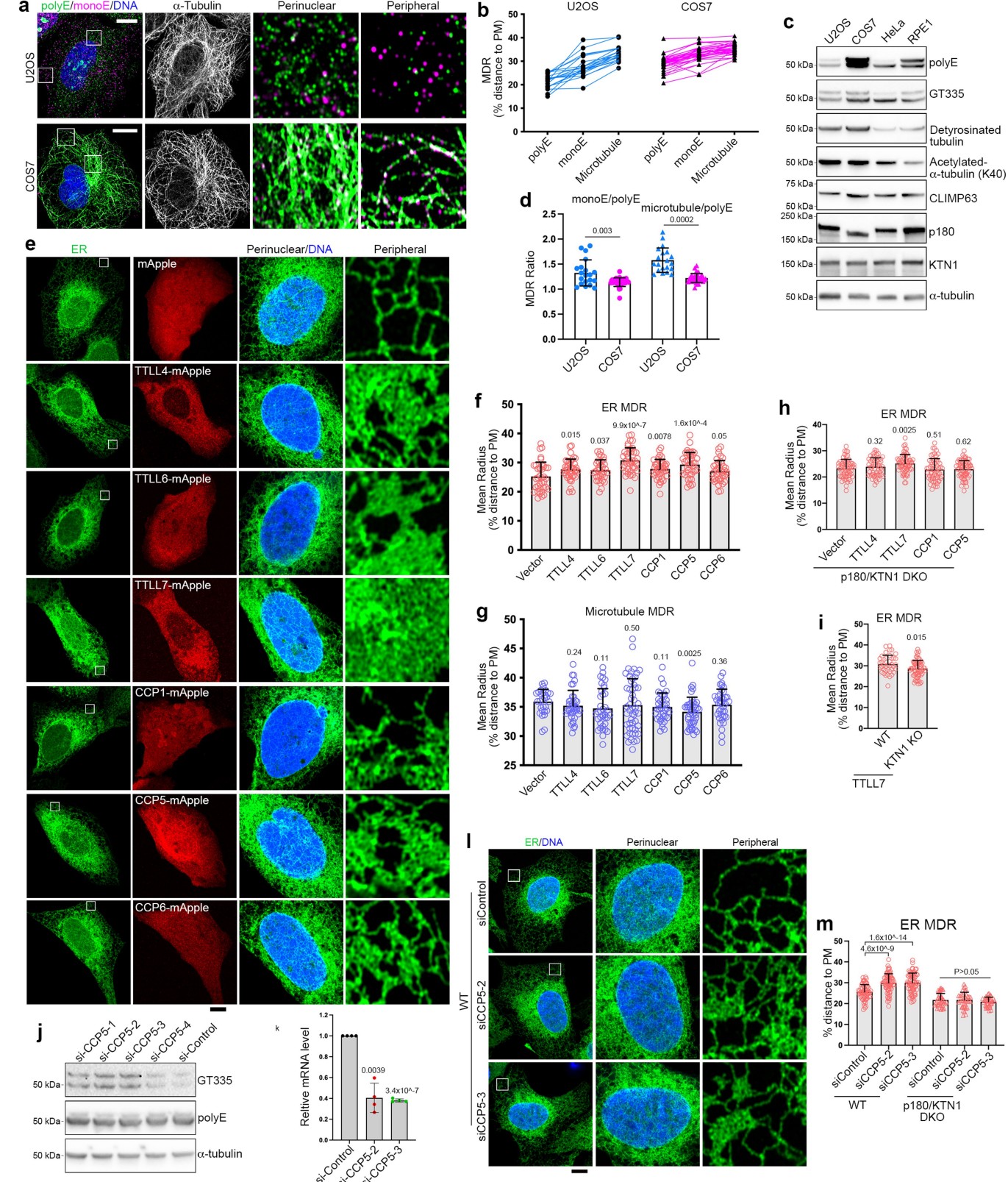

**Extended Data Fig. 7** | See next page for caption.

**Extended Data Fig. 7 | Modulating tubulin glutamylation affects ER distribution. a**, Representative images of U2OS and COS7 cells sequentially immunolabeled for polyE (polyglutamylation), polyE plus GT335 (glutamylation) and Alexa 647 conjugated α-tubulin. GT335 signal indicates monoglutamylation in this case because polyglutamylation is pre-saturated with polyE antibody. **b**, Quantifications of signal distributions for cells as in (a). Data points from the same cell are linked by solid lines. $n = 20$ cells for U2OS, 35 cells for COS7. **c**, Western blot analysis of U2OS, COS7, HeLa and RPE1 cells. **d**, MDR ratios of monoE (indicated by GT335 labeling) to polyE, or microtubules (indicated by α-tubulin labeling) to polyE. $n = 20$ cells for U2OS, 35 cells for COS7. **e**, Representative images of U2OS cells overexpressing the indicated TTLLs or CCPs. **f**, **g**, Quantifications of ER and microtubule MDRs of cells as in e; $n = 37, 40, 34, 40, 40, 41, 39$ cells (left to right) for (f); $n = 29, 38, 37, 51, 38, 46, 39$ cells for (g). **h**, Quantifications of ER MDR of cells in p180/KTN1 double knockout cells overexpressing the indicated TTLLs or CCPs; $n = 63, 61, 64, 63, 61$ cells (left to right). **i**, Quantifications of ER MDR of wild-type or KTN1 knockout cells overexpressing TTLL7; $n = 40, 64$ cells (left to right). **j**, U2OS cells transfected with the indicated siRNAs for 72 h were analyzed by western blotting. **k**, U2OS cells transfected with the indicated siRNAs for 48 h were analyzed by real time PCR. $n = 4$ repeats. **l**, **m**, Representative image and quantification of ER in control or CCP5 siRNA transfected cells. Scale bars, 10 μm. $n = 64, 70, 69, 39, 39, 39$ cells (left to right). Data are mean ± s.d. $P$ values shown on top, two-tailed $t$-test. See Supplementary Information for uncropped western blots.

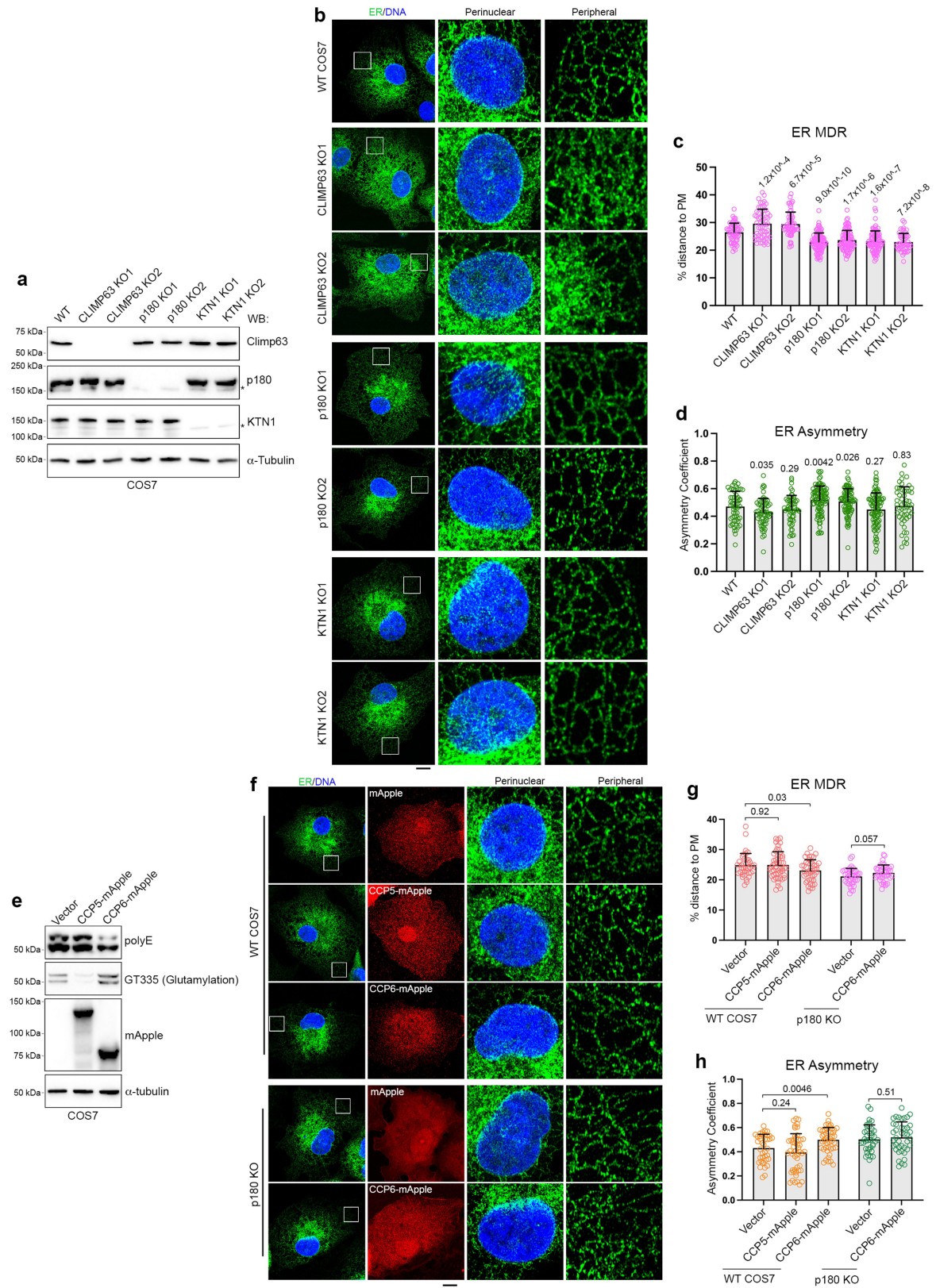

**Extended Data Fig. 8 | Knockout of CLIMP63, p180 and KTN1, or overexpression of CCPs in COS7 cells changes ER distribution. a**, Western blots of wild-type or else CLIMP63, p180, or KTN1 knockout cells. **b**, Representative images of wild-type or CLIMP63, p180, KTN1 knockout cells. ER is labeled by endogenous mEmerald-calreticulin. Scale bar, 10 μm. **c**, **d**, Quantifications of ER distribution. $n = 60, 60, 58, 96, 88, 96, 51$ cells (left to right) for both (c) and (d).

**e**–**h**, Wild-type or p180 knockout COS7 cells with endogenous mEmerald-calreticulin (ER, green) were transfected with CCP5-mApple or CCP6-mApple for 24 h and analyzed by western blotting (e) and confocal imaging (f). ER distributions are quantified (g, h). Scale bar, 10 μm. $n = 39, 49, 45, 41, 40$ cells (left to right) for both (g) and (h). Data are mean ± s.d. $P$ values shown on top, two-tailed $t$-tests. See Supplementary Information for uncropped western blots.

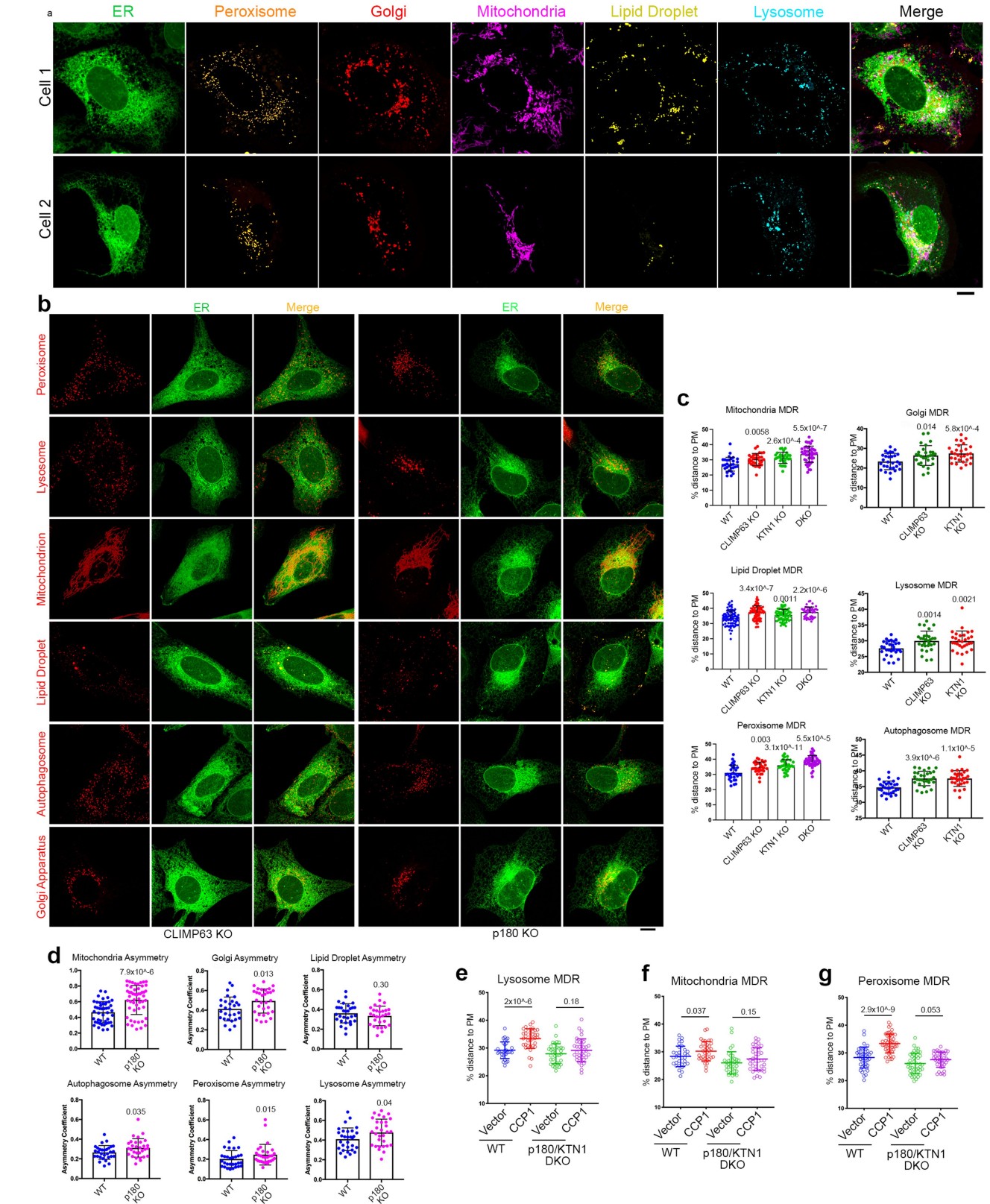

**Extended Data Fig. 9** | See next page for caption.

**Extended Data Fig. 9 | Organelle distribution in knockout cells.**
**a**, Simultaneous live imaging of six organelles with spectral unmixing.
Two representative cells are shown. Note that where there is more ER, there
tends to be more of the other organelles. **b**, Representative images of organelle
distributions in CLIMP63 or p180 knockout cells. Markers used: mEmerald-
Sec61β for ER; anti-TOM20 for mitochondria; anti-GM130 for Golgi apparatus;
anti-EEA1 for endosomes; anti-LC3 for autophagosomes; anti-catalase for
peroxisomes, anti-Lamp1 for lysosomes; and LD540 for lipid droplets. Cells
labeled for autophagosome distribution were starved in EBSS for 2 h. ER is
shown in green, while other organelles are in red. **c**, **d**, Quantifications of the
distributions of different organelles labeled with specific markers in wild-type
or else CLIMP63, KTN1, or p180 knockout cells. $n$ = 30, 31, 30, 46 cells (left to

right) for mitochondria MDR, $n$ = 83, 86, 87, 46 cells for lipid droplet MDR, $n$ = 30,
31, 31, 46 cells for peroxisome MDR, $n$ = 29 cells for Golgi MDR, $n$ = 30 cells for
Autophagosome and lysosome MDR; $n$ = 51, 46 cells for mitochondria
asymmetry, $n$ = 29, 31 cells for Golgi asymmetry, $n$ = 49, 50 cells for lipid droplet
asymmetry, $n$ = 31 cells for autophagosome, lysosome and peroxisome
asymmetry. **e**–**g**, Quantifications of lysosome, mitochondria, and peroxisome
distributions in wild-type or p180/KTN1 double knockout cells transfected
with control vector or CCP1. $n$ = 32, 36, 36, 37 cells (left to right) for (e) and (f),
$n$ = 41, 48, 44, 40 cells for (g). Scale bars, 10 μm. All bars represent mean ± s.d.
with individual data points shown. $P$ values are shown along the top, two-tailed
$t$-tests.

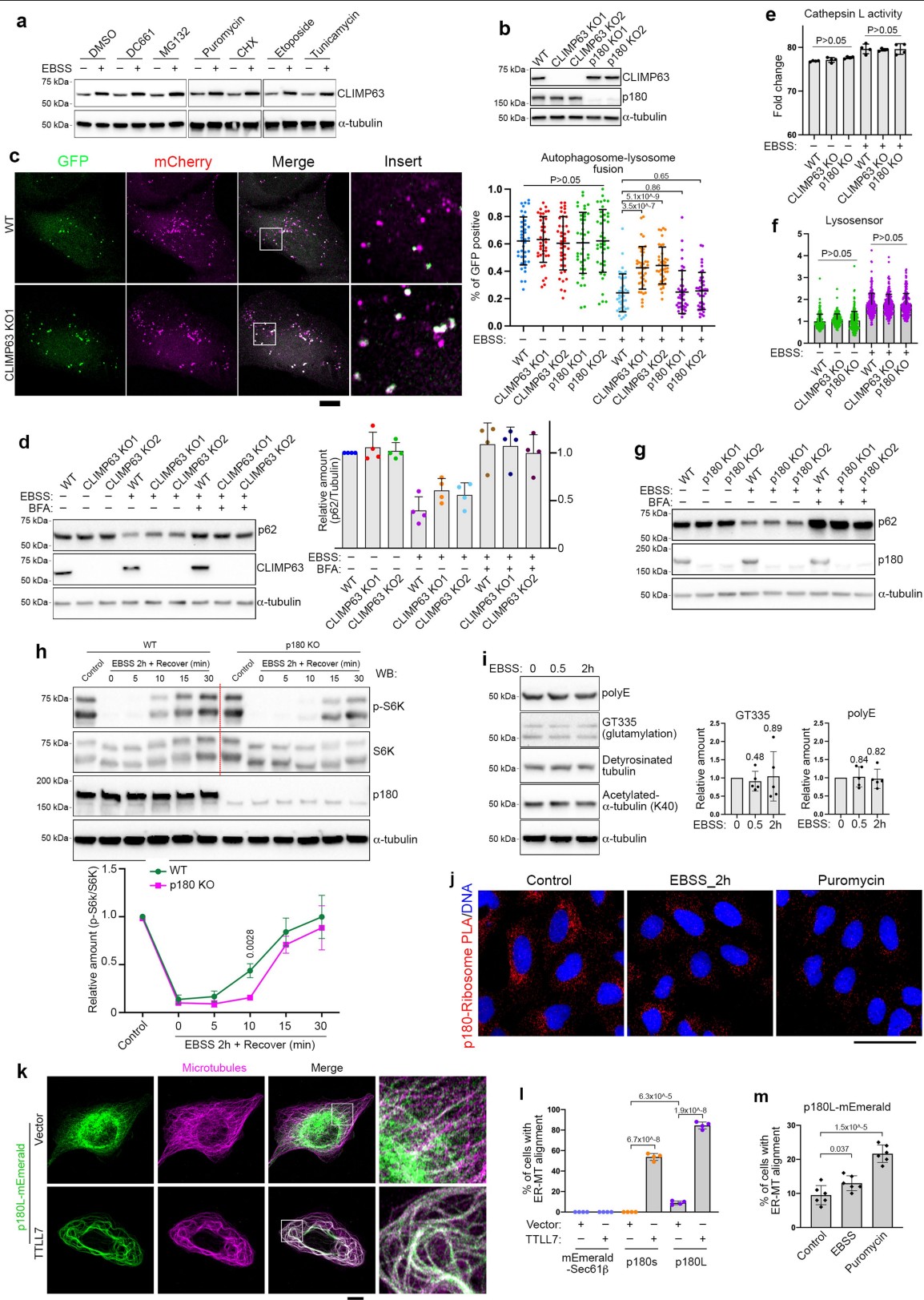

**Extended Data Fig. 10** | See next page for caption.

**Extended Data Fig. 10 | CLIMP63 and p180 regulate autophagic flux.**
**a**, U2OS cells were pre-treated with the indicated compounds for 2 h (except etoposide, which was added for 16 h), then with EBSS plus the same compound for 30 min, and then subjected to western blotting. Compounds used: 1 μM lysosome degradation inhibitor DC661, 1 μM proteosome degradation inhibitor MG132, 6 μg/mL ER translation inhibitor puromycin, 100 μg/mL protein translation inhibitor cycloheximide (CHX), 50 μM etoposide that causes DNA damage and also blocks protein expression, and 1 μm *N*-linked glycosylation inhibitor tunicamycin. **b**, Western blot analysis of wild-type, CLIMP63 or p180 knockout U2OS cells (Parental, no stable mEmerald-Sec61β expression). **c**, Representative images and quantifications for wild-type, CLIMP63 or p180 knockout U2OS cells expressing GFP-mCherry-LC3. The GFP signal is quenched by the acidic environment of lysosomes, so only autophagosomes that have not yet fused with lysosomes have green GFP signal, while autolysosomes only exhibit mCherry signal. *n* = 40 cells. Scale bar, 10 μm. **d**, Wild-type or CLIMP63 knockout cells were starved in EBSS for 0 or 8 h, with or without brefeldin A treatment to block lysosomal degradation, and then immunoblotted. Relative amounts of autophagic substrate p62 are shown in the lower panel. *n* = 4 experiments. **e**, Cathepsin L activity as determined by substrate reaction. *n* = 4 experiments. **f**, Lysosome acidification analysis using Lysosensor Green. *n* = 200 cells. **g**, Wild-type or p180 knockout cells were starved in EBSS for 0 or 8 h, with or without brefeldin A treatment to block lysosomal degradation, and western blotted. **h**, Wild-type or p180 knockout cells were starved in EBSS for 2 h, then re-supplemented with regular medium for 0-30 min. Cells were immunoblotted for phosphorylated S6K (p-S6K) and total S6K to indicate activity of mTOR signaling. *n* = 4 experiments. **i**, Western blots reveal the indicated microtubule modifications with or without EBSS starvation for 0.5 and 2 h. Relative intensities of GT335 and polyE immunoreactive signals are quantified at the right. *n* = 5 experiments. **j**, Representative images for PLA of p180 and the ribosomal marker RPL3 at 0 or 2 h of EBSS starvation, or else for cells treated with 2 μg/mL puromycin for 2 h. Scale bar, 50 μm. **k**, Representative images of U2OS cells transfected with mEmerald-Sec61β, p180s-mEmerald, or p180L-mEmerald with or without co-transfection of TTLL7-3×flag. Scale bar, 10 μm. **l**, Quantifications of microtubule alignments in cells transfected with the indicated constructs; *n* = 4 experiments, with at least 100 cells counted per experiment. **m**, Microtubule alignments of p180L-mEmerald in cells treated with EBSS or 2 μg/mL puromycin for 2 h. *n* = 6 experiments, with at least 100 cells counted per experiment. Data are mean ± s.d. *P* values are shown along the top, two-tailed *t*-tests. See Supplementary Information for uncropped western blots.

# Reporting Summary

## Statistics

For all statistical analyses, confirm that the following items are present in the figure legend, table legend, main text, or Methods section.

| n/a | Confirmed | |
|---|---|---|
| ☐ | ☒ | The exact sample size (*n*) for each experimental group/condition, given as a discrete number and unit of measurement |
| ☐ | ☒ | A statement on whether measurements were taken from distinct samples or whether the same sample was measured repeatedly |
| ☐ | ☒ | The statistical test(s) used AND whether they are one- or two-sided <br> *Only common tests should be described solely by name; describe more complex techniques in the Methods section.* |
| ☐ | ☒ | A description of all covariates tested |
| ☐ | ☒ | A description of any assumptions or corrections, such as tests of normality and adjustment for multiple comparisons |
| ☐ | ☒ | A full description of the statistical parameters including central tendency (e.g. means) or other basic estimates (e.g. regression coefficient) AND variation (e.g. standard deviation) or associated estimates of uncertainty (e.g. confidence intervals) |
| ☐ | ☒ | For null hypothesis testing, the test statistic (e.g. *F*, *t*, *r*) with confidence intervals, effect sizes, degrees of freedom and *P* value noted <br> *Give P values as exact values whenever suitable.* |
| ☒ | ☐ | For Bayesian analysis, information on the choice of priors and Markov chain Monte Carlo settings |
| ☐ | ☒ | For hierarchical and complex designs, identification of the appropriate level for tests and full reporting of outcomes |
| ☒ | ☐ | Estimates of effect sizes (e.g. Cohen's *d*, Pearson's *r*), indicating how they were calculated |

*Our web collection on statistics for biologists contains articles on many of the points above.*

## Software and code

Policy information about availability of computer code

| Data collection | No software was used to collect the data |
|---|---|
| Data analysis | Computer algorithms can be accessed at https://github.com/cjobara/ProbabilityDensityIntegrator. |

For manuscripts utilizing custom algorithms or software that are central to the research but not yet described in published literature, software must be made available to editors and reviewers. We strongly encourage code deposition in a community repository (e.g. GitHub). See the Nature Portfolio guidelines for submitting code & software for further information.

## Data

Policy information about availability of data

All manuscripts must include a data availability statement. This statement should provide the following information, where applicable:
- Accession codes, unique identifiers, or web links for publicly available datasets
- A description of any restrictions on data availability
- For clinical datasets or third party data, please ensure that the statement adheres to our policy

All data are presented in the main text or Supplementary Materials.

# Field-specific reporting

Please select the one below that is the best fit for your research. If you are not sure, read the appropriate sections before making your selection.

☒ Life sciences  ☐ Behavioural & social sciences  ☐ Ecological, evolutionary & environmental sciences

For a reference copy of the document with all sections, see nature.com/documents/nr-reporting-summary-flat.pdf

# Life sciences study design

All studies must disclose on these points even when the disclosure is negative.

| Sample size | No statistical method was used to predetermine sample size. |
|---|---|
| Data exclusions | No data were excluded |
| Replication | At least three repeats were performed unless otherwise stated |
| Randomization | All groups were randomly assigned and every group represents a distinct treatment or condition. |
| Blinding | Data were not analyzed in a double-blinded manner. The phenotypes of different groups are obvious and double-blindness can't be applied. |

# Reporting for specific materials, systems and methods

We require information from authors about some types of materials, experimental systems and methods used in many studies. Here, indicate whether each material, system or method listed is relevant to your study. If you are not sure if a list item applies to your research, read the appropriate section before selecting a response.

## Materials & experimental systems

| n/a | Involved in the study |
|---|---|
| ☐ | ☒ Antibodies |
| ☐ | ☒ Eukaryotic cell lines |
| ☒ | ☐ Palaeontology and archaeology |
| ☒ | ☐ Animals and other organisms |
| ☒ | ☐ Human research participants |
| ☒ | ☐ Clinical data |
| ☒ | ☐ Dual use research of concern |

## Methods

| n/a | Involved in the study |
|---|---|
| ☒ | ☐ ChIP-seq |
| ☒ | ☐ Flow cytometry |
| ☒ | ☐ MRI-based neuroimaging |

# Antibodies

| Antibodies used | mouse monoclonal anti-AKAP450 (BD Biosciences, 611518, Clone 7/AKAP450, immunoblot 1:250), rabbit polyclonal anti-Atlastin2 (Bethyl Laboratories, A303-333A, immunoblot 1:500), rabbit polyclonal anti-Atlastin3 (Proteintech, 16921-1-AP, immunoblot 1:1000), rabbit monoclonal anti-Catalase (Cell Signaling Technology, 12980, clone D4P7B, immunofluorescence 1:800), mouse monoclonal anti-Climp63 (Enzo, ALX-804-604, clone G1/296, immunofluorescence 1:500 immunoblot 1:5000), mouse monoclonal anti-Flag M2 (Sigma-Aldrich, F1804, clone M2, immunoblot 1:1000), rabbit polyclonal anti-GFP (MBL, 598, immunoblot 1:5000, immunofluorescence 1:500), mouse monoclonal anti-GM130 (BD Biosciences, 610822, Clone 35/GM130, immunofluorescence 1:200), rabbit polyclonal anti-GM130 (Proteintech, 11308-1-AP, immunofluorescence 1:200), mouse monoclonal anti-HA (Covance, MMS-101P, clone 16B12, immunofluorescence 1:500, immunoblot 1:5000), rabbit polyclonal anti-kinectin (Proteintech, 19841, immunoblot 1:2000), rabbit monoclonal anti-kinectin (Cell Signaling Technology, 13243, clone D5F7J, immunofluorescence 1:100), mouse monoclonal anti-Lamp1 (DSHB, clone 1D4B, immunofluorescence 1:2000), rabbit polyclonal anti-LC3 (Cell Signaling Technology, 4108, immunofluorescence 1:200, immunoblot 1:1000), rabbit polyclonal anti-Lunapark (Sigma-Aldrich, HPA014205, immunoblot 1:250), mouse monoclonal anti-Myc (Santa Cruz, sc-40, clone 9E10, immunoblot 1:2000), rabbit polyclonal anti-p180 (Thermo Fisher Scientific, PA5-21392, immunofluorescence 1:500, immunoblot 1:5000), rabbit polyclonal anti-Pericentrin (Abcam, ab4448, immunofluorescence 1:1000), rabbit polyclonal anti-polyglutamation (polyE) (AdipoGen, AG-25B-0030, immunofluorescence 1:200, immunoblot 1:1000), mouse monoclonal anti-glutamylation clone GT335 (AdipoGen, AG-20B-0020, immunofluorescence 1:200, immunoblot 1:200), rabbit polyclonal anti-REEP2 (Proteintech, 15684, immunoblot 1:3000), rabbit polyclonal anti-REEP3 (Abcam, ab106463, immunoblot 1:1000), rabbit polyclonal anti-REEP4 (Proteintech, 26650, immunoblot 1:1000), rabbit polyclonal anti-REEP5 (Proteintech, 14643, immmunoblot 1:1000), rabbit polyclonal anti-reticulon3 (Proteintech, 12055, immunoblot 1:2000), rabbit polyclonal anti-reticulon4 (Proteintech, 10740, immunoblot 1:1000), rabbit polyclonal anti-RPL3 (Proteintech, 66130, immunofluorescence 1:100), rabbit polyclonal anti-TOM20 (Santa Cruz, sc-11415, immunofluorescence 1:1000), mouse monoclonal anti-TOM20 (BD Biosciences, 612278, Clone 29/Tom20, immunofluorescence 1:1000), rabbit polyclonal anti-TRAPα (Proteintech, 10583, immunofluorescence 1:50), rat monoclonal anti-α-tubulin Alexa Fluor 647 (Abcam, ab195884, clone YOL1/34, immunofluorescence 1:50), mouse monoclonal anti-α-tubulin (Proteintech, 66031, clone 1E4C11, immunofluorescence 1:1000, WB 1:10000), mouse monoclonal anti-β-tubulin (Proteintech, 66240, clone 1D4A4, immunofluorescence 1:1000). Alexa Fluor |

405/488/568/633 conjugated goat anti-rabbit/mouse IgG (H+L) highly cross-adsorbed secondary antibodies were from Thermo Fisher Scientific. HRP-conjugated goat anti-mouse/rabbit secondary antibodies were from Santa Cruz Biotechnology.

Validation

ALL antibodies were commercial and verified by the company with the following applications: mouse monoclonal anti-AKAP450 (WB), rabbit polyclonal anti-Atlastin2 (WB, IP), rabbit polyclonal anti-Atlastin3 (WB, IP), rabbit monoclonal anti-Catalase (WB, IP, IF), mouse monoclonal anti-Climp63 (WB, IP, IF), mouse monoclonal anti-Flag M2 (WB, IP, IF), rabbit polyclonal anti-GFP (WB, IP, IF), mouse monoclonal anti-GM130 (WB, IF), rabbit polyclonal anti-GM130 (WB, IF), mouse monoclonal anti-HA (WB, IP, IF), rabbit polyclonal anti-kinectin (WB, IP, IF), rabbit monoclonal anti-kinectin (WB, IP, IF), mouse monoclonal anti-Lamp1 (IF), rabbit polyclonal anti-LC3 (WB, IF), rabbit polyclonal anti-Lunapark (WB), mouse monoclonal anti-Myc (WB, IP), rabbit polyclonal anti-p180 (WB, IF), rabbit polyclonal anti-Pericentrin (IF), rabbit polyclonal anti-polyglutamation (polyE) (WB, IF), mouse monoclonal anti-glutamylation clone GT335 (WB, IF), rabbit polyclonal anti-REEP2 (WB), rabbit polyclonal anti-REEP3 (WB), rabbit polyclonal anti-REEP4 (WB), rabbit polyclonal anti-REEP5 (WB), rabbit polyclonal anti-reticulon3 (WB), rabbit polyclonal anti-reticulon4 (WB), rabbit polyclonal anti-RPL3 (WB, IF), rabbit polyclonal anti-TOM20 (WB, IF), mouse monoclonal anti-TOM20 (WB, IF), rabbit polyclonal anti-TRAPα (WB, IF), rat monoclonal anti-α-tubulin Alexa Fluor 647 (IF), mouse monoclonal anti-α-tubulin (WB, IF), mouse monoclonal anti-β-tubulin (WB, IF).

# Eukaryotic cell lines

Policy information about cell lines

Cell line source(s)

All cell lines were obtained from the American Type Culture Collection (ATCC) including HEK293T (CRL-11268), COS7 (CRL-1651), HeLa (CCL-2), RPE1 (CRL-4000) and U2OS (HTB-96) cells

Authentication

All cell lines were authenticated by ATCC using STR profiling.

Mycoplasma contamination

All cell lines tested negative for mycoplasma contamination.

Commonly misidentified lines
(See ICLAC register)

no commonly misidentified cell lines were used in the study.

