## [Peer Review File · Nature]

Manuscript Title: ER Proteins Decipher the Tubulin Code to Regulate Organelle Distribution

Reviewer Comments & Author Rebuttals

Reviewer Reports on the Initial Version:

Referee #1 (Remarks to the Author):

The manuscript of Zheng et al. addresses the question of intracellular organization from the angle of the endoplasmic reticulum (ER). The work aims at demonstrating the important role of different microtubule sub-population in ER shaping, and link this to the regulation of organelle distribution in cells. Moreover, the authors determine how glutamylation, a posttranslational modification of tubulin, affects ER topology and organelle distribution. The work uses mostly cell biological approaches. To strengthen the findings on the role of tubulin glutamylation, the authors use complementary *in vitro* reconstitution assays.

The key findings of this study are:

- i) Spatial distribution of ER in cells is differentially controlled by three ER-membrane-bound proteins: CLIMP63, Kinectin and p180 using CRISPR/Cas9.
- ii) The authors investigate the hypothesis that these three proteins could specifically recognize distinct microtubules subpopulations in cells. They test this by depleting either centrosome- or Golgi-derived microtubules. The experiments suggest that CLIMP63 preferentially binds centrosomal microtubules, while KTN1 and p180 bind perinuclear and peripheral microtubules, respectively.
- iii) Using *in-vitro* reconstitution assays, the authors show that KTN1 and p180 bind preferentially to microtubules with increased levels of glutamylation. They further test this hypothesis in cells by overexpressing several glutamylases and deglutamylases to change the glutamylation patterns.
- iv) The authors next connect ER topology to organelle localization in cells. Knockout of CLIMP63, p180 and KTN1, which perturbs ER shape, affected the distribution of other organelles such as mitochondria, Golgi apparatus, autophagosomes, peroxisomes, and lysosomes. A similar effect can be obtained by overexpression of the deglutamylase CCP1, which in the first place causes dispersed ER, and is accompanied by perturbed distribution of other organelles. As ER morphology is controlled by microtubule glutamylation, the authors propose that glutamylation regulates the distribution of organelles via changes in ER topology.
- v) Finally, the authors provide a potential link to the physiological importance of their findings. They show that during starvation of the cells, the ER moves towards the perinuclear region of the cells, a movement that is dependent CLIMP63, but not p180.

The manuscript is well-written, the figures are of high quality, and experiments have been carefully analyzed and described. The work provides a significant step forward in the understanding of how cells organize intracellular space by using specific microtubule populations, and how the so-far little explored tubulin code participates in the generation of these microtubule subtypes. These conceptually new discoveries are very interesting to a broad cell biology community.

The key message of the work is that ER-membrane-bound proteins specifically bind to sub-populations of microtubules. The authors explore specific functions of centrosomal vs. Golgi microtubules, as well as the role of tubulin glutamylation in generating functionally distinct microtubule species. While overall the work is carefully done, it contains a number of weak points, and at times over-interpretation of the actual data, which must be addressed before the manuscript could be further considered. Moreover, the work is very data-heavy, which might be hard to follow for a non-specialist reader. The authors should consider including data representations and analyses paradigms that allow an easy appreciation of the overall outcome of some of their quantification experiments, or alternatively, show schematics explaining their data /

experimental approach.

Major points:

1) Concerns regarding experiments that address the specific localization of proteins between centrosomal vs. Golgi microtubules:

1a) The authors use proximity ligation assay (PLA) to visualize the subcellular distribution of ER-microtubule interactions mediated by either CLIMP63, KTN1, or p180 (Extended Data Fig. 8a-c). Following the conclusions drawn from Fig. 1a (knockout of each of these proteins show distinct effect on ER morphology in perinuclear vs peripheral regions), it would be expected that CLIMP63, KTN1 and p180 show preference towards perinuclear or peripheral region of the cells. For instance, knockout of CLIMP63 and KTN1 lead to dispersed ER at the periphery, suggesting a role of CLIMP63 and KTN1 in connecting ER to peripheral microtubules. However, the images shown in extended Data Fig. 8a-c do not show such preferences. The authors should quantitatively assess perinuclear vs. peripheral localization of PLA intensities from Extended Data Fig. 8a-c and discuss these results in the view of conclusions drawn from Fig. 1.

1b) Figs. 1a,b show defects in spatial distribution/morphology of the ER after knockout of CLIMP63, KTN1 and p180. Figs. 1g-i show relative PLA intensity (scoring ER-MT association events through each of these ER-membrane bound proteins) measured per cells after depletion of centrosome- or Golgi-derived microtubules. Based on these two sets of figures (Figs. 1a, b and Figs. 1g-i), the authors state "Together with the above ER phenotypes, we inferred that CLIMP63 preferentially binds centrosomal microtubules, KTN1 preferentially binds perinuclear microtubules derived from either centrosome or Golgi apparatus, and p180 preferentially binds more peripheral microtubules regardless of origin." However, PLA intensities quantified in Figs. 1g-i are normalized to the whole cell and therefore do not allow to infer the association of CLIMP63, KTN1 and p180 at perinuclear vs. peripheral regions of the cell. Thus, the authors should reanalyze their data to infer about association with perinuclear vs. peripheral region.

On a side note, the authors should also show images of PLA intensity after Centrinone B (centrosome microtubule depletion) and siAKAP450 (Golgi-derived microtubule depletion) treatment.

1c) The authors base their conclusions on the specific roles of centrosome- vs. Golgi-derived microtubules on the selective depletion of these two microtubule species. However, it has been suggested before that both microtubule species are interdependent: Centrinone-B treatment, which depletes centrosome-derived microtubules, bolsters the formation of other acentrosomal microtubules, for instance from Golgi (Wu et al. 2016 Dev Cell; Gavilan et al. 2018 EMBO rep). The authors should discuss the potential implications of these findings in the interpretation of their data shown in Fig. 1g-i.

1d) The authors show that overexpressed p180L (the longer isoform of p180) binds stronger to cellular microtubules (Extended Data Fig. 19a, b) than the shorter p180 isoform (Extended Data Fig. 10a, b). This difference is reported in manuscript "Importantly, unlike p180s that lacks these ribosome-binding repeats (Extended Data Fig. 1a), overexpression of p180L already led to some microtubule alignment (Extended Data Fig. 19a, b)". Given these isoform-related differences, the authors should discuss how the fact that the p180 antibody used in PLA assays, which appear to detect both isoforms, affects the interpretation these experiments (Fig. 1i, Extended Data Fig. 1f, Extended Data Fig. 18b and Extended Data Fig. 16c).

2) Concerns regarding experiments on the role of microtubule glutamylation in ER topology and function:

2a) The authors aim at determining the role of microtubule glutamylation in the localization of ER-

membrane-bound proteins by overexpressing these proteins together with tubulin-modifying enzymes, i.e. the glutamylases TLL4 and TLL7, followed by a localization of the fluorescent protein. In extended Fig. 10, the authors show that in contrast to over-expressed p180 and KTN1, CLIMP63 does not change its subcellular localization following forced glutamylation of the microtubules. The authors conclude from this that CLIMP63 is not sensitive to glutamylation. However, given the particularly strong microtubule localization of over-expressed CLIMP63 alone (extended Fig. 10a), it is possible that CLIMP63 binds microtubules with much higher affinity than p180 and KTN1, hence its binding is less dependent / affected by glutamylation/deglutamylation. It is a typical short-coming of overexpression experiments. Therefore, the authors must not omit *in vitro* reconstitution assays for CLIMP63 (see Fig. 2), as they could reveal a preference of this proteins for specifically glutamylated microtubules that could not be revealed in the overexpression studies.

2b) The authors show an impressive series of experiments to underpin that glutamylation generated by TLL4 and TLL7 have specific effects on the subcellular localization of ER-membrane-associated proteins, and thus, on ER shape and impact on vesicle transport. However, the imaging of the distribution of differentially glutamylated microtubule species is rather poor, and thus not entirely conclusive. It appears that U2-OS cells have very low levels of glutamylation. In the images shown in extended Fig. 13, antibody staining for glutamylation is far from convincing on microtubule: both glutamylation antibodies stain a spotty pattern throughout the cell, and are frequently not co-localizing with microtubules. What should be expected from antibodies staining tubulin glutamylation is a clear microtubule association of the staining, and even though a weak labelling could appear spotty, some correlation with the microtubule labelling. The random distribution of the currently shown labelling, which is not even excluded from the nucleus, rather suggests non-specific staining.

Given the key importance of this experiment for the model the authors propose (they write: "Microtubules in the perinuclear region show more polyglutamylation, while mono-glutamylation was generally more peripheral (Extended Data Fig. 13), consistent with KTN1 binding preferentially to perinuclear microtubules and p180 to more peripheral ones"), they must provide more conclusive experiments to prove this subcellular distribution of microtubule glutamylation. Specific techniques to preserve the cytoskeleton during fixation, or to remove the cytosol before fixation could be exploited to improve the specific staining of microtubules with glutamylation-specific antibodies.

2c) The authors use a set of glutamylating and deglutamylating enzymes to test the specificity of tubulin glutamylation patterns on ER topology. This is an appropriate approach, as it also reflects what is expected to happen in the cells: specific enzymes might selectively modify specific sub-species of microtubules, thus generating characteristic patterns.

From the interpretation of *in vitro* assays, KTN1 shows a preferential binding to TLL7- vs. TLL4-modified microtubules (extended data Fig. 10b). The authors interpret this result such as KTN1 showing a preference for polyglutamylated microtubules, considering that TLL7 generates polyglutamylation in contrast to TLL4, which exclusively generates monoglutamylation. However, in the PLA experiments (Extended Data Fig. 11e), KTN1-microtubule interactions are reduced after overexpression of two different deglutamylases, CCP1 and CCP5. Given that CCP5 is known to specifically remove monoglutamylation from tubulin, this result does not entirely match the interpretation of the *in vitro* experiments: removing monoglutamylation should have no or only a little effect on KTN1 localization if KTN1 preferentially binds to polyglutamylated microtubules. There are several ways to interpret these results, which is needed to be done in the manuscript. One is related to the specificities of TLL4 and TLL7, see next point.

3) Concerns on interpretation of enzymatic specificities of glutamylases:

The authors assume that TLL4 is a monoglutamylase, while TLL7 is a polyglutamylase. This is based on previous work from the authors showing that TLL7 can incorporate substantial numbers of glutamates into tubulin. However, it is not excluded that TLL7 generates a large number of

shorter glutamates chains (including mono-glutamate chains), of which not all are necessarily elongated to generate polyglutamylation. This model would provide a wider spectrum of potential interpretations of the observed phenotypes, including the possibility that TTL7 generates a mixture of glutamylation types on tubulin (mono- AND poly-glutamylation). Interpreting the data in the light of this possibility would also avoid the binary monoglutamylation vs. polyglutamylation logic the authors have put forward in the current version of the manuscript. From all the current literature plus the experiments shown in this manuscript, it is safe to say that TTL4 does not generate polyglutamylation.

4) Concerns with the proposed model:

The authors should provide a unifying model that brings together all of their major findings. In the current form of the manuscript, the authors do not discuss how the results obtained from experiments addressing the roles of centrosomal- vs. Golgi-derived microtubules integrate with experiments testing the role(s) of microtubule glutamylation. This becomes particularly critical in the interpretation of the starvation experiments, which induced changes in the spatial distribution of ER, while no differences in the glutamylation levels of microtubules could be found. This suggests there might be more than one mechanism to spatially regulate ER topography in the cells. The authors should discuss all these points in context and provide a unifying model for their results.

Minor Points:

- 1) There is a recurrent typo (distance) at the x-axes of several graphs in extended data figures.
- 2) The authors should explain why they particularly chose to test microtubule glutamylation as tubulin modification to study in the context of ER topology. Did they do some preliminary tests with other tubulin modifications?
- 3) Concerning the quantification method used in Fig. 2, authors need to explain how the fluorescence intensities were normalized to the length of the microtubules.

Referee #2 (Remarks to the Author):

The interactions between microtubule and organelles play a key role in organelle shaping and positioning. In this manuscript, the authors systematically analyzed interactions between ER sheet-shaping proteins, including Climp63, p180 and KTN1, and microtubule. They were able to identify certain specificity in microtubule recognition by these proteins that may have physiological relevance. They then proposed that the interacting network is important for positioning of other organelles and demonstrated as an example that Climp63 and p180-mediated microtubule interactions play a role in autophagy, likely by regulating localization of lysosomes. In general, the data are convincing and the findings are interesting and may be useful to the field. Some concerns need to be addressed before the publication of the work:

- 1) The work is based entirely on the usage of U2OS derived knockout cells line. Although it is appreciated that the authors systematically generate these cell lines with CRISPR/Cas9 system, it raises concern whether similar mechanism would be applicable in other cell types. It would be very helpful if the phenotype were to be verified in at least one more cellular system. It would be even more interesting if different ER morphologies are scored in cells expressing different endogenous levels of these proteins.
- 2) The MDR and asymmetry scoring system is very powerful in analyzing subtle differences in the ER. However, in some cases, scoring type was not picked appropriately. In Figure 4, the MDR was not scored, but data in Figure 1 suggest that MDR, and not asymmetry, is more sensitive to levels

of Climp63. Similarly, in Extended Data Fig. 5d, asymmetry needs to be analyzed.

3) In second section of the results, the authors started by narrowing down the microtubule-binding regions in these proteins. These mutants could be very important as they are expected to compromise just the microtubule interactions, but not other functions of the molecules. This point corresponds to a large amount of efforts, but unfortunately was very briefly touched in the text. It is recommended that representative mutants mentioned in the text. It has been shown that phosphorylation of Climp63 disrupts association with microtubule, the phosphor-mimetic mutant could be tested as in Extended Data Fig. 5b. It would also be plausible to put the representative mutants to test in experiments like Figure 2 as a negative control.

4) The claim that "despite distinct phenotypes, these ER morphology alterations in CLIMP63, p180 and KTN1 knockout cells are all microtubule-binding-dependent." is overstated and is recommended to be tuned down.

5) Figure 2 is not easy to follow. How are the numbers of Glu modification were determined in these cases should be briefly introduced, at least in the legends. It is also problematic that the purified proteins themselves frequently form bright puncta even without microtubule association, see point 3 for possible solution.

6) It is worrisome, and at the same time expected, that the autophagic defects seen in Climp63 deletion are rather minor. In general, many ER-resident proteins, in addition to the three ones focused here, interact with microtubule and may contribute jointly to the overall homeostasis of the ER morphology and functions. For example, ER sheet-localized Sec61b has been shown to interact with microtubule. It is recommended that the limitation on physiological relevance of the current study to be mentioned in discussion.

7) Along the same line, it has been shown that p180, but not Climp63 or KTN1, plays a specific role in distribute the ER into axons, which is consistent with the findings here. It would be useful to discuss and compare these findings. In addition, microtubule-interacting proteins may in turn affect microtubule dynamic. In the case here, it is mentioned that microtubule MDR or asymmetry was largely unchanged, but is the specific modification altered when corresponding proteins were deleted or overexpressed?

Referee #3 (Remarks to the Author):

Zheng and colleagues sought to investigate the role of three ER membrane-bound proteins that bind microtubules: CLIMP63, p180 and KTN1. The study focuses on their implication in shaping the ER through their ability to bind different subpopulations of microtubules defined according to their origin and glutamylation status. CLIMP63 preferentially binds centrosomal microtubules, KTN1 binds preferentially perinuclear polyglutamylated microtubules from either centrosome or Golgi origin, and p180 binds more peripheral microtubules regardless of their origin and which are glutamylated with either short or long chains. The authors used microscopy-based algorithms to measure the changes in ER distribution and defined two relevant parameters as ER mean radius and ER asymmetry.

ER distribution impacts organelles positioning. The authors showed that the knock-out of CLIMP63, p180 and KTN1 affects the positioning of several organelles in a similar way to their effect on ER distribution.

The study is interesting and brings some novelty and knowledge on ER shaping. However, the title is misleading as the tubulin code refers to various posttranslational modifications modulating the properties and functions of the microtubule, while the authors focus exclusively on glutamylation. Moreover, tubulin glutamylation modulation is exclusively investigated by overexpressing enzymes, which should be complemented with knock-out or silencing experiments. Finally, the

authors focus on the regulation of autophagy through ER-dependent lysosome clustering. The authors should check the fusion between autophagosomes and lysosomes, as well as the lysosomal activity.

Specific comments:

1. Effect of siAKAP450 and centrinoneB on MT polymerization should be checked by MT sedimentation assays to make sure that the effects of CLIMP63, KTN1 and p180 on MT binding is not due to changes in tubulin polymerization.
2. Most studies on Tubulin glutamylation in the manuscript (Ext Fig10,11, 3d- f) uses overexpression of tubulin glutamylases and deglutamylases (TTLL4, TTLL7, CCP1, CCP5). These are known to affect MT stability and dynamics. Therefore, it is difficult to say whether the ER distribution changes noted in these cells are due to changes in MT dynamics/stability or whether ER proteins differentially bind MTs depending on their glutamylation state. siRNA mediated KD of tubulin glutamylases will be more appropriate in these studies. ER-MT alignment studies may also be due to changes in MT polymerization.
3. In vitro assay in Figure 2 clearly shows that glutamylated MTs bind p180 and KTN1. Graphs in 2d, and 2e should be merged as authors comment on comparison between protein binding to mono and polyglutamylated MTs. Also these results should be supported with fractionation assays to show the proportion of polymerized MTs available for binding to p180 and KTN1.
4. In figure 3d-f why was mean radius measured instead of asymmetry in p180/KTN1 double KOs as p180 was established to cause ER asymmetry changes in Figure 1.
5. Figure 4 has some interesting observations on how ER and lysosome asymmetry is affected by starvation induced autophagy. However, these observations seem premature and are not directly linked to the rest of the manuscript since they are independent of tubulin PTMs as shown in Ext fig 17. There are also several open questions like why protein levels of CLIMP63 change on starvation, and how this controls p62 levels in fig 4d.
6. Ribosome binding of p180 decreases on Puromycin treatment. This leads to an increase in MT binding as shown in PLA assays. These findings need more proof since the MT binding domain and ribosome binding decapeptide repeats are in different parts of the protein sequence as mentioned in extended figure 18.

Additional comments:

1. Descriptive statistics are missing in the figure legends.
2. Ext. Fig 9c: are the representative images from wildtype or CLIMP63/KTN1 double-knockout cells?
3. Ext. Fig 10b: is the n number referring to the number of cells or number of fields? Significances are missing.
4. Fig 3b: CLIMP63 KO still seem to affect the distribution of the lipid droplets; why would that be? How is the double KO CLIMP63/KTN1 affecting the MDR of the different organelles?
5. Fig 3: is tubulin glutamylation itself affecting the localisation of the organelles?
6. Fig 4: is the lysosome asymmetry related to tubulin glutamylation? Why is CLIMP63 protein level affected by starvation?
7. Ext. Fig 19: Why would the ribosome-binding repeats of p180L be able to bind microtubule when there is a domain already dedicated to this interaction? Further investigations are required there. What makes the binding exclusive?

Author Rebuttals to Initial Comments:

Referee #1 (Remarks to the Author):

The manuscript of Zheng et al. addresses the question of intracellular organization from the angle of the endoplasmic reticulum (ER). The work aims at demonstrating the important role of different microtubule sub-population in ER shaping, and link this to the regulation of organelle distribution in

cells. Moreover, the authors determine how glutamylation, a posttranslational modification of tubulin, affects ER topology and organelle distribution. The work uses mostly cell biological approaches. To strengthen the findings on the role of tubulin glutamylation, the authors use complementary in vitro reconstitution assays.

The key findings of this study are:

- i) Spatial distribution of ER in cells is differentially controlled by three ER-membrane-bound proteins: CLIMP63, Kinectin and p180 using CRISPR/Cas9.
- ii) The authors investigate the hypothesis that these three proteins could specifically recognize distinct microtubules subpopulations in cells. They test this by depleting either centrosome- or Golgi-derived microtubules. The experiments suggest that CLIMP63 preferentially binds centrosomal microtubules, while KTN1 and p180 bind perinuclear and peripheral microtubules, respectively.
- iii) Using in-vitro reconstitution assays, the authors show that KTN1 and p180 bind preferentially to microtubules with increased levels of glutamylation. They further test this hypothesis in cells by overexpressing several glutamylases and deglutamylases to change the glutamylation patterns.
- iv) The authors next connect ER topology to organelle localization in cells. Knockout of CLIMP63, p180 and KTN1, which perturbs ER shape, affected the distribution of other organelles such as mitochondria, Golgi apparatus, autophagosomes, peroxisomes, and lysosomes. A similar effect can be obtained by overexpression of the deglutamylase CCP1, which in the first place causes dispersed ER, and is accompanied by perturbed distribution of other organelles. As ER morphology is controlled by microtubule glutamylation, the authors propose that glutamylation regulates the distribution of organelles via changes in ER topology.
- v) Finally, the authors provide a potential link to the physiological importance of their findings. They show that during starvation of the cells, the ER moves towards the perinuclear region of the cells, a movement that is dependent CLIMP63, but not p180.

The manuscript is well-written, the figures are of high quality, and experiments have been carefully analyzed and described. The work provides a significant step forward in the understanding of how cells organize intracellular space by using specific microtubule populations, and how the so-far little explored tubulin code participates in the generation of these microtubule subtypes. These conceptually new discoveries are very interesting to a broad cell biology community.

The key message of the work is that ER-membrane-bound proteins specifically bind to subpopulations of microtubules. The authors explore specific functions of centrosomal vs. Golgi microtubules, as well as the role of tubulin glutamylation in generating functionally distinct microtubule species. While overall the work is carefully done, it contains a number of weak points, and at times over-interpretation of the actual data, which must be addressed before the manuscript could be further considered. Moreover, the work is very data-heavy, which might be hard to follow

for a non-specialist reader. The authors should consider including data representations and analyses paradigms that allow an easy appreciation of the overall outcome of some of their quantification experiments, or alternatively, show schematics explaining their data / experimental approach.

Response: We greatly appreciate the reviewer's positive comments on the impact and quality of our work. We have answered each of the reviewer's concerns as follows.

Major points:

1) Concerns regarding experiments that address the specific localization of proteins between centrosomal vs. Golgi microtubules:

1a) The authors use proximity ligation assay (PLA) to visualize the subcellular distribution of ER-microtubule interactions mediated by either CLIMP63, KTN1, or p180 (Extended Data Fig. 8a-c). Following the conclusions drawn from Fig. 1a (knockout of each of these proteins show distinct effect on ER morphology in perinuclear vs peripheral regions), it would be expected that CLIMP63, KTN1 and p180 show preference towards perinuclear or peripheral region of the cells. For instance, knockout of CLIMP63 and KTN1 lead to dispersed ER at the periphery, suggesting a role of CLIMP63 and KTN1 in connecting ER to peripheral microtubules. However, the images shown in extended Data Fig. 8a-c do not show such preferences. The authors should quantitatively assess perinuclear vs. peripheral localization of PLA intensities from Extended Data Fig. 8a-c and discuss these results in the view of conclusions drawn from Fig. 1.

Response: We concur with the reviewer that distributions of the PLA signals between microtubules (MT) and these three ER proteins represent important information. Therefore, we have quantified the distribution of the PLA signal for each protein (Extended Data Fig. 5f, g). The results show that, in wild-type cells, PLA signals for CLIMP63-MT distribute more asymmetrically compared with p180 and KTN1, yet their MDRs are similar. We reasoned that this is because these three proteins are each enriched in the perinuclear ER, which is very dense (see staining below). Therefore, in wild-type cells, these PLA signals would always concentrate in the perinuclear region. Even though p180-MT and KTN1-MT could have moderate differences, it is impossible to see as the signals are so packed.

Immunostaining of endogenous Climp63, p180 and KTN1 in U2OS cells

To address this issue, we re-quantified p180-MT and KTN1-MT distributions in CLIMP63 knockout cells, where the ER is much more dispersed (Fig. 1). Indeed, in this situation, the MDR of p180-MT is larger than that of KTN1-MT (Extended Data Fig. 5g).

1b) Figs. 1a,b show defects in spatial distribution/morphology of the ER after knockout of CLIMP63, KTN1 and p180. Figs. 1g-i show relative PLA intensity (scoring ER-MT association events through each of these ER-membrane bound proteins) measured per cells after depletion of centrosome- or Golgi-derived microtubules. Based on these two sets of figures (Figs. 1a, b and Figs. 1g-i), the authors state “Together with the above ER phenotypes, we inferred that CLIMP63 preferentially binds centrosomal microtubules, KTN1 preferentially binds perinuclear microtubules derived from either centrosome or Golgi apparatus, and p180 preferentially binds more peripheral microtubules regardless of origin.” However, PLA intensities quantified in Figs. 1g-i are normalized to the whole cell and therefore do not allow to infer the association of CLIMP63, KTN1 and p180 at perinuclear vs. peripheral regions of the cell. Thus, the authors should reanalyze their data to infer about association with perinuclear vs. peripheral region.

On a side note, the authors should also show images of PLA intensity after Centrinone B (centrosome microtubule depletion) and siAKAP450 (Golgi-derive microtubule depletion) treatment.

Response: As mentioned above, we quantified the PLA distribution (Extended Data Fig. 5f, g), and these data further support our proposed model. We also provided the representative images of PLA data as the reviewer suggested above (Extended Data Fig. 5e).

1c) The authors base their conclusions on the specific roles of centrosome- vs. Golgi-derived microtubules on the selective depletion of these two microtubule species. However, it has been suggested before that both microtubule species are interdependent: Centrinone-B treatment, which depletes centrosome-derived microtubules, bolsters the formation of other acentrosomal microtubules, for instance from Golgi (Wu et al. 2016 Dev Cell; Gavilan et al. 2018 EMBO rep). The authors should discuss the potential implications of these findings in the interpretation of their data shown in Fig. 1g-i.

Response: We thank the reviewer for this suggestion. We agree that there was a gap in our results for centrosome-vs Golgi-derived microtubules and glutamylation. Therefore, in the revised manuscript, we examined whether depleting certain microtubule originations affects microtubule glutamylation. Notably, both centrosome depletion and knock down of AKAP450 reduced total polyglutamylation levels, although not to a large extent (Extended Data Fig. 6b, c). This would explain why centrosome depletion and knock down of AKAP450 decrease KTN1-microtubule binding in PLA (Fig 1h), and because this is a minor reduction, this may explain why we did not see a difference in p180-microtubule binding in PLA studies (Fig 1i).

1d) The authors show that overexpressed p180L (the longer isoform of p180) binds stronger to cellular microtubules (Extended Data Fig. 19a, b) than the shorter p180 isoform (Extended Data Fig. 10a, b). This difference is reported in manuscript “Importantly, unlike p180s that lacks these ribosome-binding repeats (Extended Data Fig. 1a), overexpression of p180L already led to some microtubule alignment (Extended Data Fig. 19a, b)”. Given these isoform-related differences, the authors should discuss how the fact that the p180 antibody used in PLA assays, which appear to detect both isoforms, affects the interpretation these experiments (Fig. 1i, Extended Data Fig. 1f, Extended Data Fig. 18b and Extended Data Fig. 16c).

Response: The short isoform is not expressed at detectable levels in both U2OS and COS7 cells (Extended Data Fig. 3d). Therefore, for any imaging of endogenous p180, the signal represents the long isoform only.

2) Concerns regarding experiments on the role of microtubule glutamylation in ER topology and function:

2a) The authors aim at determining the role of microtubule glutamylation in the localization of ER-membrane-bound proteins by overexpressing these proteins together with tubulin-modifying enzymes, i.e. the glutamylases TTLL4 and TTLL7, followed by a localization of the fluorescent protein. In extended Fig. 10, the authors show that in contrast to over-expressed p180 and KTN1, CLIMP63 does not change its subcellular localization following forced glutamylation of the microtubules. The authors conclude from this that CLIMP63 is not sensitive to glutamylation. However, given the particularly strong microtubule localization of over-expressed CLIMP63 alone (extended Fig. 10a), it is possible that CLIMP63 binds microtubules with much higher affinity than p180 and KTN1, hence its binding is less dependent / affected by glutamylation/deglutamylation. It is a typical short-coming of overexpression experiments. Therefore, the authors must not omit in vitro reconstitution assays for CLIMP63

(see Fig. 2), as they could reveal a preference of this proteins for specifically glutamylated microtubules that could not be revealed in the overexpression studies.

Response: We appreciate the reviewer's suggestion. In the revised manuscript, we purified CLIMP63 and performed in vitro microtubule binding assays using TTLL7-modified microtubules. Unlike p180 and KTN1, which show robust binding to microtubules with 1.2E (Fig 2c, d), CLIMP63 did not show binding to microtubules with 1.4E or 2.7E (Extended Data Fig. 6q, r). Only when the $\langle n^E \rangle$ reached 3.8 did CLIMP63 start to show binding (Extended Data Fig. 6q, r). Therefore, hyperglutamylation can indeed enhance CLIMP63-microtubule binding in vitro, but combined with the results that overexpression of CCP1 or CCP5 did not seem to affect CLIMP63's microtubule binding (Extended Data Fig. 6e, j), and centrosome depletion indeed suppressed CLIMP63 overexpression-mediated ER-MT alignments (Extended Data Fig 6f), we believe that another tubulin modification or interaction is likely responsible for CLIMP63's preferential binding toward centrosome microtubules.

2b) The authors show an impressive series of experiments to underpin that glutamylation generated by TTLL4 and TTLL7 have specific effects on the subcellular localization of ER-membrane-associated proteins, and thus, on ER shape and impact on vesicle transport. However, the imaging of the distribution of differentially glutamylated microtubule species is rather poor, and thus not entirely conclusive. It appears that U2-OS cells have very low levels of glutamylation. In the images shown in

extended Fig. 13, antibody staining for glutamylation is far from convincing on microtubule: both glutamylation antibodies stain a spotty pattern throughout the cell, and are frequently not co-localizing with microtubules. What should be expected from antibodies staining tubulin glutamylation is a clear microtubule association of the staining, and even though a weak labelling could appear spotty, some correlation with the microtubule labeling. The random distribution of the currently shown labeling, which is not even excluded from the nucleus, rather suggests non-specific staining. Given the key importance of this experiment for the model the authors propose (they write: “Microtubules in the perinuclear region show more polyglutamylation, while mono-glutamylation was generally more peripheral (Extended Data Fig. 13), consistent with KTN1 binding preferentially to perinuclear microtubules and p180 to more peripheral ones”), they must provide more conclusive experiments to prove this subcellular distribution of microtubule glutamylation. Specific techniques to preserve the cytoskeleton during fixation, or to remove the cytosol before fixation could be exploited to improve the specific staining of microtubules with glutamylation-specific antibodies.

Response: We thank the reviewer for these suggestions. The amount of glutamylation in U2OS cells is indeed very low, making the staining appear spotty rather than a more linear microtubular pattern, but we believe that these “spots” are indeed mostly on microtubules (extended Fig. 13 in previous manuscript). Regarding the nuclear signal, that is expected, as many nuclear proteins are also reported to be glutamylated via the same TTLLs (van Dijk, JBC, 2008, 283:3915-22). Also, TTLLs and CCPs, at least when overexpressed, have some localization within the nucleus (Extended Data Fig. 7e). However, 1) we only quantified the cytoplasmic signal in the MDR quantification; 2) the majority of glutamylation signal still comes from microtubules (van Dijk, JBC, 2008, 283:3915-22); and 3) we also labeled cells after CCP1 or CCP5 overexpression, which showed that the polyglutamylation signal was indeed dramatically removed:

Therefore, we believe that this represents a good estimate of glutamylation distribution in the cytoplasm. Also, since the perinuclear microtubules are less motile and longer-lived, they would be expected to have higher glutamylation, as we observed.

Nevertheless, we did try two previously reported protocols for permeabilizing cells before fixation, one from the Lippincott-Schwartz lab at Janelia Research Campus (Lorenz, Nature Protocols, 2006, 276–279) using HKM buffer and digitonin, as well as one from the Mitchison lab at Harvard Medical School using microtubule stabilization buffer and Triton X-100 (https://mitchison.hms.harvard.edu/files/mitchisonlab/files/fluorescence_procedures_for_the_actin_and_tubulin_cytoskeleton_in_fixed_cells.pdf). As shown below, the first method, with low digitonin concentrations, cannot completely remove the cytoplasmic background; when used with high digitonin concentrations, the cytoplasmic GT335 signal indeed all localized to microtubules (note that the nuclear staining is more prominent in this one, likely because nuclear proteins can't be washed out with this method), but the microtubules also become very fragmented at this concentration. Similar microtubule fragmentation was observed using the second method. Therefore, in our hands, we think that the methanol fixation is still the best way to label glutamylation while largely preserving microtubule structure.

Moreover, during this revision process, we also repeated some key experiments using a different cell line, COS7 (Extended Data Fig. 8), whose microtubules are highly polyglutamylated (Extended Data Fig. 7a-d). Using the same labeling and quantification method as for U2OS cells, we were able to clearly see distribution differences between U2OS and COS7 cells (Extended Data Fig. 7b, d), further supporting that the labeling and quantification method we used can be used to reliably quantify glutamylation distribution in cells.

2c) The authors use a set of glutamylating and deglutamylating enzymes to test the specificity of tubulin glutamylation patterns on ER topology. This is an appropriate approach, as it also reflects what is expected to happen in the cells: specific enzymes might selectively modify specific sub-species of microtubules, thus generating characteristic patterns.

From the interpretation of in vitro assays, KTN1 shows a preferential binding to TTLL7- vs. TTLL4-modified microtubules (extended data Fig. 10b). The authors interpret this result such as KTN1 showing a preference for polyglutamylated microtubules, considering that TTLL7 generates polyglutamylation in contrast to TTLL4, which exclusively generates monoglutamylation. However, in the PLA experiments (Extended Data Fig. 11e), KTN1-microtubule interactions are reduced after overexpression of two different deglutamylases, CCP1 and CCP5. Given that CCP5 is known to specifically remove monoglutamylation from tubulin, this result does not entirely match the interpretation of the in vitro experiments: removing monoglutamylation should have no or only a little effect on KTN1 localization if KTN1 preferentially binds to polyglutamylated microtubules. There are several ways to interpret these results, which is needed to be done in the manuscript. One is related to the specificities of TTLL4 and TTLL7, see next point.

Response: Yes, CCP5 is supposed to only remove the branch-point glutamate. However, in our overexpression experiments, CCP5 overexpression not only decreased GT335 (glutamylation) signal, but also led to a dramatic decrease in polyE signal (Extended Data Fig. 6i, also see staining in response to reviewer's point 2b). This is likely because U2OS cells contain some endogenous CCPs that could cooperate with the overexpressed CCP5 to reduce polyglutamylation. During this revision, we also performed experiments using COS7 cells. COS7 cells have very high polyglutamylation, and unlike U2OS cells, CCP5 overexpression in COS7 cells did not affect the polyglutamylation signal (Extended Data Fig. 8e). This indicates that COS7 cells may have low activity of endogenous CCP compared with U2OS cells, which may explain why COS7 cells have much higher polyglutamylation.

3) Concerns on interpretation of enzymatic specificities of glutamylases:

The authors assume that TTLL4 is a monoglutamylase, while TTLL7 is a polyglutamylase. This is based on previous work from the authors showing that TTLL7 can incorporate substantial numbers of glutamates into tubulin. However, it is not excluded that TTLL7 generates a large number of shorter glutamates chains (including mono-glutamate chains), of which not all are necessarily elongated to generate polyglutamylation. This model would provide a wider spectrum of potential interpretations of the observed phenotypes, including the possibility that TTLL7 generates a mixture of glutamylation types on tubulin (mono- AND poly-glutamylation). Interpreting the data in the light of this possibility would also avoid the binary monoglutamylation vs. polyglutamylation logic the authors have put forward in the current version of the manuscript. From all the current literature plus the experiments shown in this manuscript, it is safe to say that TTLL4 does not generate polyglutamylation.

Response: TTLL7 can indeed both initiate and elongate glutamylation (Mukai, Biochemistry 2009, 48:1084–1093; Garnham, PNAS 2017, 114: 6545-6550) and the reviewer is correct that it generates tails with a mixture of chains of different glutamate lengths (the chains are initiated primarily from two sites on the tail as shown in Garnham, PNAS 2017, 114: 6545-6550). We have also shown that TTLL4 adds only monoglutamates using MS/MS in our recent paper (Mahalingan, Nature Struct Mol Biol, 2020, 27:802–813). On the other hand, MS/MS experiments showed that TTLL6 is an elongase that adds primarily a long polyglutamate chain (Mahalingan, Nature Struct Mol Biol, 2020, 27:802–813). And yes, the reviewer brings up a good point about the binary model. We show that KTN1 binds stronger to microtubules that are polyglutamylated than monoglutamylated (Fig. 2d, e) even though they have the same mean glutamate numbers, suggesting a preference for the longer glutamate chain for this protein and a weaker binding affinity for monoglutamylation. However, we cannot exclude that the increased affinity for the TTLL7 modified microtubules comes from the additional chains that this enzyme initiates on the tubulin tails and not only from the introduction of longer chains. On the other hand, p180 is more sensitive to any increase in glutamate numbers on the tubulin tail, and thus shows robust binding to both monoglutamylated and polyglutamylated microtubules.

4) Concerns with the proposed model:

The authors should provide a unifying model that brings together all of their major findings. In the

current form of the manuscript, the authors do not discuss how the results obtained from experiments addressing the roles of centrosomal- vs. Golgi-derived microtubules integrate with experiments testing the role(s) of microtubule glutamylation. This becomes particularly critical in the interpretation of the starvation experiments, which induced changes in the spatial distribution of ER, while no differences in the glutamylation levels of microtubules could be found. This suggest there might be more than one mechanism to spatially regulate ER topography in the cells. The authors should discuss all these points in context and provide a unifying model for their results.

Response: We appreciate the reviewer's suggestion. We have added model figures to summarize our findings. (Extended Data Figs. 9a; 10i, j).

We also tested tubulin glutamylation levels in centrosome- or Golgi-microtubule depleted cells. Interestingly, both centrosome depletion and Golgi-microtubule depletion led to decreases in polyglutamylation (Extended Data Fig. 6b, c), and this likely explains why KTN1-microtubule binding is decreased in these two scenarios (Fig. 1h). Meanwhile, p180-microtubule binding not affected (Fig. 1i) possibly because the decreases are minor (Extended Data Fig. 6b, c).

For the ER movement during autophagy, we have provided mechanistic insight. Although the microtubule modifications are not changed, the protein level of CLIMP63 is dramatically increased (Fig 4c), which would be predicted to concentrate the ER toward the centrosomal region, and explains the ER clustering at 30 min post-EBSS starvation. For the reverse movement of ER back to the periphery, we show that p180L gains extra microtubule-binding ability as autophagy proceeds, due to the positively-charged nature of the ribosome-binding domain that is now exposed (Fig. 4e, g; Extended Data Fig. 10i-n). We drew a model to illustrate this (Extended Data Fig. 10i, j). Nevertheless, we completely agree that there are other mechanisms regulating ER distribution.

Minor Points:

1) There is a recurrent typo (distrance) at the x-axes of several graphs in extended data figures.

Response: We thank the reviewer for noticing this. We have corrected them.

2) The authors should explain why they particularly chose to test microtubule glutamylation as tubulin

modification to study in the context of ER topology. Did they do some preliminary tests with other tubulin modifications?

Response: We chose glutamylation because in previous reports, such as (Bobinnec, *Cell Motility and the Cytoskeleton*, 1998, 39:223–232), the staining of microtubule glutamylation looked to be enriched in the perinuclear region. Nevertheless, in the revised manuscript, we also found that tubulin polyglutamylation level are decreased upon centrosome or Golgi microtubule depletion (Extended Data Fig. 6b, c), providing another reason (albeit retrospective) for focusing on glutamylation.

3) Concerning the quantification method used in Fig. 2, authors need to explain how the fluorescence intensities were normalized to the length of the microtubules.

Response: In the quantification of Fig. 2, we just drew a width 7 line along each microtubule, regardless of its length, and then obtained the average intensity along that microtubule. Because it is average intensity, the length is already normalized.

Referee #2 (Remarks to the Author):

The interactions between microtubule and organelles play a key role in organelle shaping and positioning. In this manuscript, the authors systematically analyzed interactions between ER sheet-shaping proteins, including Climp63, p180 and KTN1, and microtubule. They were able to identify certain specificity in microtubule recognition by these proteins that may have physiological relevance. They then proposed that the interacting network is important for positioning of other organelles and demonstrated as an example that Climp63 and p180-mediated microtubule interactions play a role in autophagy, likely by regulating localization of lysosomes. In general, the data are convincing and the findings are interesting and may be useful to the field. Some concerns need to be addressed before the publication of the work:

Response: We thank the reviewer for appreciating the importance of our study, and for their very helpful comments and suggestions, which we address below.

1) The work is based entirely on the usage of U2OS derived knockout cells line. Although it is appreciated that the authors systematically generate these cell lines with CRISPR/Cas9 system, it raises concern whether similar mechanism would be applicable in other cell types. It would be very helpful if the phenotype were to be verified in at least one more cellular system. It would be even more interesting if different ER morphologies are scored in cells expressing different endogenous levels of these proteins.

Response: We agree with the reviewer that repeating key experiments in another cell line would greatly improve the study and strengthen our conclusions. Therefore, in the revised manuscript, we present experiments in COS7 cells, which fortuitously turn out to have very high endogenous tubulin polyglutamylation (Extended Data Fig. 7a-d). We first made a complete set of knockout lines in COS7 cells and looked at the effects on ER morphology (Extended Data Fig. 8a-d). The CLIMP63 and p180 knockout cells showed similar phenotypes as U2OS cells. Interestingly, KTN1 knockout in COS7 cells leads to clustered ER in contrast to more dispersed ER in U2OS cells. This is because COS7 microtubules are highly polyglutamylated, even in the very peripheral regions (Extended Data Fig. 7a-d), thus KTN1 in COS7 cells will bind both peripheral and perinuclear microtubules with similar affinity, rather than preferentially binding perinuclear microtubules as in U2OS cells, mimicking the TLL7 overexpressing condition in U2OS cells (Extended Data Fig. 7i).

We further tested the effects of overexpressing CCPs in COS7 cells. For unclear reasons, we were unable to get CCP1 to express well in COS7 cells (possibly due to its large size?), so we chose CCP6, which is smaller yet also shortens the polyglutamate chain just like CCP1. In U2OS cells, overexpression of either CCP1, CCP5 or CCP6 would decrease the binding of p180 and KTN1 toward perinuclear glutamylated microtubules, thus leading to dispersed ER (Extended Data Fig. 7e, f). In contrast, in COS7 cells, p180 and KTN1 normally bind both perinuclear and peripheral microtubules due to their high polyglutamylation; thus, overexpression of CCP6 leads to clustered ER (Extended Data Fig. 8e-h). It is worth mentioning that overexpression of CCP5 in U2OS cells decreases both polyglutamylation and glutamylation (Extended Data Fig. 6i), while in COS7 cells CCP5 only affected glutamylation but not polyglutamylation (Extended Data Fig. 8e). CCP5 is reported to be only able to remove the branch point, i.e., mono-glutamylation, so the result in COS7 cells is consistent with this CCP5 enzymatic specificity. The results in U2OS cells likely indicate that U2OS cells have stronger endogenous activity of CCPs, which cooperate with overexpressed CCP5 to reduce polyglutamylation.

2) The MDR and asymmetry scoring system is very powerful in analyzing subtle differences in the ER. However, in some cases, scoring type was not picked appropriately. In Figure 4, the MDR was not scored, but data in Figure 1 suggest that MDR, and not asymmetry, is more sensitive to levels of Climp63. Similarly, in Extended Data Fig. 5d, asymmetry needs to be analyzed.

Response: In Figure 4B, the major point is that ER and lysosome become clustered upon starvation, which needs the Asymmetry quantification to demonstrate. If we perform MDR quantification, the data would be like this:

As the reviewer can see, the CLIMP63 KO MDR is of course higher than WT, and we did not see any differences upon starvation, but this can't support the conclusion for our main point as the ER MDR in WT cells also barely changed upon starvation.

In the previous Extended Data Fig. 5d, which is now Extended Data Fig. 4f, we did used Asymmetry. See Fig 1e in both previous and revised manuscript.

3) In second section of the results, the authors started by narrowing down the microtubule-binding regions in these proteins. These mutants could be very important as they are expected to compromise just the microtubule interactions, but not other functions of the molecules. This point corresponds to a large amount of efforts, but unfortunately was very briefly touched in the text. It is recommended that representative mutants mentioned in the text. It has been shown that phosphorylation of Climp63 disrupts association with microtubule, the phosphor-mimetic mutant could be tested as in Extended Data Fig. 5b. It would also be plausible to put the representative mutants to test in experiments like Figure 2 as a negative control.

Response: We thank the reviewer's kind suggestions. We added some description of these mutants in the text. We also performed rescue assays using the phosphor-mimetic mutant as the reviewer suggested which, as expected, did not rescue the dispersed ER in CLIMP63 knockout cells (Extended Data Fig. 4b-d).

4) The claim that “despite distinct phenotypes, these ER morphology alterations in CLIMP63, p180 and KTN1 knockout cells are all microtubule-binding-dependent.” is overstated and is recommended to be tuned down.

Response: We tuned down this statement.

5) Figure 2 is not easy to follow. How are the numbers of Glu modification were determined in these cases should be briefly introduced, at least in the legends. It is also problematic that the purified proteins themselves frequently form bright puncta even without microtubule association, see point 3 for possible solution.

Response: We apologize for not making the method clear. The extent of glutamylation was determined by liquid chromatography-electrospray mass spectrometry (LC-MS). The spectra display the characteristic distribution of masses with peaks separated by 129 Da corresponding to one glutamate. We added description to the figure legends and revised methods part to make it clearer and with all the previous studies using this method referenced.

For the puncta seen in the *in vitro* assay, the proteins do indeed show some nonspecific binding to glass despite our best efforts (glass treatment, use of pluronic, etc.). This sometimes cannot be avoided, especially when working with proteins that have long, disordered and charged amino acid stretches like these proteins do, but it is important to emphasize that all these images were background subtracted for quantification. Assays were performed multiples times, in multiple chambers and on different days and with multiple microtubule preparations and they all showed the same robust differences between the various glutamylated microtubule species. We are including a larger field of view for one of our assays which show that the differences in binding are pretty dramatic and easily picked up by simple visual inspection.

6) It is worrisome, and at the same time expected, that the autophagic defects seen in Climp63 deletion are rather minor. In general, many ER-resident proteins, in addition to the three ones focused here, interact with microtubule and may contribute jointly to the overall homeostasis of the ER morphology and functions. For example, ER sheet-localized Sec61b has been shown to interact with microtubule. It is recommended that the limitation on physiological relevance of the current study to be mentioned in discussion.

Response: Thanks for the reviewer's suggestions. We think the autophagic defects in CLIMP63 KO are not dramatic because this regulation is not directly affecting core autophagy machinery, even not directly affecting lysosome movement, but instead affecting ER distribution which further regulates lysosome positioning.

It is true that many other ER proteins, both on tubular and perinuclear ER, can also bind microtubules, and there is good chance that at least some of them would have specificity toward certain microtubule modifications including glutamylation. Therefore, we added text for this in the Discussion section.

7) Along the same line, it has been shown that p180, but not Climp63 or KTN1, plays a specific role in distribute the ER into axons, which is consistent with the findings here. It would be useful to discuss and compare these findings. In addition, microtubule-interacting proteins may in turn affect microtubule dynamic. In the case here, it is mentioned that microtubule MDR or asymmetry was largely unchanged, but is the specific modification altered when corresponding proteins were deleted or overexpressed?

Response: Axonal microtubules are highly polyglutamylated, so there is very good chance that p180, which is present in axonal ER (Farias, Neuron, 2019, 102:184-201), specifically recognizes polyglutamylated microtubules in axons, thus regulating interplay of axonal ER and microtubules. We added this in Discussion section.

We did see changes in microtubule distribution in the knockout cells (Extended Data Fig. 2m), but the differences are very small. Following the reviewer's suggestion, we further checked the microtubule modifications, and the results show that tubulin modifications are unlikely to be affected by knockout of CLIMP63, p180 or KTN1 (Extended Data Fig. 6a).

Referee #3 (Remarks to the Author):

Zheng and colleagues sought to investigate the role of three ER membrane-bound proteins that bind microtubules: CLIMP63, p180 and KTN1. The study focuses on their implication in shaping the ER through their ability to bind different subpopulations of microtubules defined according to their origin and glutamylation status. CLIMP63 preferentially binds centrosomal microtubules, KTN1 binds preferentially perinuclear polyglutamylated microtubules from either centrosome or Golgi origin, and p180 binds more peripheral microtubules regardless of their origin and which are glutamylated with either short or long chains. The authors used microscopy-based algorithms to measure the changes in ER distribution and defined two relevant parameters as ER mean radius and ER asymmetry. ER distribution impacts organelles positioning. The authors showed that the knock-out of CLIMP63, p180 and KTN1 affects the positioning of several organelles in a similar way to their effect on ER distribution.

The study is interesting and brings some novelty and knowledge on ER shaping. However, the title is misleading as the tubulin code refers to various posttranslational modifications modulating the properties and functions of the microtubule, while the authors focus exclusively on glutamylation. Moreover, tubulin glutamylation modulation is exclusively investigated by overexpressing enzymes, which should be complemented with knock-out or silencing experiments. Finally, the authors focus on the regulation of autophagy through ER-dependent lysosome clustering. The authors should check the fusion between autophagosomes and lysosomes, as well as the lysosomal activity.

Response: We thank the reviewer for the positive comments and have revised the manuscript following the reviewer's suggestions. For instance, we performed RNAi knock down of CCP5 (Extended Data Fig. 7j-l), and also tested autophagosome and lysosome fusion and lysosomal activity (Extended Data Fig. 10b-f). Please see detailed points below.

As for the title, yes, our major finding is about glutamylation, but we also show that CLIMP63 preferentially binds centrosomal microtubules, possibly through additional modification(s). Also, our title proposes that the three ER proteins we studied are tubulin code readers, but we certainly believe that they are not the only tubulin code readers, nor do they read all the tubulin codes. Surely there are other tubulin code readers both on the ER and other cellular compartments.

Specific comments:

1. Effect of siAKAP450 and centrinoneB on MT polymerization should be checked by MT

sedimentation assays to make sure that the effects of CLIMP63, KTN1 and p180 on MT binding is not due to changes in tubulin polymerization.

Response: We performed MT polymerization assay of cells with CNB treatment or AKAP450 knockdown. As shown below, we did not see clear differences:

However, this assay is performed using cell lysates, so may not be exactly reflecting the situation in live cells. It is reasonable that centrosome depletion or Golgi-microtubule depletion could affect microtubule polymerization and dynamics in cells, as the centrosome and Golgi are important microtubule nucleation centers. However, we believe the changes are very minor, and shall have a minimal effect on our conclusions.

In the revised manuscript, we also tested the overall level of polyglutamylation in these cells. Interestingly, depletion of either centrosome or Golgi microtubules led to a decrease in microtubule polyglutamylation (Extended Data Fig. 6b, c). This may explain why KTN1-microtubule binding is decreased in these cells (p180-microtubule binding is not affected, like because the decrease in polyglutamylation is minor) (Fig 1h, i).

2. Most studies on Tubulin glutamylation in the manuscript (Ext Fig10,11, 3d- f) uses overexpression of tubulin glutamylases and deglutamylases (TLL4, TLL7, CCP1, CCP5). These are known to affect MT stability and dynamics. Therefore, it is difficult to say whether the ER distribution changes noted in these cells are due to changes in MT dynamics/stability or whether ER proteins differentially bind MTs depending on their glutamylation state. siRNA mediated KD of tubulin glutamylases will be more appropriate in these studies. ER-MT alignment studies may also be due to changes in MT polymerization.

Response: Thanks for the reviewer's suggestions. The reason why we did not do knockdown assays is that there are 9 TLL glutamylases (TLL1, 2, 4, 5, 6, 7, 9, 11, 13) that are thought to

be mostly functionally redundant to one another, and we couldn't find information about which are predominantly expressed in U2OS cells. To make matters worse, none of them has well-established antibodies. Nevertheless, during the revision, we did order siRNAs for multiple TTLLs, transfected them into cells, and tested whether any of them could change the tubulin glutamylation level. Unfortunately, as shown below, we did not identify any meaningful changes. Also, because we can't find antibodies for these TTLLs, we don't know whether this is because the siRNA knockdown was inadequate, or because the corresponding TTLL did not express in U2OS cells, or the knockdown is compensated by other TTLLs in the cell:

However, we did address this point using a complementary approach, recognizing that among the six CCPs, CCP5 is the only one that can remove monoglutamylation (last "E") (Extended Data Fig. 6g; Rogowski, *Cell*, 2010, 143:564–578). Therefore, we ordered siRNAs against CCP5, and successfully knocked it down (Extended Data Fig. 6j). Consistent with our model, knockdown of CCP5, which leads to increased tubulin glutamylation, results in dispersed ER in wild-type but not p180 KO cells (Extended Data Fig. 6j-l).

Moreover, we recognize that the reviewer is not convinced by the TTLL/CCP overexpression experiments, partly because overexpression of TTLL and CCP give a similar phenotype (dispersed ER, Extended Data Fig. 7e-g). However, this is because, under normal conditions, the perinuclear MT and peripheral MT are very different in terms of glutamylation state, and therefore KTN1 and p180 show different binding specificity to perinuclear and peripheral MTs. With overexpression of either TTLL or CCP, we essentially make the MT identical, thus removing p180 and KTN1's preferential binding toward perinuclear MTs, and making the ER dispersed. Interestingly, during the revision, we identified another cell line, COS7 cells, whose MTs are highly polyglutamylated both perinuclearly and peripherally

(Extended Data Fig. 7a-d). Thus, in COS7 cells, p180 and KTN1 readily bind both perinuclear and peripheral MTs with strong affinity. Indeed, overexpression of CCP6, which decreases MT polyglutamylation, leads to a clustered ER (smaller MDR and higher asymmetry) (Extended Data Fig. 8e-h), strongly supporting our model.

Regarding the reviewer's concern that "ER-MT alignment studies may also be due to changes in MT polymerization." Indeed, high overexpression of TLL7 will cause some microtubule bundling, but without co-overexpression of CLIMP63, p180 or KTN1, we have never seen any cells with prominent ER-MT alignment when overexpressing TLL7 only (Extended Data Fig. 10m). To make this clearer, we overexpressed a well-known MT bundling inducer, PRC1. As shown below, overexpression of PRC1 causes severe MT bundling, but it won't make a tight ER-MT alignment, but instead makes the ER very dispersed, close to what is seen in nocodazole-treated cells, and suggesting that normally MT bundling inhibits ER binding rather than enhancing it.

3. In vitro assay in Figure 2 clearly shows that glutamylated MTs bind p180 and KTN1. Graphs in 2d, and 2e should be merged as authors comment on comparison between protein binding to mono and polyglutamylated MTs. Also these results should be supported with fractionation assays to show the proportion of polymerized MTs available for binding to p180 and KTN1.

Response: We did not merge panels Fig 2d and 2e because the scales are so different (i.e., the polyglutamylated binding enhancement is so much larger than the monoglutamylated). If we put them on the same scale, then the effect of the monoglutamylated is not visible, like this:

Importantly, these are absolute intensity values and thus comparable between the graphs since they were collected at the same exposure, laser intensity, camera gain, TIRF angle etc.

The in vitro binding experiments were done with similar densities of MTs (there is no free tubulin in the system) in the chambers for each of the types of MTs, thus the sedimentation assays are not necessary to confirm this. Moreover, it is not possible to obtain enough material of differentially modified microtubules to perform bulk biochemical assays. These types of TIRF-based microtubule binding assays are widely used in the field (e.g. Tan et al. Nature Cell Bio. 2019, 21:1078-1085; Monroy et al. Dev Cell 2020, 53:60-72; Valenstein and Roll-Mecak, Cell 2016, 164:911-921).

4. In figure 3d-f why was mean radius measured instead of asymmetry in p180/KTN1 double KO as p180 was established to cause ER asymmetry changes in Figure 1.

Response: We use MDR because the phenotype of CCP1 overexpression is an MDR change, and the effect of CCP1, but not p180/KTN1 DKO, is what we want to assess in this section. Thus, with the MDR method, we can see that overexpression of CCP1 increases organelle MDR in wild-type cells, but not p180/KTN1 DKO cells. However, if we use the asymmetry method, then even the wild-type cells won't show any difference, making it difficult to arrive at any firm conclusions.

5. Figure 4 has some interesting observations on how ER and lysosome asymmetry is affected by starvation induced autophagy. However, these observations seem premature and are not directly linked to the rest of the manuscript since they are independent of tubulin PTMs as shown in Ext fig 17. There are also several open questions like why protein levels of CLIMP63 change on starvation, and how this controls p62 levels in fig 4d.

Response: We thank the reviewer for these suggestions. Our previous knockout lines were all based on U2OS cells with stable mEmerald-Sec61 β expression, which occupies the green channel. For the revised manuscript, we re-prepared CLIMP63 knockout cells in parental U2OS cells (Extended Data Fig. 10b), and performed experiments suggested by the reviewer in these new knockout cell lines. The results show that knockout of CLIMP63 causes a delay in autophagosome-lysosome fusion, but it does not change lysosomal activity (Extended Data Fig. 10c-f).

During autophagy, cells first upregulate CLIMP63, which binds centrosomal MTs, and drags the ER and lysosome toward the perinuclear region (Fig. 4a, b). This would bring lysosomes and autophagosomes (derived from the ER) much closer to one another, thus facilitating autophagosome-lysosome fusion. Therefore, in CLIMP63 knockout cell, p62 degradation is compromised.

For the reviewer's comments about "these observations seem premature and are not directly linked to the rest of the manuscript since they are independent of tubulin PTMs", we respectfully disagree. Yes, the MT PTMs themselves are not changed upon starvation, but the cells do change the tubulin code readers, CLIMP63 protein levels, and p180-MT binding. We think the MTs are sort of like the road for organelles to move, while the ER, with its tubulin code readers, are like a control center that senses the changes of environment and give orders to other organelles. Thus, when signals (like starvation) come, it is of course easier to have the control center react than re-building the roads. Therefore, this part of data shows a physiological example of how cells read the tubulin code, and react to the environment, and thus is closely related to the other parts of the manuscript.

6. Ribosome binding of p180 decreases on Puromycin treatment. This leads to an increase in MT binding as shown in PLA assays. These findings need more proof since the MT binding domain and ribosome binding decapeptide repeats are in different parts of the protein sequence as mentioned in extended figure 18.

Response: We apologize for not making our point clearly. In the revised manuscript, we draw a model to depict this (Extended Data Fig. 10i, j). Briefly, p180L contains two potential MT binding domains, one is at its N-terminus, and has selectivity toward glutamylated MTs. The other one, which is composed of dozens of positively-charged decapeptide repeats, is normally

the ribosome-binding domain, and occupied by ribosomes to exclude MT association. When cells are under starvation, ribosomes dissociate (Fig. 4g; Extended Data Fig. 10i-k), then these decapeptide repeats are exposed, conferring MT binding ability. With this mechanism, cells cleverly take advantage of an existing process (ribosome dissociation from the ER due to lack of nutrients) to precisely control ER and lysosome movement during the autophagy response.

Additional comments:

1. Descriptive statistics are missing in the figure legends.

Response: We apologize for the missing information. We have added these in the revised manuscript.

2. Ext. Fig 9c: are the representative images from wildtype or CLIMP63/KTN1 double-knockout cells?

Response: They are wild-type cells. We added labels to clarify this (new Extended Data Fig. 5h, j).

3. Ext. Fig 10b: is the n number referring to the number of cells or number of fields? Significances are missing.

Response: Sorry for the unclear labeling. The n is number of experiments. We now make this more clear in the revised figure legends. We also added significance information to the figure (Now Extended Data Fig. 7e).

4. Fig 3b: CLIMP63 KO still seem to affect the distribution of the lipid droplets; why would that be? How is the double KO CLIMP63/KTN1 affecting the MDR of the different organelles?

Response: We noticed that for the lipid droplets, even though KTN1 KO was not significant, looking at the number, it was still a little higher than wild type (Fig. 3b in previous manuscript). So in the revised manuscript, we analyzed more cells, and this time, KTN1 knockout also has a

significant larger MDR than the wild type cells. Nevertheless, CLIMP63 KO still led to a much more dramatic increase in MDR. We think this could be due to CLIMP63 having some direct roles in lipid droplet morphology. This wouldn't be surprising as many ER-shaping proteins, like REEPs and Atlastin, directly affect lipid droplet morphology (Flak, Human Mutation, 2014, 35:497-504; Klemm, Cell Rep, 2013, 3:1465-75).

We analyzed selected organelle MDR in CLIMP63/KTN1 double knockout cells. As expected, double knockout of CLIMP63/KTN1 led to significantly higher MDR (Extended Data Fig. 9c-e).

5. Fig 3: is tubulin glutamylation itself affecting the localisation of the organelles?

Response: We show that overexpression of CCP1, which deglutamylates MTs, caused both ER and organelles to disperse (MDR increase) (Fig. 3d-f; Extended Data Fig. 7e, f). This MDR increase becomes insignificant in p180/KTN1 double knockout cells (Fig. 3d-f; Extended Data Fig. 7h). We thus speculate that this organelle dispersal effect requires p180 and KTN1 to read the glutamylation change. Nevertheless, we believe CLIMP63, p180 and KTN1 can't be the only tubulin code readers, and other tubulin code readers would surely also be expected to contribute to organelle dynamics upon glutamylation change.

6. Fig 4: is the lysosome asymmetry related to tubulin glutamylation? Why is CLIMP63 protein level affected by starvation?

Response: The ER and lysosomes become clustered because the protein level of CLIMP63 increases significantly during early autophagy (Fig 4c). CLIMP63 preferentially binds centrosomal microtubules (Fig. 1g; Extended Data Fig. 6f). Besides, we did not observe changes in tubulin glutamylation during autophagy (Extended Data Fig.10h). To make this clearer in the revised manuscript, we added a 0.5h time point (Extended Data Fig. 10h) as this is when ER and lysosomes become clustered (Fig. 4a, b). The glutamylation level indeed did not change upon starvation (Extended Data Fig. 10h). As stated in our response to major point 5, we believe it is much easier and controllable for the cells to regulate tubulin code readers rather than MT themselves during environmental changes.

For the moment, we don't know how cells upregulate CLIMP63 during autophagy. We did try several compounds to address this point, such as protein translation inhibitor, lysosomal degradation inhibitor, proteasomal degradation inhibitor, and ER glycosylation inhibitor, but none of these seemed to block this CLIMP63 upregulation (Extended Data Fig. 10a).

7. Ext. Fig 19: Why would the ribosome-binding repeats of p180L be able to bind microtubules when there is a domain already dedicated to this interaction? Further investigations are required there. What makes the binding exclusive?

Response: As we addressed in our response to major point 6, these ribosome-binding repeats do not bind microtubules under normal cellular conditions as they are occupied by ribosomes. Thus, in normal conditions, only the microtubule-binding domain located at the N-terminal of p180 binds glutamylated microtubules and, together with CLIMP63 and KTN1, maintains the proper asymmetric distribution of the ER. During starvation, ribosomes dissociate from p180, exposing the positively-charged ribosome-binding repeats, and p180 gains extra affinity to microtubules. In this way, rather than involving extra mechanisms like post-translational modification, cells take advantage of an existing feature of starvation (ribosome dissociation) to nicely coordinate organelle movement during the autophagy response. In the revised manuscript, we draw a model to better explain this (Extended Data Fig. 10i, j).

Reviewer Reports on the First Revision:

Referee #1 (Remarks to the Author):

In the revision of their manuscript, Zheng et al have made a great effort to address the concerns of the referees. They have performed a number of additional experiments to address the points raised, and provided thoughtful answers to most of the points.

The current manuscript has substantially improved, and major issues have been solved. There are, however, a couple of points that must be addressed before making a final decision:

Point 1c): the authors provide an explanation that actually does not address the point raised here. It was asked to discuss the implications of the previous findings on interdependency of centrosome- and Golgi-derived microtubules (Centrinone-B treatment, which depletes centrosome-derived microtubules, bolsters the formation of other acentrosomal microtubules, for instance from Golgi (Wu et al. 2016 Dev Cell; Gavilan et al. 2018 EMBO rep)) on the current manuscript, especially while interpreting their data shown in Fig. 1g-I. Instead, the authors discussed glutamylation of centrosome-vs Golgi-derived microtubules.

Point 2a): the authors provide a detailed answer to the referee, but did not discuss the point in the manuscript text. This, however, would be important to share these thoughts with the reader. This

is particularly important for the following part of their answer:

“Therefore, hyperglutamylation can indeed enhance CLIMP63-microtubule binding in vitro, but combined with the results that overexpression of CCP1 or CCP5 did not seem to affect CLIMP63’s microtubule binding (Extended Data Fig. 6e, j), and centrosome depletion indeed suppressed CLIMP63 overexpression-mediated ER-MT alignments (Extended Data Fig 6f), we believe that another tubulin modification or interaction is likely responsible for CLIMP63’s preferential binding toward centrosome microtubules.”

Point 3): The authors provide a very nice discussion about the potential enzymatic activity of TLL7, which however is not found in the manuscript text. It will be important to introduce the discussion in the answer to the referee also to the manuscript text:

“We show that KTN1 binds stronger to microtubules that are polyglutamylated than monoglutamylated (Fig.2d, e) even though they have the same mean glutamate numbers, suggesting a preference for the longer glutamate chain for this protein and a weaker binding affinity for monoglutamylation. However, we cannot exclude that the increased affinity for the TLL7 modified microtubules comes from the additional chains that this enzyme initiates on the tubulin tails and not only from the introduction of longer chains. On the other hand, p180 is more sensitive to any increase in glutamate numbers on the tubulin tail, and thus shows robust binding to both monoglutamylated and polyglutamylated microtubules.”

If the authors can address these concerns, a revised version of the manuscript could be considered for publication.

Referee #2 (Remarks to the Author):

In the revision, the authors have addressed most of my concerns. The tests in COS-7 cells and additional discussion are in line with the current model. In points 3 and 5, I meant to use mutants of purified p180/KTN fragments as negative controls for experiments done in Figure 2. If the authors have not done so. It may be useful in follow-up studies.

Referee #3 (Remarks to the Author):

In their revised manuscript, Zheng and colleagues have addressed all my previous comments. However, some details should still be clarified.

1. Regulation of autophagy flux by CLIMP63 and p180 (extended Fig 10): the data doesn’t fully support this statement. (1.1) The autophagy flux (GFP-mCherry-LC3 staining) should be also be assessed in starved condition (EBSS) in order to push the flux; if the lack of CLIMP63 impairs the autolysosome formation, this effect should be more obvious then. (1.2) Similarly, the lysosomal activity should be evaluated in starved condition. (1.3) The implication of p180 in regulating the autophagy flux is not properly assessed. The evaluation of flux and lysosomal activity in p180 KD cells should be done to support the statement, as for CLIMP63 (p62 protein level, GFP-mCherry-LC3, lysosomal activity; in full media and EBSS).
2. The lysosomal movement is difficult to evaluate from fixed samples. It would be more appropriate to use live imaging.
3. The western blot showing the silencing of CCP5 is not the most convincing. It would be appropriate to evaluate the mRNA level. Similarly, the silencing efficiency of the TLLs could be assessed. Alternatively, it could be sensible to assess their function in COS7, as the authors show in their revised manuscript that the MTs are highly polyglutaminated compared to U2OS (assuming that their expression is sufficient).

Author Rebuttals to First Revision:

Referee #1 (Remarks to the Author):

In the revision of their manuscript, Zheng et al have made a great effort to address the concerns of the referees. They have performed a number of additional experiments to address the points raised, and provided thoughtful answers to most of the points.

The current manuscript has substantially improved, and major issues have been solved. There are, however, a couple of points that must be addressed before making a final decision:

Point 1c): the authors provide an explanation that actually does not address the point raised here. It was asked to discuss the implications of the previous findings on interdependency of centrosome- and Golgi-derived microtubules (Centrinone-B treatment, which depletes centrosome-derived microtubules, bolsters the formation of other acentrosomal microtubules, for instance from Golgi (Wu et al. 2016 Dev Cell; Gavilan et al. 2018 EMBO rep)) on the current manuscript, especially while interpreting their data shown in Fig. 1g-I. Instead, the authors discussed glutamylation of centrosome- vs Golgi-derived microtubules.

Point 2a): the authors provide a detailed answer to the referee, but did not discuss the point in the manuscript text. This, however, would be important to share these thoughts with the reader. This is particularly important for the following part of their answer:

“Therefore, hyperglutamylation can indeed enhance CLIMP63-microtubule binding in vitro, but combined with the results that overexpression of CCP1 or CCP5 did not seem to affect CLIMP63’s microtubule binding (Extended Data Fig. 6e, j), and centrosome depletion indeed suppressed CLIMP63 overexpression-mediated ER-MT alignments (Extended Data Fig 6f), we believe that another tubulin modification or interaction is likely responsible for CLIMP63’s preferential binding toward centrosome microtubules.”

Point 3): The authors provide a very nice discussion about the potential enzymatic activity of TTLL7, which however is not found in the manuscript text. It will be important to introduce the discussion in the answer to the referee also to the manuscript text:

“We show that KTN1 binds stronger to microtubules that are polyglutamylated than monoglutamylated (Fig.2d, e) even though they have the same mean glutamate numbers, suggesting a preference for the longer glutamate chain for this protein and a weaker binding affinity for monoglutamylation. However, we cannot exclude that the increased affinity for the TTLL7 modified microtubules comes from the additional chains that this enzyme initiates on the tubulin tails and not

only from the introduction of longer chains. On the other hand, p180 is more sensitive to any increase in glutamate numbers on the tubulin tail, and thus shows robust binding to both monoglutamylated and polyglutamylated microtubules.”

If the authors can address these concerns, a revised version of the manuscript could be considered for publication.

Response: We appreciate the reviewer’s continued enthusiasm and thoughtful suggestions. We have edited our discussion in the revised version in response to the above comments, although due to space limitation of Nature, we have made some of the statements more concise. Please see the last four lines of paragraph 1 (page 10) of Discussion for point 1; partial paragraph at very top of page 8 for point 2; and lines 4-12 of paragraph 2 of Discussion (page 11) for point 3.

Referee #2 (Remarks to the Author):

In the revision, the authors have addressed most of my concerns. The tests in COS-7 cells and additional discussion are in line with the current model. In points 3 and 5, I meant to used mutants of purified p180/KTN fragments as negative controls for experiments done in Figure 2. If the authors have not done so. It may be useful in follow-up studies.

Response: We thank the reviewer for his/her continued enthusiasm and insight. With regards to the in vitro experiments in Figure 2, we use established controls as in previous studies, and have not yet investigated what amino acids or regions are key to their microtubule binding. We are enthusiastic about pursuing additional biophysical and structural studies going forward, as the reviewing outlines

Referee #3 (Remarks to the Author):

In their revised manuscript, Zheng and colleagues have addressed all my previous comments.

However, some details should still be clarified.

1. Regulation of autophagy flux by CLIMP63 and p180 (extended Fig 10): the data doesn't fully support this statement. (1.1) The autophagy flux (GFP-mCherry-LC3 staining) should be also be assessed in starved condition (EBSS) in order to push the flux; if the lack of CLIMP63 impairs the autolysosome formation, this effect should be more obvious then. (1.2) Similarly, the lysosomal activity should be evaluated in starved condition. (1.3) The implication of p180 in regulating the autophagy flux is not properly assessed. The evaluation of flux and lysosomal activity in p180 KD cells should be done to support the statement, as for CLIMP63 (p62 protein level, GFP-mCherry-LC3, lysosomal activity; in full media and EBSS).

Response: We thank the reviewer for his/her enthusiasm as well as for these suggestions. In the revised manuscript, we have repeated these experiments with both CLIMP63 and p180 knockout cells. We also did both control and starved situation (Extended Data Fig. 10b-g).

For the fusion assay using GFP-mCherry-LC3, we apologize for the missing information. We had indeed performed the assay in starved cells in our previous revision, but we neglected to state that in the legend. The difference between wild-type and CLIMP63 knockout is small, likely because this machinery does not directly affect lysosome-autophagosome fusion but instead controls ER distribution, which then regulate lysosome distribution and further, fusion.

2. The lysosomal movement is difficult to evaluate from fixed samples. It would be more appropriate to use live imaging.

Response: We performed live imaging for wild-type, CLIMP63 knockout and p180 knockout cells upon EBSS treatment (Extended Data Movie 1).

3. The western blot showing the silencing of CCP5 is not the most convincing. It would be appropriate to evaluate the mRNA level. Similarly, the silencing efficiency of the TTLs could be assessed. Alternatively, it could be sensible to assess their function in COS7, as the authors show in their revised manuscript that the MTs are highly polyglutaminated compared to U2OS (assuming that their expression is sufficient).

Response: We agree that the CCP5 western blot is not convincing enough. Given that we have not been able to procure better CCP antibodies, in the revised manuscript we used real-time PCR to assess the knockdown efficiency of CCP5 (Extended Data Figure 7k).

For knockdown of TTLLs, we agree with the reviewer that it is particularly appropriate to perform this in COS7 cells, which have very high polyglutamylation levels. The challenge is that there are nine TTLL's, and it would be impractical to deplete all nine. Therefore, we first designed and synthesized RT-PCR primers for all the nine TTLL glutamylases, and checked their expression in COS7 cells, in the hope that only a few TTLLs might be expressed highly and thus targeted for depletion:

It appears that TTLL2 and TTLL13 are hardly detectable in COS7 cells (Primer 1 for TTLL2 did show a band, but it migrates higher than expected, so may be non-specific). We inferred that if all three primer sets can't amplify a clear band at the expected size, then the corresponding TTLL is not expressed in COS7 cells. Thus, we were left with seven TTLLs in COS7 cells.

We then transfected siRNAs targeting the remaining seven TTLLs into COS7 cells, and performed RT-PCR analyses to gauge their knockdown efficiency:

At least some of these siRNAs were effective at the mRNA level. (Note that COS7 is a monkey cell line, so not all the siRNAs targets designed against human genes can be used, though most can. Thus, for some TLLs we find three siRNAs, some only two siRNAs). However, when we performed immunoblotting to examine polyglutamylation levels, we don't see any clear decrease (RT-PCR analyses were performed 48h post-transfection, while WB were performed 72h post-transfection):

We also attempted to transfect pooled siRNAs using those that worked best for each TLLs at the mRNA level, yet still couldn't achieve a reasonable reduction in polyglutamylation levels:

The siRNAs used here are siTTLL1-1, siTTLL5-1, siTTLL 7-1, and siTTLL 11-1, repeated twice.

Despite our best efforts, based on these initial results and the fact that we would potentially need to deplete seven proteins to obtain an acceptable reduction in polyglutamylation in COS7 cells, we hope the reviewer can understand that we do not believe this is feasible.

Reviewer Reports on the Second Revision:

Referee #1 (Remarks to the Author):

In the second round of review, the authors have carefully addressed the remaining points.

It remains slightly unsatisfying that they could not discuss point 1) in more detail in the manuscript due to space restrictions, perhaps they could find a way of elaborate the discussion a bit more?

Other than this, the current manuscript could be considered for publication.

Referee #3 (Remarks to the Author):

The work of Zheng et al., show nicely how ER bounded proteins interact with microtubules to drive organelle distribution. ER residing proteins (CLIMP63, Kinectin and p180) interacts with glutamylated or non-modified tubulin which regulates the ER positioning and distribution. Consequence of this, other organelle's distribution is affected while ER is relocated presumably by the contacts between the membranes. The authors have responded to all questions raised in the previous review.

Author Rebuttals to Second Revision:

Response to Referee #1 (Remarks to the Author):

“In the second round of review, the authors have carefully addressed the remaining points. It remains slightly unsatisfying that they could not discuss point 1) in more detail in the manuscript due to space restrictions, perhaps they could find a way of elaborate the discussion a bit more?”

Response: We appreciate the reviewer's suggestion. This point refers to discussing in more detail implications of published findings on interdependency of centrosome- and Golgi-derived microtubules. As the reviewer noted in a previous review: “Centrinone-B treatment, which depletes centrosome-derived microtubules, bolsters the formation of other acentrosomal microtubules, for instance from Golgi (Wu et al. 2016 Dev Cell; Gavilan et al. 2018 EMBO

rep).” The reviewer was particularly interested in discussion of this point in the context of interpreting data shown in Fig. 1g-i in an earlier version of this manuscript (now Extended Data Fig. 5f-h). In these panels, we show that microtubule association of CLIMP63 is sensitive to centrosome depletion but not Golgi microtubule depletion, while KTN1-microtubule association is sensitive to both; p180-microtubule association is not sensitive to either. In the studies cited by the reviewer, it is shown that eliminating the centrosome via CNB stimulates AKAP450-dependent microtubule nucleation at the Golgi. Even so, the inhibition of microtubule nucleation at the Golgi through AKAP450 depletion does not alter centrosome activity. This is addressed in the Results section where the actions of these interventions are presented along with the data. We focus particularly on implications for the CLIMP63-microtubule association, where there are differential effects.

By addressing this important issue in the Results section rather than in the Discussion, we could accomplish this without lengthening the manuscript.

Reviewer Reports on the Third Revision:

Referee #1 (Remarks to the Author):

The authors have satisfyingly addressed the last concern.